



# Closing the Gap: An Algorithmic Approach to Reconciling In-Situ and Remotely Sensed Aerosol Particle Properties

Sanja Dmitrovic[1,*], Joseph S. Schlosser[2,3,*], Ryan Bennett[4], Brian Cairns[5], Gao Chen[2], Glenn S. Diskin[2], Richard A. Ferrare[2], Johnathan W. Hair[2], Michael A. Jones[6], Jeffrey S. Reid[7], Taylor J. Shingler[2], Michael A. Shook[2], Armin Sorooshian[1,8,9], Kenneth L. Thornhill[2,10], Luke D. Ziemba[2], and Snorre Stamnes[2]

[1]University of Arizona, James C. Wyant College of Optical Sciences, Tucson, Arizona, USA
[2]NASA Langley Research Center, Hampton, Virginia, USA
[3]NASA Postdoctoral Program, NASA Langley Research Center, Hampton, Virginia, USA
[4]Bay Area Environmental Research Institute, Ventura, California, USA
[5]NASA Goddard Institute for Space Studies, New York, New York, USA
[6]University of Massachusetts Lowell, Lowell Center for Space Science and Technology, Lowell, Massachusetts, USA
[7]U.S. Naval Research Laboratory, Marine Meteorology Division, Monterey, California, USA
[8]University of Arizona, Department of Chemical and Environmental Engineering, Tucson, Arizona, USA
[9]University of Arizona, Department of Hydrology and Atmospheric Sciences, Tucson, Arizona, USA
[10]Analytical Mechanics Associates, Hampton, Virginia, USA
[*]These authors contributed equally to this work.
Correspondence to: Joseph S. Schlosser (joseph.s.schlosser@nasa.gov)

**Abstract.** Remote sensors such as lidars and polarimeters are increasingly being used to understand atmospheric aerosol particles and their role in critical cloud and marine boundary layer processes. Therefore, it is essential to ensure these instruments' retrievals of aerosol optical and microphysical properties are consistent with measurements taken by in-situ instruments (i.e., external closure). However, achieving rigorous external closure is challenging because in-situ instruments often 1) provide dry (relative humidity (RH) < 40%) aerosol measurements while remote sensors typically provide retrievals in ambient conditions and 2) only sample a limited aerosol particle size range due to aircraft sampling inlet cutoffs. To address these challenges, we introduce the In Situ Aerosol Retrieval Algorithm (ISARA) in the form of a Python toolkit that converts dry in-situ aerosol data into ambient, humidified data and accounts for the contribution of coarse-mode aerosol particles in its retrievals. We apply ISARA to the NASA Aerosol Cloud meTeorology Interactions oVer the western ATlantic Experiment (ACTIVATE) field campaign data set to perform a preliminary consistency analysis of this campaign's aerosol measurements. Specifically, we compare ISARA-calculated ambient aerosol properties with corresponding measurements from 1) ACTIVATE's in-situ instruments (i.e., internal consistency), 2) Monte Carlo in-situ data simulations (i.e., synthetic consistency), and 3) ACTIVATE's Second Generation High Spectral Resolution Lidar (HSRL-2) and Research Scanning Polarimeter (RSP) instruments (i.e., external consistency). This study demonstrates that 1) appropriate *a priori* assumptions for aerosol particles lead to consistency between in-situ measurements and remote sensing retrievals in the ACTIVATE campaign, 2) ambient aerosol properties retrieved from dry in-situ and the RSP polarimetric data are shown to be consistent for the first time in literature, 3) measurements are externally consistent even when moderately absorbing (imaginary refractive index (IRI) > 0.015) aerosol is present,





and 4) ISARA is limited by probable under-sampling of coarse-mode particles in its calculations. The overall success of this preliminary consistency analysis shows that ISARA can enable systematic, streamlined closure of field campaign aircraft aerosol data sets at large.

## 1 Introduction

Researchers extensively study atmospheric aerosol particles due to these particles' ability to scatter and absorb solar radiation and act as nuclei by which cloud droplets form (i.e., cloud condensation nuclei (CCN)). Since these particles are involved in such important atmospheric processes at weather and climate scales, various ground, airborne, and spaceborne instruments are frequently deployed to measure and/or derive particle optical and microphysical characteristics. As deployed instruments vary greatly in terms of design and individual error characteristics, it is important to verify that measurements from such diverse instrument platforms agree with one another (i.e., external closure) and ultimately provide accurate values for the global climate models to strive for. External closure can involve comparison between in-situ measurements and remote sensing measurements, between measurements from different remote sensing instruments on the same platform, or between measurements from the same or different remote sensing instruments on separate platforms. Analyses of external closure are critical because they can 1) validate the accuracy and uncertainty of satellite and airborne remote sensing aerosol products, 2) create improved a priori aerosol particle properties to better constrain satellite retrievals, 3) aid atmospheric modeling efforts by establishing new applications for remote sensing products such as deriving the hygroscopicity parameter parameter ($\kappa$; Petters and Kreidenweis, 2007), and 4) advance the methods used to derive ambient aerosol properties from in-situ measurements. This study specifically focuses on the state-of-the-art Second Generation High Spectral Resolution Lidar (HSRL-2) and Research Scanning Polarimeter (RSP) remote sensors. The HSRL-2 and RSP are advanced airborne remote sensors that provide vertically-resolved and column retrievals of aerosol optical and microphysical properties, amongst other products. Details on these instruments will be provided in Sect. 2.2.

There have been past efforts to perform closure of aerosol data between airborne in-situ and NASA's HSRL-2 and RSP instruments. For example, previous works have attempted to demonstrate the consistency of airborne HSRL-2 extinction and backscatter measurements and HSRL-2-retrieved aerosol effective radius have been evaluated using collocated in-situ aerosol products (Müller et al., 2014; Sawamura et al., 2017; Pistone et al., 2019). Also, evaluations of RSP retrievals of aerosol optical and microphysical properties have been performed using data from the AErosol RObotic NETwork (AERONET) (e.g., Wu et al., 2015; Fu et al., 2020). There have also been a limited number of case studies successfully making comparisons between RSP retrievals and airborne in-situ data from the Arctic Research of the Composition of the Troposphere from Aircraft and Satellites (ARCTAS; Knobelspiesse et al., 2011) and ObseRvations of Aerosols above CLouds and their intEractionS (ORACLES; Pistone et al., 2019) field campaigns. However, total, fine-mode, and coarse-mode aerosol microphysical properties from RSP have not been systematically evaluated against in-situ aerosol data as of the date of this study.



Despite the important findings from these studies, systematic and streamlined closure of aerosol data sets has not been yet achieved. Closure is challenging to perform due to the increasing volume and complexity of these instrument data. Also, these aerosol data sets can have error coming from plumbing losses, calibration, and most importantly, the alteration of the aerosol's relative humidity (RH) while sampling through aircraft inlets. In order to avoid the complexities of RH variation, in-situ in-

struments will often dry the critical optical property and size distribution measurements of aerosol particles to a RH of ≤40%. Note that 40% RH is the minimum efflorescence point of the majority of atmospheric aerosol species (Li et al., 2014). This drying process is often achieved using Nafion before measurement (Sorooshian et al., 2023).

In contrast to the controlled environments of many in-situ instruments, remote sensors such as lidars and polarimeters retrieve
these particles' properties without altering their RH (i.e.,in ambient conditions). Aircraft in-situ instruments can measure ambient RH and apply it to dry aerosol data sets, but this measurement can have high relative error that can be as high as 15% (Diskin et al., 2002). Adding to this complexity, the measurement of hygroscopcity (e.g., hygroscopic growth function ($f$(RH)) and hygroscopic growth factor ($gf$)) can also have a high uncertainty (Shingler et al., 2016). These sources of error can therefore significantly alter the measured or derived microphysical and optical characteristics of aerosol particles. The other major
difficulty of closure is that in-situ instruments cannot efficiently sample coarse-mode particles due to limitations in the inlet cutoff diameter (i.e., typical cutoff particle diameter (D) < 5 μm). Cloud probes are at times used to estimate coarse mode aerosol properties, but are designed in such a way that the sizing can be highly uncertain (e.g., Reid et al., 2003, 2006). In addition to the coarse-mode sampling limitations, particles are lost between the external inlet of the aircraft and the inlets of the instruments (Baron and Willeke, 2001; Kulkarni et al., 2011). To strive towards and eventually streamline systematic closure
of in-situ and remote sensing aerosol data sets within aircraft field campaigns, this study aims to perform a preliminary closure exercise of in-situ and remote sensing measurements by determining if consistency can be achieved between these disparate data sets. Since this paper details preliminary efforts in external closure, the terms "consistency" and "consistency analysis" will be used throughout the following discussion.

To address these challenges in hopes of advancing towards closure of field campaign data sets at large, the In Situ Aerosol Retrieval Algorithm (ISARA) is introduced. Fundamentally, ISARA is a Python-based algorithm that utilizes the Fortran-based Modeled Optical Properties of Ensembles of Aerosol Particles package (MOPSMAP; Gasteiger and Wiegner, 2018) to allow researchers to use a "common" suite of in-situ instrument types and retrieve the necessary information to calculate the optical and microphysical properties of ambient aerosol particles. More specifically, the algorithm applies hygroscopic growth to dry
in-situ aerosol measurements to convert these data to ambient aerosol properties. Studies such as Ziemba et al. (2013) and Sawamura et al. (2017) have created similar algorithms, but as those study regions had less coarse-mode aerosol, they assume there is no significant coarse-mode aerosol contribution when calculating these properties and therefore limit their calculations to fine-mode aerosol particles only. Note that in this study, the fine-mode particle regime is defined as aerosol particles with dry particle diameter range of 0.09–1.00 μm and the coarse-mode regime is defined as aerosol particles with ambient particle di-
ameters ≥1.0 μm. Coarse-mode species such as dry sea salt and dust are difficult to consider because they can have D > 1.0 μm



(Hussein et al., 2005). Also, dry sea salt is non-spherical, is non-absorbing, and can have large values of $\kappa$ (Sorribas et al., 2015; Ferrare et al., 2023). Similar to dry sea salt, dust can also be non-spherical but can be moderately absorbing and has a complex refractive index (CRI) that is dependent on wavelength (Voshchinnikov and Farafonov, 1993; Veselovskii et al., 2010; Wagner et al., 2012; Sorribas et al., 2015). The ISARA attempts to overcome these limitations by estimating the contribution

of coarse-mode particles to calculated ambient optical and microphysical data. It is hoped that the methods developed in this study serve as a theoretical and measurement framework that can be used fully understand the information train between all manner of measurements and enable systematic closure of field campaign aerosol data.

The ability of ISARA to perform consistency analyses is tested by applying the algorithm to synthesized data as well as data

collected during the NASA Aerosol Cloud meTeorology Interactions oVer the western ATlantic Experiment (ACTIVATE) field campaign, a mission dedicated to characterizing aerosol-cloud-meteorology interactions by using two spatially-synchronized aircraft to provide systematic and simultaneous airborne measurements from 2020 to 2022 (Sorooshian et al., 2019). The spatial synchronization of these aircraft is ideal for performing a consistency analysis between in-situ, lidar, and polarimetric measurements since rigorous spatiotemporal collocation between these data sets can be achieved. The ACTIVATE data set

also enables investigation of numerous atmospheric processes over the western North Atlantic Ocean, including aerosol-cloud interactions that represent the largest uncertainty in estimates of total anthropogenic radiative forcing (Field et al., 2014).

In addition to having a data set with ample amounts of collocated data, special effort was made to sample the North American anthropogenic outflow over the western North Atlantic ocean (Sorooshian et al., 2019). The fine-mode particles of this anthro-

pogenic outflow ($D < 1.0\,\mu m$) are predominately composed of fresh or aged sulfate and organics (Dadashazar et al., 2022a), and the coarse-mode particles ($D \geq 1.0\,\mu m$) are predominately composed of sea salt. While there are cases of diverse aerosol species that are sampled during ACTIVATE such as amines (Corral et al., 2022), dust (Ajayi et al., 2024), and smoke (Soloff et al., 2024), this study serves only to establish the utility of the ISARA and focuses on the less complex fine-mode spherical aerosol species, while the coarse-mode is assumed to be sea salt. This will limit the scope of this study and allow for future

work to be done in analyzing consistency between the ambient aerosol particle properties derived from in-situ and remote sensing measurements of specific aerosol species. The ACTIVATE data set is discussed in more detail in Sect. 2.1.

With this background, the three-fold consistency analysis performed in this paper is outlined here. Section 2 details the methods used to perform the consistency analysis of 1) synthetically-generated aerosol data (i.e., synthetic consistency) 2) ACTIVATE's

in-situ aerosol measurements (i.e., internal consistency), and 3) remote sensing aerosol retrievals from the HSRL-2 and RSP (i.e., external consistency). These methods are presented in the following order: 1) ACTIVATE mission, 2) cloud filtering of in-situ data, 3) ISARA methodology including retrieval descriptions for dry CRI and $\kappa$, 4) synthetic in-situ data generation, 5) HSRL-2 and RSP data processing including cloud filtering of remote sensing data and matching HSRL-2 data to the RSP resolution, 6) collocation of in-situ data to the remote sensing data. Section 3 presents results of synthetic, internal, and ex-

ternal measurement consistency. The external measurement consistency analysis focuses on ensuring consistency under ideal





conditions defined as: i) good data spatiotemporal collocation (defined as a spatiotemporal separation of <6 min and <15 km) between the platforms, ii) a single observed aerosol layer dominated by spherical fine-mode particles (e.g., anthropogenic outflow), and iii) absence of clouds within remote sensing retrievals and in-situ measurements. Section 4 summarizes key points of this study and suggests potential avenues for future work.

## 2 Methods

### 2.1 ACTIVATE Mission Description

The ACTIVATE featured 162 coordinated science flights across six ACTIVATE deployments that occurred from 14 February 2020 to 18 June 2022. The six ACTIVATE deployments occurred between the following dates:

1. 14 February – 12 March 2020,

2. 13 August – 30 September 2020,

3. 27 January – 2 April 2021,

4. 13 May – 30 June 2021,

5. 30 November 2021 – 29 March 2022,

6. 3 May 2022 – 18 June 2022.

During the first five and a half ACTIVATE deployments, the majority of these joint flights were carried out using NASA Langley Research Center in Virginia as a base of operations. The final half of the sixth ACTIVATE deployment featured Bermuda as the base of operations. The ACTIVATE methodology and data set are described in more detail in Sorooshian et al. (2023). The ACTIVATE mission follows previous studies that aim to study aerosol-cloud interactions in the dynamic western North Atlantic environment (e.g., Quinn et al., 2019; Sorooshian et al., 2020; Dadashazar et al., 2021b, a; Corral et al., 2021; Painemal et al., 2021).

An important feature of the ACTIVATE data set is the extensive collocated advanced passive and active remote sensing and in-situ data. The ACTIVATE aircraft executed flights that can be broadly categorized into two mission types: "process studies" and "statistical surveys". This study focuses on statistical survey flights, where the lower-flying HU-25 Falcon aircraft collected data at various vertical levels (i.e., legs) in and above the marine boundary layer (MBL) for ~3.3 hours (Dadashazar et al., 2022b). During these statistical surveys the Falcon would also make occasional vertical profiles (i.e., controlled ascents and descents) through the atmosphere. Simultaneously, the higher-flying King Air at approximately 9 km would conduct remote sensing and launch dropsondes while being spatially coordinated with the Falcon. These flights comprised 90% of missions and allowed for the efficient in-situ characterization of gas, cloud, aerosol, and meteorological quantities of the MBL across multiple flights and deployments (Dadashazar et al., 2022b; Sorooshian et al., 2023). As noted previously, the focus on spatial




coordination of the two aircraft during the flights is beneficial for external consistency analysis, which is later described in Sect. 2.5.

## 2.2 Measurements

### 2.2.1 Remote Sensor Instrument Descriptions

The Second Generation High Spectral Resolution Lidar, HSRL-2, is an active lidar remote sensor that provides vertically-resolved profiles of various aerosol and cloud properties for campaigns such as the Cloud, Aerosol and Monsoon Processes Philippines Experiment (CAMP2Ex; Reid et al., 2023), Deriving Information on Surface Conditions from COlumn and VERtically Resolved Observations Relevant to Air Quality (DISCOVER-AQ; Sawamura et al., 2017), and ACTIVATE (Sorooshian et al., 2023). Unlike standard elastic backscatter lidars such as Cloud-Aerosol Lidar with Orthogonal Polarization (CALIOP),

the HSRL-2 has the ability to measure total and molecular backscatter separately from which aerosol backscatter, extinction, and lidar ratio can be derived (Hair et al., 2008; Burton et al., 2016, 2018). The HSRL-2 has channels at 355 and 532 nm with an additional elastic backscatter channel at 1064 nm where the ambient extinction coefficient at 355 and 532 nm is retrieved from the measured lidar ratio. In this study, the HSRL-2 measurement of total ambient extinction coefficient at 532 nm serves as the standard relative to the in-situ-derived ambient extinction coefficient as was done in Sawamura et al. (2017). The HSRL-2

products include ambient vertically−resolved lidar backscatter coefficient and linear depolarization ratio (LDR) at wavelengths of 355, 532, and 1064 nm and extinction coefficient at 355 and 532 nm wavelengths (Fernald et al., 1984; Hair et al., 2008; Burton et al., 2018). The HSRL-2 field of view is 1 mrad, which corresponds to a 9 m footprint for an aircraft at 9 km altitude.

Complementing the HSRL-2 is the RSP, which is a passive polarimetric remote sensor that uses highly accurate multispectral

and hyperangular photopolarimetric measurements to characterize aerosol and cloud properties (Cairns et al., 1999, 2003). The aerosol products are based on an optimal estimate using the Research Scanning Polarimeter Microphysical Aerosol Properties from Polarimetery (RSP-MAPP) algorithm (Stamnes et al., 2018). Fine- and coarse-mode aerosol optical and microphysical properties are retrieved using seven channels that measure the total and polarized radiance across the visible-shortwave spectrum (wavelength range = 410–2260 nm) with over 100 viewing angles between $\pm55°$. The RSP has a field of view of 14 mrad,

which results in a 126 m along-track footprint for an aircraft at 9 km altitude. As a result, the RSP provides accurate column-averaged retrievals of aerosol optical and microphysical properties such as real refractive index (RRI), IRI, effective radius ($r_{eff}$), and single scattering albedo (SSA).

The relevant King Air products are described in Table 1 along with their associated vertical resolutions, temporal resolutions,

and uncertainties. Note that the native resolution of the extinction coefficients measured by the HSRL-2 is 225 m vertically and 60 seconds temporally. The provided HSRL-2 coefficients are smoothed from the subsampled resolution of 15 m × 1 s to the native resolution 225 m × 60 s by taking the arithmetic mean of all subsampled points within each native bin. In addition to analyzing the consistency of the standard HSRL-2 and RSP aerosol particle products, this study will analyze the consistency



of the novel vertically-resolved aerosol particle number concentration ($N_a$) (Schlosser et al., 2022). Complete details on the

derivation of vertically-resolved $N_a$ are discussed in Schlosser et al. (2022), but note that this quantity relies on the HSRL-2-derived aerosol extinction coefficient at 532 nm and the RSP-derived aerosol extinction cross section at 532 nm.

**Table 1.** Summary of the King Air payload including relevant Second Generation High Spectral Resolution Lidar (HSRL-2) and Research Scanning Polarimeter (RSP) ambient aerosol particle products with associated native resolutions and uncertainties.

| Instrument | Parameter Description | Vertical/Temporal Resolution | Uncertainty |
|---|---|---|---|
| Second Generation High Spectral Resolution Lidar (HSRL-2) | Total aerosol particle extinction coefficient at 355, 532, and 1064 nm wavelengths | 225 m / 60 seconds | 0.01 km$^{-1}$ |
| | Total backscatter coefficient at 355, 532, and 1064 nm wavelengths | 30 m / 10 seconds | 0.2 Mm$^{-1}$ sr$^{-1}$ |
| | Total linear depolarization ratio (LDR) at 355, 532, and 1064 nm wavelengths | 225 m / 10 seconds | 2 − 5%* |
| | Column aerosol optical depth (AOD) at 355 and 532 nm wavelengths | – / 60 seconds | 0.02 |
| Research Scanning Polarimeter (RSP) | Total, fine-mode, and coarse-mode hyper-spectral column AOD from 410 to 2250 nm wavelengths | – / 4.167 seconds | 0.04 / 0.015** |
| | Column−averaged total, fine-mode, and coarse-mode aerosol particle number concentration ($N_a$) | – / 4.167 seconds | 10 − 100% |
| | Column−averaged total, fine-mode, and coarse-mode aerosol particle size distribution effective radius ($r_e$) | – / 4.167 seconds | 0.02 / 0.15 µm** |
| | Column−averaged total, fine-mode, and coarse-mode aerosol particle extinction cross-section ($\sigma_{ext}$) | – / 4.167 seconds | – |
| | Column−averaged total, fine-mode, and coarse-mode single scattering albedo (SSA) | – / 4.167 seconds | 0.02 / 0.04** |
| | Column−averaged fine-mode real refractive index (RRI) | – / 4.167 seconds | 0.02 |
| | Column−averaged fine-mode imaginary refractive index (IRI) | – / 4.167 seconds | 0.02 |

* Uncertainty values are approximate and dependent on scattering levels.

** Uncertainty values are for the fine-mode / coarse-mode, respectively.





### 2.2.2 In-Situ Instrument Descriptions

In-situ measurements of size-resolved dry $N_a$ are taken from the Scanning Mobility Particle Sizer (SMPS) (Model 3085 DMA, Model 3776 Condensation Particle Counter (CPC), and Model 3088 Neutralizer; TSI, Inc.) and a Laser Aerosol Spectrometer (LAS) (Model 3340; TSI, Inc.). The SMPS measures concentrations of particles with mobility D ranging in size from 2.97 to 94 nm at a 45-second temporal resolution (Moore et al., 2017). The LAS measures concentrations of particles with optical equivalent D ranging in sizes from 94 to 7500 nm at a 1-second temporal resolution (Froyd et al., 2019). The LAS sampled particles that were actively dried with a 6" Monotube dryer (Perma-Pure, Model 700) for all flights except the 30 from 14 May through 30 June, 2021 that were only dried passively. All SMPS data relied on passive drying from ram heating and a generally warmer cabin temperature than ambient air. Note that all drying was done to an RH of $\leq$40%. The $N_a$ measurements provided by the SMPS and LAS are provided at standard temperature and pressure (273.15 K and 1013 mb).

While the LAS has a measurement range up to 7500 nm, the maximum cutoff D of the sample inlet prevents the measurement of particles with ambient D > 5000 nm for ACTIVATE (McNaughton et al., 2007; Chen et al., 2011). The effective size cut at $D = 5000$ nm for all 2020 data. For this data only include particles with a maximum of dry optical D up to 3488 nm were used. This is done because the next logarithmically-spaced bin starts at 3488 nm and extends beyond the limit for efficient transmission into the isokinetic inlet. For 2021 and 2022 data sets, a cyclone was installed upstream of the nephelometers that results in a 1000 nm cutoff for those data. The impact on the absorption coefficient from particles above 1000 nm is assumed to be negligible in the calculation of extinction coefficients. It is noted that the LAS particle sizing is calibrated using an assumed dry CRI and shape. The systematic error introduced by using an assumed dry CRI and shape is minimized by performing the LAS calibration with respect to spherical ammonium sulfate particles with dry CRI of $1.53 + 0i$, which is among the most common aerosol species (Ebert et al., 2004; Sawamura et al., 2017). The SMPS sizing is calibrated using National Institute of Standards and Technology (NIST)-traceable polystyrene latex spheres, while size-dependant concentrations were calibrated in the laboratory using monodispersed aerosol and a reference CPC. These calibrations resulted in good stitching between the SMPS and LAS distributions and good closure between the integrated number concentrations measured by ancillary CPC measurements (see Figure 7 of Sorooshian et al., 2023).

The in-situ optical measurements are taken by the nephelometer (Model 3563; TSI, Inc.) and the tricolor Particle Soot Absorption Photometer (PSAP) (Radiance Research) (Sorooshian et al., 2023). The nephelometer measures dry ($\leq$40%) and humidified (~85% RH) particle scattering coefficients ($C_{\mathrm{scat,RH=40}}$ and $C_{\mathrm{scat,RH=85}}$) at wavelengths equal to 450, 550, and 700 nm at a 1-second temporal resolution (Ziemba et al., 2013) while the PSAP measures dry absorption coefficient ($C_{\mathrm{abs,RH=40}}$) at 470, 532, and 660 nm at a 1-second temporal resolution (Mason et al., 2018). The scattering coefficients are corrected for truncation errors using Anderson and Ogren (1998) and the absorption coefficients are corrected for a variety of errors using Virkkula (2010). The parallel dry and humidified nephelometer deployment allows for scattering coefficients to be adjusted to any RH up to saturation (RH = 99%) through computation of the $f$(RH). The $f$(RH) product used in this study is defined as the ratio





between the scattering coefficients at RH = 80% and at RH = 20%, which is represented with the following equation:

$$f(\mathrm{RH}) \equiv f\left(\frac{\mathrm{RH}=80\%}{\mathrm{RH}=20\%}\right) = \frac{C_{\mathrm{scat,RH=80}}}{C_{\mathrm{scat,RH=20}}}. \tag{1}$$

For the purposes of this study we will only use scattering and absorption data when their signals are above $1\,\mathrm{Mm}^{-1}$ in each of the 6 channels.

225

Measurements of ambient liquid water content (LWC) and cloud drop number concentration ($N_\mathrm{d}$) are used to classify in-situ data as cloud-free, ambiguous, or cloud. This classification becomes important because ISARA retrievals are performed for cloud-free cases. Ambient LWC and $N_\mathrm{d}$ are both derived from ambient particle size distribution measured by a Cloud Droplet Probe (CDP) (Droplet Measurement Technologies, Sinclair et al., 2019). The CDP can measure particles in the ambient D size range of 2000–50000 nm. Measurements where LWC is between 0.001 and 0.02 g m$^{-3}$ and $N_\mathrm{d}$ is between 5 and 50 cm$^{-3}$ are classified as ambiguous, i.e., not entirely cloud-free. Therefore, measurements are considered cloud-free where LWC and $N_\mathrm{d}$ are less than 0.001 g m$^{-3}$ and 5 cm$^{-3}$, respectively (Schlosser et al., 2022).The ambient aerosol particle size distribution measured by the CDP also helps account for coarse aerosol particles when calculating the final properties of the ambient aerosol particles (see Section 2.3).

235

To round off the suite of in-situ instruments is the Diode Laser Hygrometer (DLH), which provides ambient RH data. Ambient RH is used for the final calculation of ambient aerosol properties as described in Sect. 2.3. Note that the DLH measures water vapor density, which is used with ambient pressure and temperature data to derive ambient RH to an relative accuracy of 15% of the measured RH (Diskin et al., 2002). The relevant Falcon measurements are described in Table 2 along with their associated size ranges, temporal resolutions, and uncertainties.



**Table 2.** Summary of the relevant Falcon measurements and payload with associated size ranges, resolutions, and one standard deviation uncertainties.

| Measurement | Instrument | Systematic uncertainty (accuracy) | Random uncertainty (precision) | Size range (nm) | Native time resolution (s) |
|---|---|---|---|---|---|
| Dry logarithmic size-resolved aerosol particle number concentration ($n^{\circ}$) | Scanning Mobility Particle Sizer (SMPS) | 10% | – | 2.97 – 94.0 | 45 |
| Dry $n^{\circ}$ | Laser Aerosol Spectrometer (TSI LAS–3340). | 10% | – | 93.9 – 3487.5 | 1 |
| Dry scattering coefficient at 450, 550, and 700 nm wavelengths. | Nephelometer at RH ≤40% (TSI–3563) | 10% | $2\,\mathrm{Mm}^{-1}$ | <1000* | 1 |
| Humidified scattering coefficient at 450, 550, and 700 nm wavelengths | Nephelometer at RH ≃80% (TSI–3563) | 10% | $2\,\mathrm{Mm}^{-1}$ | <1000* | 1 |
| Dry absorption coefficient at 470, 532, and 660 nm wavelengths | Tricolor Particle Soot Absorption Photometer (PSAP) | 7.5% | $1\,\mathrm{Mm}^{-1}$ | <5000 | 1 |
| Relative humidity (RH) | Diode Laser Hygrometer (DLH) | 7.5% | - | - | 0.05 |
| Liquid water content (LWC), cloud drop number concentration ($N_{\mathrm{d}}$), and coarse-mode ambient $n^{\circ}$ | Cloud Droplet Probe (Droplet Measurement Technologies CDP) | 10% | - | 2000 – 50000 | 1 |

* For ACTIVATE 2020, this was <5000 nm. See Sect. 2.2 for details.

## 2.3 In Situ Aerosol Retrieval Algorithm (ISARA) Description

The first step of this algorithm is to match all in-situ data to the lowest time resolution of the suite of instruments. In the case of ACTIVATE, the SMPS has the lowest time resolution of 45 seconds. The 45-second resolution is a shortcoming of the SMPS on an aircraft. As such, the external consistency analysis is most useful from vertical profiles where the vertical extent is more than 1000 m. The data merge is handled by the NASA Airborne Science Data for Atmospheric Composition online merge tool (see www-air.larc.nasa.gov). While there might be more effective methods for averaging, the 1-second data is averaged using the methods standardized by the merging tool. After this step, ISARA is used to calculate the aerosol optical and microphysical properties relevant to this study. Computation of scattering and absorption coefficients is accomplished using MOPSMAP (Bohren and Huffman, 2008; Gasteiger and Wiegner, 2018). Note that the size range for the number concentration



measurements used in these retrievals is 94–3487.5 nm for ACTIVATE 2020 data, however this range is truncated to 94–1000 nm for ACTIVATE 2021-2022. For the purposes of this study, MOPSMAP is limited to Mie theory to perform these calculations. Mie theory assumes that the aerosol particle is a homogeneous dielectric sphere with a complex refractive index (CRI). An aerosol particle's CRI is a complex number defined as follows:

$$\mathrm{CRI} = \mathrm{RRI} + \mathrm{IRI} \times i, \tag{2}$$

where RRI and IRI are the real and imaginary components of CRI, respectively. For the purposes of this study we will assume that the CRI does not change with wavelength (e.g., the CRI is spectrally flat), which is a good assumption for organic and sulphate aerosol species observed for much of ACTIVATE within the 450–700 nm range of wavelengths (Bain et al., 2019). MOPSMAP first calculates spectral scattering and absorption efficiencies ($Q_{\mathrm{scat}}$ and $Q_{\mathrm{abs}}$) based on the particle diameter, wavelength, and CRI and then integrates these efficiencies over the aerosol particle size distribution to compute the scattering and absorption coefficients $C_{\mathrm{scat}}$ and $C_{\mathrm{abs}}$:

$$C_{\mathrm{scat,abs}}(\lambda) = \int_{\mathrm{d\log D_{min}}}^{\mathrm{d\log D_{max}}} \left[ \frac{\pi \mathrm{D}^2}{4} \times Q_{\mathrm{scat,abs}}(\lambda, \mathrm{CRI}, \mathrm{D}) \times n^{\mathrm{o}}(\mathrm{D}) \right] \mathrm{d\log D}, \tag{3}$$

where D is particle diameter, $\lambda$ is the wavelength of the measurement source, $\mathrm{d\log D}$ is the logarithmic difference between the upper and lower diameter cutoffs of each bin, and $n^{\mathrm{o}}$ is the logarithmic size-resolved aerosol particle number concentration. The integral bounds $\mathrm{d\log D_{min}}$ and $\mathrm{d\log D_{max}}$ correspond to the $\mathrm{d\log D}$ of the smallest and largest bins of the particle size distribution. The term $n^{\mathrm{o}}$ is used per convention to represent the following:

$$n^{\mathrm{o}} = \frac{\mathrm{dN}}{\mathrm{d\log D}} \tag{4}$$

and $\mathrm{d\log D}$ represents the following equation:

$$\mathrm{d\log D} \equiv \mathrm{d\log} \frac{\mathrm{D_2}}{\mathrm{D_1}} = \log \mathrm{D_2} - \log \mathrm{D_1}. \tag{5}$$

As the remote sensors are more sensitive to particle volume concentration, this work also discusses the logarithmic size-resolved aerosol particle volume concentration ($v^{\mathrm{o}}$), which is defined as follows for spherical particles:

$$v^{\mathrm{o}} = n^{\mathrm{o}} \times \frac{\pi \mathrm{D}^3}{6} = \frac{\mathrm{dV}}{\mathrm{d\log D}}. \tag{6}$$

While MOPSMAP is an accurate forward model for calculating particle optical coefficients, ISARA is a retrieval algorithm that can invert dry and humidified (i.e., wet) aerosol measurements to retrieve the dry aerosol particle properties while accounting for changes in optical properties due to hygroscopicity, allowing for the derivation of ambient aerosol properties. The first main step of the ISARA retrieval is calculating a total dry CRI since this is a critical parameter for $Q$ and $C$ as mentioned previously. This step focuses on retrieving CRI. Eq. 3 is rewritten as follows to denote the calculation of dry parameters (Eq. 7):

$$C_{\mathrm{scat,abs,dry}}(\lambda) = \int_{\mathrm{d\log D_{min,dry}}}^{\mathrm{d\log D_{max,dry}}} \left[ \frac{\pi \mathrm{D_{dry}^2}}{4} \times Q_{\mathrm{scat,abs}}(\lambda, \mathrm{CRI_{dry}}, \mathrm{D_{dry}}) \times n^{\mathrm{o}}(\mathrm{D_{dry}}) \right] \mathrm{d\log D_{dry}}. \tag{7}$$



To provide the dry size distribution, the user inputs stitched LAS and SMPS size distribution data to represent the full range of
aerosol particle sizes measured in ACTIVATE. Figure 1 shows logarithmic size-resolved aerosol particle number concentration
($n^\mathrm{o}$) and logarithmic size-resolved aerosol particle volume concentration ($v^\mathrm{o}$) as a function of dry particle diameter ($\mathrm{D_{dry}}$) from
all of the ACTIVATE 2020–2022 data. It is observed that the ACTIVATE region is mostly comprised of fine-mode particles
with very low concentrations of coarse-mode particles.

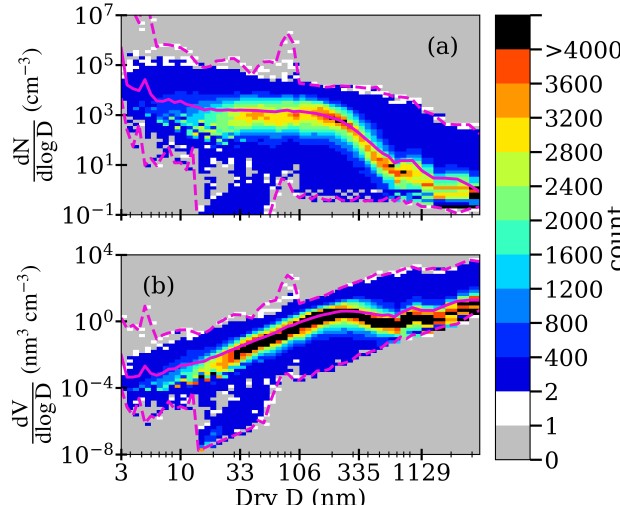

**Figure 1.** Heat map plot of logarithmic size-resolved aerosol particle (a) number concentration $\left(n^\mathrm{o} = \frac{\mathrm{dN}}{\mathrm{d\log D}}\right)$ and (b) volume concentration
$\left(v^\mathrm{o} = \frac{\mathrm{dV}}{\mathrm{d\log D}}\right)$ versus dry particle diameter (D) measured from all ACTIVATE 2020–2022 data which is comprised of 34015 size distribution
measurements at 45 second resolution. The solid line represents the arithmetic mean of each bin, the bottom and top dashed lines represent
the minimums and maximums of each bin, respectively, and the color bar indicates density of points in a given area of the plot.

A set of scattering and absorption coefficients are then calculated by iterating through dry RRI and IRI. The IRI is iterated
from 0.00 to 0.08 in increments of 0.001, which is a range suited for typical aerosol particles in the ACTIVATE region. The
RRI is iterated at 1.52, 1.53 and 1.54 to capture small deviations in RRI from the 1.53 assumed by the LAS. Given that the
scattering is dominated by particles in the LAS size range, we expect to get a good agreement between the ISARA-derived
scattering coefficient and that measured by the nephelometer. Note that this process uses the mid-point particle diameters $n^\mathrm{o}$
from each SMPS and LAS channel. After the set of scattering and absorption coefficients are calculated, ISARA retrieves
a final value of total dry CRI ($\overline{\mathrm{CRI}}$) by taking the average of all valid CRI values. For a CRI to be valid for averaging, all
three of the computed scattering coefficients must be within 20% of the corresponding measured dry scattering coefficient
$\left(\frac{|C_\mathrm{calc}-C_\mathrm{scat,RH=40}|}{C_\mathrm{scat,RH=40}} < 0.2\right)$ and all three of the calculated absorption coefficients must be within $1\,\mathrm{Mm^{-1}}$ of the measured
absorption coefficients $\left(|C_\mathrm{calc} - C_\mathrm{abs,RH=40}| < 1\,\mathrm{Mm^{-1}}\right)$. This method has been adapted from Sawamura et al. (2017) to in-



 clude all three wavelengths. A summary of this retrieval step is provided in Fig. 2.

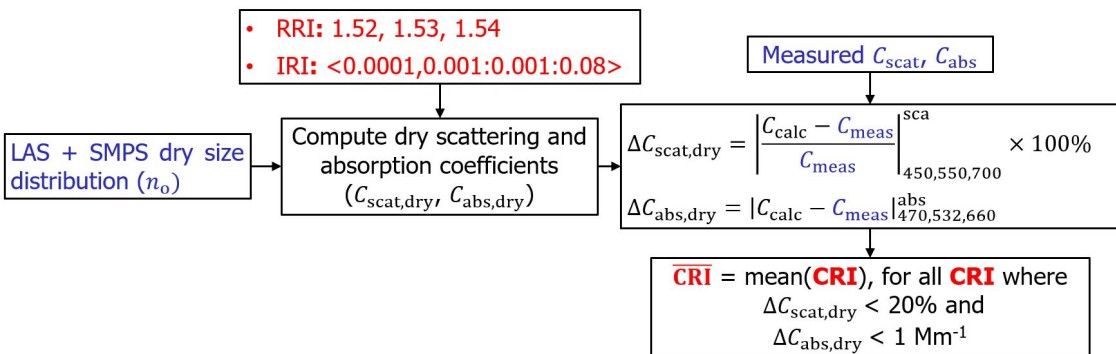

**Figure 2.** Flow chart of the dry CRI retrieval procedure. Blue text represents measured values, black text represents MOPSMAP-calculated values, and red text represents ISARA retrievals. The term $C_{\text{calc}}$ indicates calculated $C_{\text{scat,dry}}$ and $C_{\text{abs,dry}}$ while $C_{\text{meas}}$ indicates $C_{\text{scat,RH=40}}$ and $C_{\text{abs,RH=40}}$ from the nephelometer and PSAP instruments, respectively.

A final check is performed to ensure $\overline{\text{CRI}}$ results in scattering and absorption coefficients that meet the same thresholds of 20% and $1\,\text{Mm}^{-1}$, respectively. Now that dry CRI has been determined, it is then necessary to retrieve the hygroscopicity parameter ($\kappa$). Since the retrieval of $\kappa$ relies on the same Mie theory principles as the previous step, Eq. 3 is rewritten to represent wet parameters (Eq. 8).

$$C_{\text{scat,abs,wet}}(\lambda) = \int\limits_{d\log D_{\text{min,wet}}}^{d\log D_{\text{max,wet}}} \left[ \frac{\pi D_{\text{wet}}^2}{4} \times Q_{\text{scat,abs}}(\lambda, \text{CRI}_{\text{wet}}, D_{\text{wet}}) \times n^{\text{o}}(D_{\text{wet}}) \right] d\log D_{\text{wet}}. \tag{8}$$

For the retrieval of $\kappa$, the forward-modeled humidified scattering coefficients are computed by adjusting for the impact that water uptake has on the increase in particle diameter as a result of hygroscopic growth to determine the humidified particle diameters and dry CRI. For spherical particles, both the scattering coefficients and particle diameters are related to $\kappa$ by the *gf*, which is defined as the ratio between the humidified and the dry particle diameters:

$$gf = \frac{\text{Humidified Diameter}}{\text{Dry Diameter}} = \frac{D_{\text{wet}}}{D_{\text{dry}}}. \tag{9}$$

The *gf* is related to $\kappa$ by RH via the following parameterization from Petters and Kreidenweis (2007):

$$\frac{\text{RH}}{\exp\left(\frac{A}{D_{\text{dry}}gf}\right)} = \frac{gf^3 - 1}{gf^3 - (1-\kappa)}, \tag{10}$$

where *A* is the water activity. Water activity is a temperature-dependent function defined as follows:

$$A = \frac{4\sigma_{s/a}M_w}{R T \rho_w}, \tag{11}$$





where $\sigma_{s/w}$, $M_w$, R, $T$, and $\rho_w$ are surface tension of solute (i.e., aerosol) to water, molecular weight of water, ideal gas constant, temperature, and density of water, respectively. The values of $\sigma_{s/w}$, $\rho_w$, and $T$ are assumed to be $0.072\,\mathrm{J\,m^{-2}}$, $1000\,\mathrm{kg\,m^{-3}}$, and $298.15\,\mathrm{K}$, respectively (Petters and Kreidenweis, 2007). For particles larger than 80 nm, this equation becomes (Zieger et al., 2013):

$$\left(\frac{\mathrm{D_{wet}}}{\mathrm{D_{dry}}}\right)^3 = gf^3 = 1 + \kappa \times \frac{\mathrm{RH}}{1-\mathrm{RH}}, \tag{12}$$

where RH is set to 80%. Conceptually, hygroscopic growth results in the size distribution being shifted to the right by *gf* and the distribution will widen if the size distribution is graphed with diameter on the x-axis. To account for the impact that water has on dry CRI, the humidified CRI is assumed to be the volume-weighted average between dry RRI and the CRI of water ($\mathrm{CRI_{H_2O}} = 1.33 + 0i$). The volume-weighted mixing model is used because it was found to be the most robust of a variety of possible mixing models by (Nessler et al., 2005). With this model, CRI can be written as a function of *gf* (and consequently $\kappa$ using Eq. 10) as follows:

$$\mathrm{CRI}(gf) \approx \frac{\mathrm{CRI_{dry}} + \mathrm{CRI_{H_2O}} \times (gf^3 - 1)}{gf^3}. \tag{13}$$

With these relationships established, a set of wet scattering coefficients (i.e., $C_{\mathrm{scat,wet}}$) at 550 nm can now be calculated by iterating through $\kappa$ from 0.00 to 1.40 in increments of 0.001, also a range typical of ACTIVATE's measured aerosol particles.

In addition to calculating the set of forward-modeled $C_{\mathrm{scat,wet}}$ that corresponds to the range of $\kappa$ values the measured aerosol particle size distribution, and retrieved CRI, *f*(RH) is used to derive the "measured" $C_{\mathrm{scat,wet}}$ (i.e., $C_{\mathrm{scat,RH=80}}$). The *f*(RH) is derived from the tandem nephelometers as detailed in Sect. 2.2. The $C_{\mathrm{scat,RH=80}}$ is obtained by multiplying *f*(RH) with the dry measured scattering coefficient at 550 nm. After this step is performed, the smallest $\kappa$ values are taken for computed $C_{\mathrm{scat,wet}}$ that are within 1% of $C_{\mathrm{scat,RH=80}}$ ($\Delta C_{\mathrm{scat,wet}} < 1\%$). The smallest of these $\kappa$ values is then taken to be the single effective $\kappa$ ($\overline{\kappa}$) for the fine-mode aerosol particles. A summary of this retrieval process is summarized below (Fig. 3).





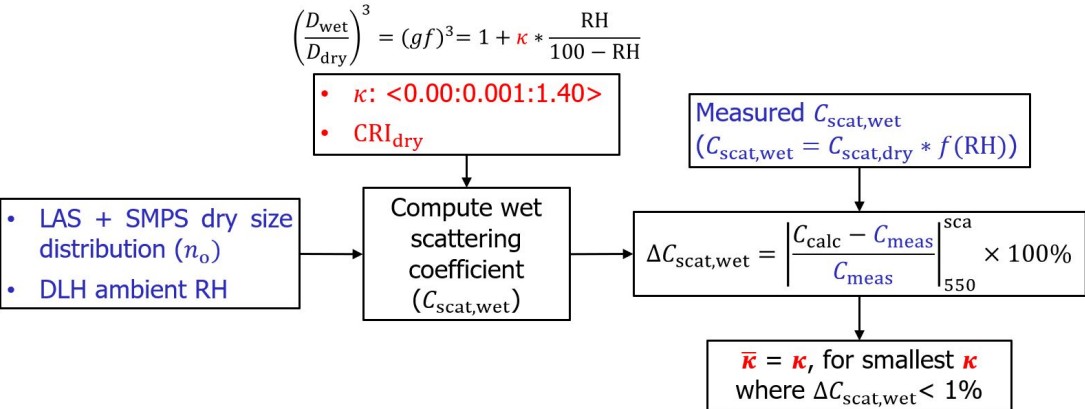

**Figure 3.** Flow chart of the hygroscopicity parameter ($\kappa$) retrieval procedure. Blue text represents measured values, black text represents MOPSMAP-calculated values, and red text represents ISARA retrievals. $C_{\text{calc}}$ refers to the calculated scattering coefficient and $C_{\text{meas}}$ refers to the wet scattering coefficient from tandem nephelometers derived from $f$(RH) relation.

Distributions of IRI and $\kappa$ are then calculated by ISARA for all ACTIVATE data (Fig. 4). It is seen that aerosol particles in the ACTIVATE region generally have low absorption ($C_{\text{abs}} \leq 1\,\text{Mm}^{-1}$) and low hygroscopicity ($\kappa \leq 0.3$).

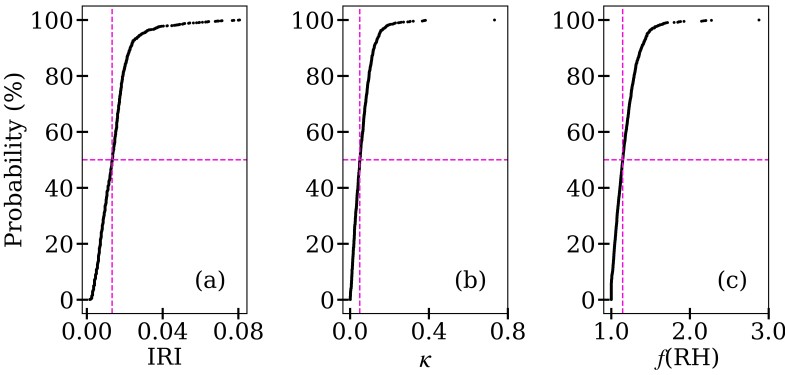

**Figure 4.** Cumulative probability distribution plots of retrieved (a) imaginary refractive index (IRI), (b) hygroscopicity parameter ($\kappa$), and (c) hygroscopic growth function ($f$(RH)) derived from ACTIVATE 2020–2022 data. The intersection of the dashed magenta lines marks the medians of the data sets, which are 0.013, 0.049, and 1.15 for IRI, (b) $\kappa$, and (c) $f$(RH), respectively.

After IRI and $\kappa$ are determined, these parameters are combined with the measured ambient RH and dry size distribution data and are used to calculate fine-mode ambient scattering and absorption coefficients ($C_{\text{scat,amb}}$ and $C_{\text{abs,amb}}$, respectively) using



the following equation:

$$C_{\text{scat,abs,amb}}(\lambda) = \int\limits_{\text{d}\log\text{D}_{\text{min,amb}}}^{\text{d}\log\text{D}_{\text{max,amb}}} \left[ \frac{\pi \text{D}_{\text{amb}}^2}{4} \times Q_{\text{scat,abs}}(\lambda, \text{CRI}_{\text{amb}}, \text{D}_{\text{amb}}) \times n^{\text{o}}(\text{D}_{\text{amb}}) \right] \text{d}\log\text{D}_{\text{amb}}. \tag{14}$$

Along with the scattering and absorption coefficients, MOPSMAP also provides aerosol optical properties such as extinction coefficient ($C_{\text{ext}} = C_{\text{scat}} + C_{\text{abs}}$) and SSA $\left( \text{SSA} = \frac{C_{\text{scat}}}{C_{\text{ext}}} \right)$ along with microphysical properties such as $N_a$ and effective radius ($\text{r}_{eff}$). Equation 14 is also to calculate the coarse-mode and total ambient optical coefficients.

For the external consistency analysis (Sect. 2.5), the coarse-mode contribution to the in-situ-derived total extinction, $N_a$, and
345 SSA is determined by using the CDP size distribution and assuming that those particles have the microphysical optical properties of hydrated sea salt. The assumptions made for this reason are as follows: 1) the refractive index of these particles is assumed to be that of water ($\text{CRI}_{\text{H}_2\text{O}} = 1.33 + 0i$), 2) the particles are fully hydrated and there is no hygroscopic adjustment needed, and 3) the particles are spherical. Note that calculation of total ambient $N_a$ was limited to optically-active particles where ambient D is $\geq$90 nm (Schlosser et al., 2022).

## 2.4 Synthetic Consistency Analysis Methodology

Synthetic data generation is accomplished using a Monte Carlo-style approach to synthetically create aerosol size distribution and composition data and apply theoretical measurement noise. This section details how to generate synthetic data by 1) creating ground truth aerosol data, which is followed by creating the synthetic in-situ measurements with appropriate measurement
noise and 2) using these synthetic measurements in ISARA for retrievals of IRI and $\kappa$. Then, the resulting optical and microphysical retrievals obtained from ISARA are detailed in Sect. 3.1.1.

Size distributions are generated by assuming the aerosol particles are spherical, since ISARA calculates resulting optical and microphysical properties using Mie theory as detailed in Sect 2.3. The synthetic ground truth size distributions are calculated
for the particle diameter range of 2.97–3487.5 nm for each of the SMPS and LAS size bins to correspond with ACTIVATE size distribution data. To limit the number of unrealistic size distributions used for the retrievals, the shape of the size distribution is fixed and only the total number concentration is allowed to vary. Specifically, the shape of the size distribution is taken to be a 5-bin smoothed ACTIVATE-mean size resolved number concentration from each of the SMPS and LAS bins (Fig. 5) and a randomly chosen scale factor is used to adjust the magnitude of the ACTIVATE-mean size resolved number concentration. The
scale factor range of 0.5–10.0 is used for this analysis. This range is used to capture a large spread in particle concentrations within the ranges observed in the ACTIVATE data.

The hygroscopicity parameter ($\kappa$) is then randomly chosen from the range of 0.00–1.40. The RH for this analysis is held at 80% and hence the corresponding *gf* range is 1.00–1.77. The dry CRI is synthesized by assuming the RRI to be 1.53 while



choosing IRI randomly from a range of 0.001–0.080. Finally, Eq. 3 is used to calculate $C_{\text{scat,dry}}$ at wavelengths of 450, 550, and 700 nm, $C_{\text{abs,dry}}$ wavelengths of 470, 532, and 660 nm, and $C_{\text{scat,wet}}$ at the 550 nm wavelength.

After synthesizing these ground truth aerosol data (i.e., the synthetic size distribution, CRI, and $\kappa$), the in-situ measurements are created using the same randomly generated aerosol properties that are used for the ground truth data synthesis, the corresponding size distribution diameter bins and ranges sampled by the instruments, and the appropriate measurement noise. Appropriate measurement noise (using Table 2 as described below) is added to the synthetic data prior to performing the data processing and retrieval. The random (precision) and systematic (accuracy) measurement uncertainties from Table 2 are applied independently assuming they follow Gaussian probability distributions:

$$\text{measurement} = Y \times \left[1 + \text{rand}_{\text{n}}\left(\text{accuracy}\right)\right] + \text{rand}_{\text{n}}\left(\frac{\text{precision}}{\sqrt{\text{n}_{\text{p}}}}\right), \tag{15}$$

where $\text{n}_{\text{p}}$ is the SMPS resolution of 45 seconds and the $Y$ is the synthesized value of the size resolved number concentration, dry spectral absorption and scattering coefficients, and humidified scattering coefficients. The $\text{rand}_{\text{n}}$ operator generates a random number from a Gaussian probability distribution, which is centered around the expected value of 0 and has a standard deviation that is given by the term in parenthesis. The accuracy from Table 2 noise is applied proportionally to every channel of each instrument to reflect the covariance of the channels. In other words, only one random sample is chosen per instrument. The precision uncertainty adjusted by dividing the uncertainties from Table 2 by the square root of the number of samples made in 45 seconds (i.e., $\text{n}_{\text{p}} = 45$) to represent the increase in measurement precision due to averaging. This adjustment in precision due to averaging is not applied to the synthetic SMPS data as its native resolution is 45 seconds (i.e., $\text{n}_{\text{p}} = 1$).

Equation 15 allows for measurement noise to be added to each synthetically-generated aerosol measurement. After adding the measurement noise to each simulated measurement, a total of 10000 synthetic aerosol distributions are inputted into MOPSMAP to generate IRI, $\kappa$, and consequently $C_{\text{scat}}$, $C_{\text{abs}}$, $C_{\text{ext}}$, SSA, and $f$(RH) (results shown in Sect. 3.1.1). However, note that 26.49% synthetically-generated measurements did not fall within appropriate delta thresholds required for the successful retrieval of IRI and $\kappa$ (see Figs. 2 and 3, respectively). The success rate of 73.51% can be improved by reducing measurement noise.

## 2.5 External Consistency Analysis Process

The platform collocation process for this work is explained in complete detail in Schlosser et al. (2024), but is summarized in this section. Additionally, this section provides a summary of the methods used to column average the in-situ data for comparison with the RSP data, which is described with more detail in Schlosser et al. (2022).

The first step of the collocation process is to match the nearest HSRL-2 time stamp to each RSP scan. In order to search for cases with a substantial presence of fine-mode aerosol particles and low expected error, HSRL-2 and RSP aerosol optical depth



(AOD) at 532 nm (Table 1) are used. Specifically, the HSRL-2 data are removed where the column AOD is <0.08. The RSP data are removed where fine-mode AOD is <0.1 and the normalized cost function of the RSP retrieval to be <0.15. To limit the

presence of coarse-mode aerosol particles in this analysis, the difference in HSRL-2-derived and RSP-derived total AOD must be <50% of HSRL-2-derived AOD or <0.05, whichever is greater. Additionally, coarse-mode AOD is limited to <0.1. Finally, to limit the scope of this analysis to spherical particles, LDR is used to filter out non-spherical from the data set (Burton et al., 2013). A LDR threshold of >13% was used to filter out non-spherical particles from the analysis. As discussed in Sect. 2.1 the ACTIVATE study region is characterized as a marine environment impacted by anthropogenic continental outflow, which is

why the maximum LDR threshold of 13% was chosen.

To collocate the RSP and HSRL-2 data to the in-situ data, the collocation data files produced in Schlosser et al. (2024) are used to filter for times where the two aircraft are within 6 minutes and 15 km. For these comparisons, the ISARA-derived products are acquired where the Falcon aircraft was making a vertical profile through the atmosphere as identified in Sorooshian et al.

(2023). Once the desired data are identified, the ISARA products data are weighted by extinction and averaged to the 225 m HSRL-2 bins or through the entire column for RSP comparisons. To provide aerosol source information for the discussion in Sect. 3.2, this study uses 72 hr back-trajectories from NOAA's Hybrid Single Particle Lagrangian Integrated Trajectory model (Stein et al., 2015). The input meteorological data used was from the North American Mesoscale Forecast System (NAM) and had a 12 km horizontal resolution (Rolph et al., 2017).

### 2.6   Statistical Consistency Analysis Procedures

This section aims to define the metrics used to quantify how well ISARA-derived in-situ data close (i.e., agree) with internal, synthetic, and external consistency data sets. The correlation coefficient ($r$) is used to quantify the strength of correlation between two data sets and is defined as follows for each point (i.e., j):

$$r = \frac{\sum_{j=1}^{n_p}[(X(j) - \bar{X}) \times (Y(j) - \bar{Y})]}{\sum_{j=1}^{n_p}[X(j) - \bar{X}]^2 \times \sum_{j=1}^{n_p}[Y(j) - \bar{Y}]^2},\tag{16}$$

where X and Y are the set of in-situ and remote sensing aerosol measurements, respectively, $n_p$ is the total number of points for each data set, and $\bar{X}$ and $\bar{Y}$ are the mean of sets X and Y, respectively. Additionally, the *p*-value, which is the probability that the two parameters are not correlated (i.e., probability that the null-hypothesis is true), is used to quantify if an *r* value is statistically significant or not. To quantify the difference between two measures of the same parameter, both bias and relative

bias (RB) are used and defined as follows:

$$\text{bias}(j) = Y(j) - X(j)\tag{17}$$

and

$$\text{RB}(j) = \frac{\text{bias}(j)}{Y(j) + X(j)} \times 2 \times 100\%.\tag{18}$$



To quantify the systematic error of a set of comparable measures, the mean relative bias (MRB) and range-normalized root-
mean square deviation (NRMSD) are used. Each of these metrics is defined as follows:

$$\text{MRB} = \frac{\sum_{j=1}^{n_p} \text{RB(j)}}{n_p} \tag{19}$$

and

$$\text{NRMSD} = \frac{100\%}{\max(X) - \min(X)} \times \sqrt{\frac{\sum_{j=1}^{n_p} [Y(j) - X(j)]^2}{n_p}}. \tag{20}$$

Finally, the MRB $\pm$ standard deviation in relative bias (SRB) is used to quantify the uncertainty of a given aerosol property
retrieval made by ISARA. The SRB is defined as follows:

$$\text{SRB} = \sqrt{\frac{\sum_{j=1}^{n_p} (\text{RB(j)} - \text{MRB})}{n_p}}. \tag{21}$$

In this paper, statistical consistency is determined to be successful for a given ambient aerosol property if MRB $< 30\%$, NRMSD
$< 25\%$, $r > 0.8$, and $p$-value $< 0.05$. Additionally, RSP is considered consistent for a given ambient aerosol property if the mean
values are within the RSP uncertainty values for that property listed in Table 2. If the results are within 5% of these values,
they are considered "partially successful". Otherwise, they are considered unsuccessful.

# 3 Results and Discussion

## 3.1 Statistical Analysis

### 3.1.1 Synthetic consistency

Now, consistency of ISARA-derived in-situ IRI and $\kappa$ with synthetically-generated ones is presented. Before delving into these
comparisons, synthetically-generated size distribution data are shown to provide context on how synthetic IRI and $\kappa$ differ
from ISARA-derived ones (Fig. 5).





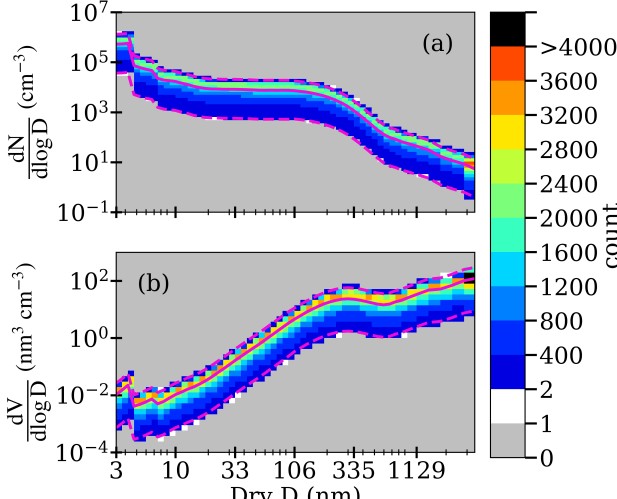

**Figure 5.** Logarithmic size-resolved aerosol particle (a) number concentration $\left(n^{\circ} = \frac{\mathrm{dN}}{\mathrm{d\log D}}\right)$ and (b) volume concentration $\left(v^{\circ} = \frac{\mathrm{dV}}{\mathrm{d\log D}}\right)$ versus dry particle diameter ($\mathrm{D_{dry}}$) from the 10,000 synthetic data points. The solid line represents the arithmetic mean of each bin, the bottom and top dashed lines represent the minimums and maximums of each bin, respectively, and the color bar indicates density of points in a given area of the plot.

The synthetic size distribution data are contained within the range of number and volume concentrations observed during ACTIVATE (Fig. 1), but the synthetic data sees overall less variance. Future work could explore the impacts of adjusting the
synthetic size distribution creation process to analyze the impact of low total concentration conditions.

As mentioned in Sect. 2.4, these theoretical size distributions are used to generate synthetic IRI and $\kappa$ values. These synthetic values are now compared to corresponding ISARA-derived data (Fig. 6).





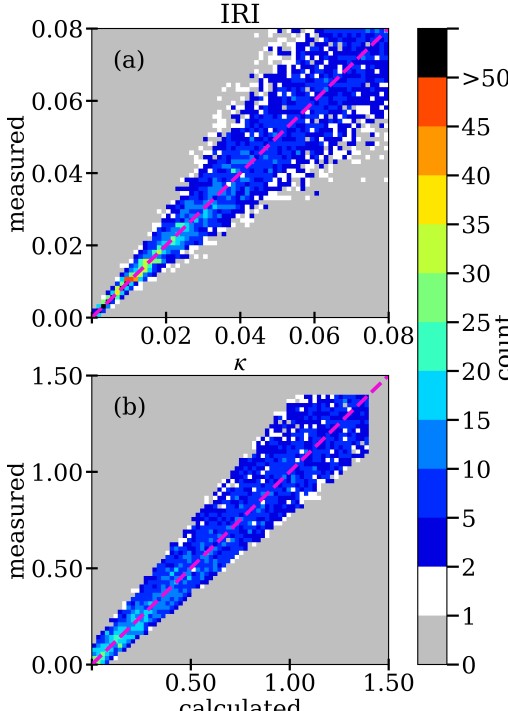

**Figure 6.** Heat map scatterplot of ISARA-retrieved versus synthetic (a) IRI (count = 7351, $r$ = 0.99, MRB $\pm$ SRB = -0.9 $\pm$ 20%, and NRMSD = 9%) and (b) $\kappa$ (count = 6823, $r$ = 0.98, MRB $\pm$ SRB = 3 $\pm$ 28%, and NRMSD = 8%). The dashed line represents the one-to-one line and the color bar indicates density of points in a given area of the plot.

Of the 10000 synthetically generated data points, 73.51% have successful retrieval of both IRI and $\kappa$. There are 5.28% lost
between the IRI retrieval and the $\kappa$ retrieval that corresponds to a success rate = 92.82%, which is close to the 100% success rate observed in the ACTIVATE retrieval of $\kappa$. It is observed that both IRI and $\kappa$ comparisons show strong correlation coefficients of 0.99 and 0.98, respectively. Additionally, the biases (MRB $\pm$ SRB) are centered near zero which are -0.9 $\pm$ 20% and 3 $\pm$ 28% for IRI and $\kappa$, respectfully. Based on these observations, the NRMSD for IRI and $\kappa$ are 9% and 8%, respectively.





In the algorithm's current state, it is expected that the ISARA-derived IRI and $\kappa$ each agree within 30% under ideal assumptions
of spherical particle shapes, a spectrally flat CRI, and a well-constrained RRI. Given the strong correlations and low biases, it
is found that error as a result of forward modeling for spherical particles with a well constrained dry RRI ($1.52 \leq \text{RRI} \leq 1.54$)
should not have a significant impact on the retrieval of a single effective IRI and $\kappa$ from the size distribution, scattering coeffi-
cient, and absorption coefficient data measured during missions such as ACTIVATE. Measurement noise does appear to have
a significant impact on the successful retrieval rate, as when it is removed the rate of successful retrievals increases to 100%.
Synthetic consistency analysis can be extended further to include non-spherical particles, particles without a constrained RRI,
and increasing the number of successful retrievals under higher noise and lower signal conditions (e.g., lower aerosol particle
concentrations, weakly scattering or weakly absorbing aerosol particles).

### 3.1.2    Internal Consistency

As mentioned in the Introduction and Sect. 2, ISARA-calculated in-situ data are first closed with corresponding measurements
from ACTIVATE's in-situ instruments to verify the robustness of the algorithm's retrieval method. First, ISARA retrievals of
dry scattering and absorption coefficients are verified against corresponding measurements from the nephelometer and PSAP
described in Sect. 2.2 (Fig. 7). There are a total of 3444 in-situ data points that met the cloud-free threshold, have valid
data (i.e., signal $> 1\,\text{Mm}^{-1}$) in all 6 of the dry scattering and absorption channels, and have at least 3 channels with non-zero
measurements of $n^{\text{o}}$ from both the SMPS and LAS instruments. Of these 3444 data points, 1319 also have $f(\text{RH})$ measurements.
There are 580 points that had the successful retrieval of CRI but not $\kappa$ and 1319 had the successful retrieval of both CRI and $\kappa$.
The observed successful retrieval rate for CRI is 55%, which is lower than the success rate of 73.51% observed for the synthetic
consistency analysis (Sect. 3.1.1). Compared to the synthetic data set, the relatively lower retrieval success rate observed in
the measured data set could be an indication that some of the measured data might be influenced by particles that violate the
sphericity or the spectrally flat CRI assumptions. The lower retrieval success rate could be as a result of higher measurement
noise than prescribed in the generation of the simulated data.





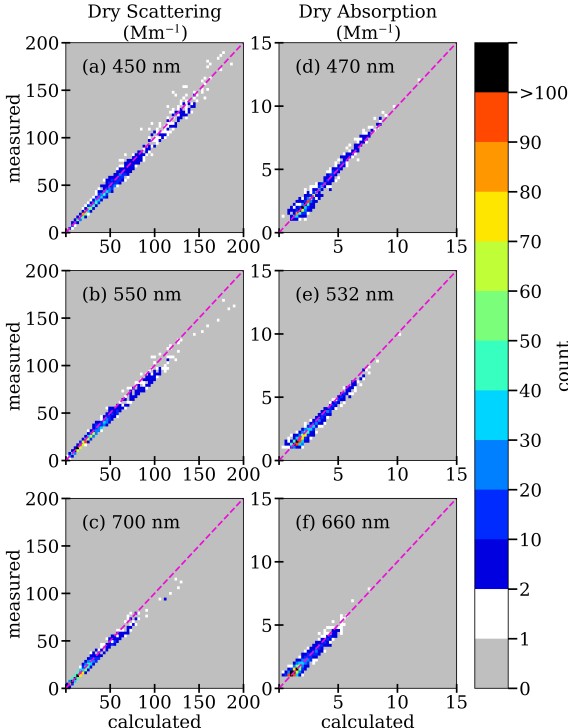

**Figure 7.** Heat map scatterplots of measured versus ISARA-derived in-situ values of dry scattering coefficient at (a) 450 nm ($r = 0.99$, MRB $\pm$ SRB = -4 $\pm$ 7%, NRMSD = 2%), (b) 550 nm ($r = 0.99$, MRB $\pm$ SRB = -8 $\pm$ 7%, NRMSD = 2%, and (c) 700 nm ($r = 0.99$, MRB $\pm$ SRB = -2 $\pm$ 11%, NRMSD = 2%). Also plotted are measured versus ISARA-derived in-situ values of dry absorption coefficient at (d) 470 nm ($r = 0.99$, MRB $\pm$ SRB = -1 $\pm$ 11%, NRMSD = 2%), (e) 532 nm ($r = 0.98$, MRB $\pm$ SRB = -5 $\pm$ 14%, NRMSD = 3%), and (f) 660 nm ($r = 0.96$, MRB $\pm$ SRB = 2 $\pm$ 17%, NRMSD = 4%). There are 1899 ISARA retrievals that resulted from the ACTIVATE 2020–2022 data set. The dashed line represents the one-to-one line and the color bar indicates density of points in a given area of the plot.

It is observed that for all three wavelengths, the two sets of dry scattering coefficient measurements correlate nearly perfectly (i.e., $r = 0.99$) and agree within 2% of each other in terms of NRMSD. Across all three channels, the nephelometer measures slightly lower values (within -8%) of the dry scattering coefficient than ISARA retrieves. The standard deviation in relative bias (SRB) ranges from 7 to 11 which suggests that there is an apparent non-zero bias in the ISARA forward model. Sources of this bias could be the assumption of a spectrally flat CRI and the assumption of a well constrained RRI of 1.53 $\pm$ 0.01. That being noted, the mean biases that resulted from the comparisons in scattering for all three channels range from -0.75 to -2.6 Mm$^{-1}$, which suggests that the  is being impacted by comparisons with relatively low signal (i.e., signal < 10 Mm$^{-1}$).

Dry absorption is also internally consistent as seen by strong $r$ (0.99, 0.98, and 0.96) and NRMSD (2%, 3%, and 4%) values. Relative to the scattering coefficient comparisons, there is a similar negative bias observed in the 470 and 532 nm channels of





absorption (MRB ± SRB = -1 ± 11% and -5 ± 14%, respectively). In contrast to the other absorption and scattering measurements, there is a slightly positive bias (MRB = 2%) seen in the 700 nm absorption coefficient comparisons. The positive bias observed in the 700 nm absorption channel comparisons do have the largest spread in bias (SRB = 17%), relative to the other

absorption and scattering channels.

As with the scattering coefficient comparisons, the mean biases between the ISARA- and PSAP-derived dry absorption are small relative to the measurement uncertainty. The mean biases range from -0.14 to -0.01 Mm$^{-1}$, which would again indicate that the corresponding MRB values observed in the absorption comparisons are being influenced by low signal. Other reasons

for the MRB observed in the absorption comparisons could be the errors associated with size distribution measurements such as differences between optical and mobility particle diameters, loss of small and larger particles through the sampling system, and instrument counting efficiencies. Overall, internal consistency of the dry absorption coefficient is deemed successful, but it is important to keep these biases in the absorption and scattering coefficients in mind when calculating secondary optical properties that rely on it, such as $C_{\mathrm{ext}}$ and SSA.


Now, ISARA retrievals of the wet scattering coefficient ($C_{\mathrm{scat,wet}}$) and $f$(RH) are evaluated to test how well the $\kappa$ retrieval performs before calculating final ambient aerosol properties (Fig. 8).



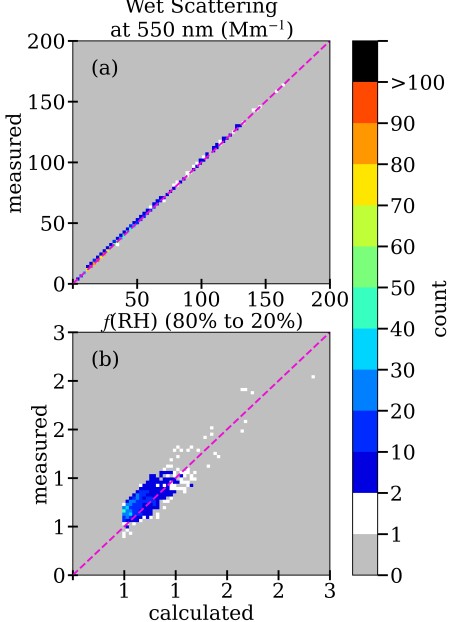

**Figure 8.** Heat map scatterplot of measured versus ISARA-derived in-situ values for (a) wet scattering coefficient ($C_{\mathrm{scat,wet}}$) at 550 nm ($r$ = 1.00, MRB $\pm$ SRB = 0.8 $\pm$ 0.2%, NRMSD = 0.2%) and (b) hygroscopic growth function ($f$(RH)) at 550 nm ($r$ = 0.83, MRB $\pm$ SRB = 7 $\pm$ 8%, NRMSD = 7%) from the 1319 successful ISARA retrievals that resulted from the 2020–2022 ACTIVATE data. The dashed line represents the one-to-one line and the color bar indicates density of points in a given area of the plot.

Strong correlation coefficients ($r$ = 1.00 and 0.83), strong agreement (NRMSD = 0.2% and 7%), and low bias (MRB $\pm$ SRB = 0.8 $\pm$ 0.2% and 7 $\pm$ 8%) are seen between the two data sets. The $f$(RH) variability (SRB = 8%) is linked directly to the vari-
ability in the retrieval of the dry scattering coefficient (SRB = 7%). In addition, there is also a limitation in assuming the $\kappa$ is uniform for the fine-mode particles since aerosol particles typically do not have homogeneous compositions. Future iterations of ISARA can involve implementing multiple $\kappa$ parameters and having a non-soluble mode, which has been shown to be useful in other consistency studies (van Diedenhoven et al., 2022). Based on the conditions detailed in Sect. 2.6, $C_{\mathrm{scat,wet}}$ and $f$(RH) at 550 nm are deemed internally consistent with the ISARA-derived values.

### 3.1.3 External Consistency

After performing the procedures detailed in Sect. 2, ISARA-derived in-situ measurements of aerosol properties can be compared to coincident HSRL-2 and RSP retrievals for 2020–2022 ACTIVATE data. There are a total of 40 vertical profiles that have both an absence of clouds and meet the required collocation thresholds. Of the 40 vertical profiles, 15 of them have more than 3 points of comparison between the in-situ and HSRL-2 data. The ancillary and consistency statistics for these 15 cases are listed in Tables 3 and 4, respectively. The ancillary information includes the case number, the profile start and stop times,



the associated RSP sample time, the minimum and maximum altitudes sampled by the Falcon aircraft, the number of smoke counts above 2.5 km identified by the HSRL-2, the RSP-derived total, fine-mode, and coarse-mode AOD, the HSRL-2-derived AOD, and the horizontal separation between the Falcon and King Air. All dates and times are provided in coordinated universal

time (UTC) and in the format "year-month-day" and "hour:minute:second", respectively. The smoke counts above 2.5 km are taken to be the sum of all the altitude bins above 2.5 km that are flagged as smoke from from the HSRL-2 typing product (Burton et al., 2012). Consistency statistics for each profile includes $r$, MRB, SRB, NRMSD, $p$-value, and number of HSRL-2 altitude bins with both in-situ and remote sensing data (i.e., count).

**Table 3.** Ancillary data for each of the 15 case studies. Ancillary information includes the case number, the profile start and stop times, the associated RSP sample time, the minimum and maximum altitudes sampled by the Falcon (i.e., in-situ) aircraft, the number of smoke counts above 2.5 km identified by the HSRL-2, the RSP-derived total, fine-mode, and coarse-mode AOD, the HSRL-2-derived AOD, in-situ-derived AOD, and the horizontal separation between the Falcon and King Air. All dates and times are provided in coordinated universal time (UTC) and in the format "year-month-day" and "hour:minute:second", respectively.

| Case # | Date | Profile start | Profile end | RSP time | Altitude (km) Min | Max | Smoke counts | RSP AOD Total | Fine | Coarse | HSRL-2 AOD | in-situ AOD | Platform Separation (m) |
|---|---|---|---|---|---|---|---|---|---|---|---|---|---|
| 1 | 2020-09-03 | 15:08:41 | 15:13:37 | 15:10:10 | 360 | 2460 | 0 | 0.23 | 0.23 | 0.00 | 0.23 | 0.11 | 12 |
| 2 | 2021-03-04 | 17:59:53 | 18:05:45 | 18:03:16 | 458 | 1578 | 0 | 0.16 | 0.16 | 0.00 | 0.15 | 0.06 | 61 |
| 3 | 2021-03-09 | 16:29:26 | 16:32:52 | 16:32:20 | 387 | 1380 | 0 | 0.12 | 0.12 | 0.00 | 0.08 | 0.02 | 121 |
| 4 | 2021-03-12 | 17:45:09 | 17:50:21 | 17:45:38 | 413 | 1449 | 0 | 0.16 | 0.11 | 0.05 | 0.13 | 0.06 | 2264 |
| 5 | 2021-03-12 | 18:42:11 | 18:46:32 | 18:45:00 | 373 | 1058 | 0 | 0.12 | 0.11 | 0.01 | 0.09 | 0.01 | 159 |
| 6 | 2021-05-14 | 19:00:12 | 19:05:42 | 19:05:20 | 407 | 1284 | 0 | 0.14 | 0.12 | 0.01 | 0.09 | 0.01 | 124 |
| 7 | 2021-05-26 | 15:07:46 | 15:13:12 | 15:09:03 | 701 | 1816 | 137 | 0.30 | 0.26 | 0.04 | 0.27 | 0.05 | 1 |
| 8 | 2021-06-15 | 16:11:58 | 16:17:01 | 16:13:50 | 504 | 1268 | 4 | 0.19 | 0.18 | 0.01 | 0.18 | 0.02 | 1840 |
| 9 | 2021-06-17 | 17:08:49 | 17:13:07 | 17:08:48 | 873 | 1833 | 0 | 0.10 | 0.10 | 0.00 | 0.08 | 0.01 | 11375 |
| 10 | 2022-03-03 | 14:56:17 | 15:22:48 | 15:19:24 | 238 | 4499 | 0 | 0.12 | 0.10 | 0.02 | 0.08 | 0.08 | 1829 |
| 11 | 2022-03-03 | 19:27:13 | 19:31:18 | 19:28:29 | 403 | 1178 | 31 | 0.16 | 0.10 | 0.05 | 0.11 | 0.03 | 106 |
| 12 | 2022-03-22 | 19:13:50 | 19:21:14 | 19:18:17 | 304 | 1366 | 0 | 0.20 | 0.12 | 0.09 | 0.16 | 0.03 | 57 |
| 13 | 2022-05-18 | 13:05:23 | 13:10:45 | 13:08:31 | 299 | 1768 | 0 | 0.13 | 0.11 | 0.02 | 0.09 | 0.02 | 15 |
| 14 | 2022-05-20 | 14:29:36 | 14:34:41 | 14:30:22 | 435 | 1466 | 28 | 0.45 | 0.39 | 0.06 | 0.45 | 0.10 | 319 |
| 15 | 2022-05-31 | 12:51:37 | 12:54:55 | 12:54:51 | 388 | 1256 | 0 | 0.19 | 0.19 | 0.00 | 0.17 | 0.05 | 893 |





**Table 4.** Closure statistics between ambient vertically-resolved total $N_a$ and extinction derived from in-situ and the remote sensors for each of the 15 case studies. Closure statistics for each profile includes MRB, NRMSD, $r$, $p$-value, and number of HSRL-2 altitude bins with both in-situ and remote sensing data (count). The $r$ and $p$-value are not reported when the profile has only two altitude bins for comparison and none of the statistics are shown for cases where there is only one point of comparison.

| Case | Ambient $N_a$ | | | | | Ambient extinction at 532 nm | | | | | count |
| # | MRB (%) | SRB (%) | NRMSD (%) | $r$ | $p$-value | MRB (%) | SRB (%) | NRMSD (%) | $r$ | $p$-value | |
|---|---|---|---|---|---|---|---|---|---|---|---|
| 1 | 35 | 21 | 21 | 0.82 | 0.18 | 27 | 17 | 32 | 0.86 | 0.14 | 4 |
| 2 | -100 | 8 | 8 | 0.87 | 0.33 | 9 | 15 | 26 | 0.73 | 0.48 | 3 |
| 3 | -57 | 46 | 46 | -0.75 | 0.46 | 37 | 48 | 49 | 0.16 | 0.90 | 3 |
| 4 | -120 | 18 | 18 | 1.00 | 0.00 | -10 | 32 | 20 | 0.91 | 0.09 | 4 |
| 5 | -79 | 1 | 1 | 0.99 | 0.07 | 65 | 3 | 74 | 0.98 | 0.14 | 3 |
| 6 | 22 | 30 | 25 | -0.33 | 0.79 | 142 | 23 | 76 | -0.26 | 0.83 | 3 |
| 7 | 72 | 26 | 26 | 0.56 | 0.44 | 82 | 17 | 44 | 0.87 | 0.13 | 4 |
| 8 | -9 | 37 | 24 | 0.28 | 0.82 | 41 | 30 | 59 | 0.56 | 0.62 | 3 |
| 9 | -163 | 10 | 10 | -0.56 | 0.62 | 15 | 13 | 33 | 0.94 | 0.21 | 3 |
| 10 | -12 | 34 | 20 | 0.98 | 0.00 | 6 | 32 | 7 | 0.98 | 0.00 | 14 |
| 11 | 25 | 11 | 11 | 0.92 | 0.25 | 50 | 4 | 59 | 0.98 | 0.12 | 3 |
| 12 | 16 | 31 | – | 0.93 | 0.02 | 64 | 29 | 63 | 0.88 | 0.05 | 5 |
| 13 | -86 | 11 | 11 | 0.92 | 0.26 | 55 | 19 | 54 | 0.79 | 0.42 | 3 |
| 14 | 44 | 17 | 17 | 0.23 | 0.77 | 19 | 25 | 45 | 0.16 | 0.84 | 4 |
| 15 | 27 | 41 | 26 | 0.79 | 0.42 | -9 | 32 | 31 | 0.93 | 0.24 | 3 |

Data from the 40 profiles provide 102 points for studying consistency of ambient $C_{\text{ext}}$ and $N_a$. First, results of performing consistency analysis between HSRL-2 and in-situ $C_{\text{ext}}$ are presented (Fig. 9).





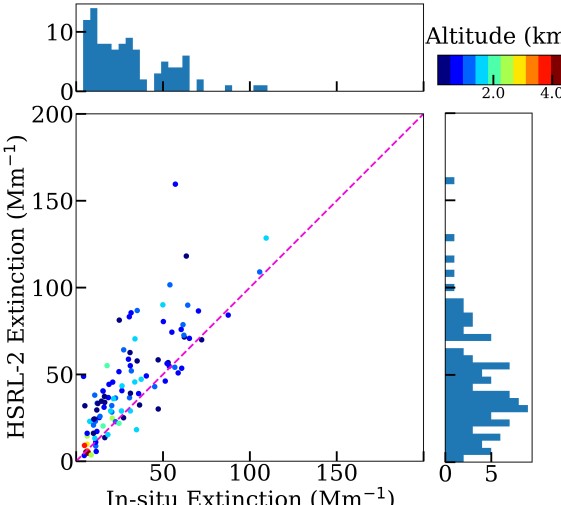

**Figure 9.** Scatterplot of HSRL-2-retrieved versus ISARA-derived in-situ values for extinction coefficient ($C_{\text{ext}}$) (count = 102, $r$ = 0.79, MRB $\pm$ SRB = 37 $\pm$ 45%, NRMSD = 15%) using 2020–2022 ACTIVATE data. The points are colored by the sample altitude in km. Also shown on the perimeter of the plot are histograms to show the sample distribution of each variable, as well as a dashed line that represents the one-to-one line.

The HSRL-2- and ISARA-derived ambient $C_{\text{ext}}$ are moderately correlated with a $r$ of 0.79. The MRB of 37% shows that the in-situ data data is biased low from the HSRL-2, showing that the in-situ instruments retrieve lower values of $C_{\text{ext}}$ than the HSRL-2 throughout the ACTIVATE campaign. This low bias result is also seen in Sawamura et al. (2017), which are MRB $\pm$

SRB = 31 $\pm$ 5% and 53 $\pm$ 11%, for California and Texas, respectively. Note that Sawamura et al. (2017)'s algorithm only considers fine-mode species in its analysis but ISARA accounts for the coarse-mode aerosol by using the CDP data and assuming the particles have the optical and microphysical properties of hydrated sea salt as described in Sect. 2.3. This discrepancy with the remote sensors is likely due to the loss of particles from the diameter cutoff of the inlet and through the in-situ sampling pathways as discussed in the Introduction and undersampling of the coarse aerosol particles by the CDP. Although in-situ

values are lower than the HSRL-2 ones, strong agreement is seen by the NMAD of 15%. Therefore, these results indicate that further work is needed to close HSRL-2 and in-situ $C_{\text{ext}}$, however the method does result in ambient extinction that meet the benchmarks set by previous works. By using ACTIVATE data for this analysis, the extinction product does work as expected even in conditions where coarse-mode sea salt is impacting the aerosol extinction. Future work can investigate better methods for measuring coarse-mode aerosol from in-situ aircraft.


Next, jointly-retrieved HSRL-2+RSP and in-situ aerosol particle number concentration ($N_a$) are compared (Fig. 10 and Table 4).





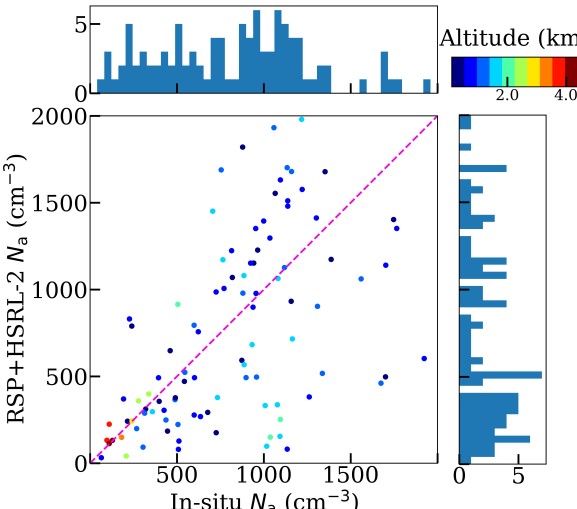

**Figure 10.** Scatterplot of HSRL-2+RSP-derived versus ISARA-derived in-situ values for aerosol particle number concentration ($N_a$) (count = 102, $r$ = 0.50, MRB $\pm$ SRB = -14 $\pm$ 65%, NRMSD = 20%) using 2020–2022 ACTIVATE data. The points are colored by the sample altitude in km. Also shown on the perimeter of the plot are histograms to show the sample distribution of each variable, as well as a dashed line that represents the one-to-one line.

These comparisons result in an *r* of 0.50, an MRB of -14%, and an NRMSD of 20% (Fig. 10). These results have a much lower
bias than shown in the $C_{\text{ext}}$ comparisons, but show that the HSRL-2+RSP product slightly underestimates in-situ $N_a$. Other than this lower bias, agreement in terms of NRMSD is comparable to the $C_{\text{ext}}$ results. While the correlation for ISARA $N_a$ comparisons is weak compared to vertically-resolved $N_{\text{LAS}}$ ones (best is 0.76 in Schlosser et al. (2022)), the *r* values for where the profile had statistically significant correlations for both $N_a$ and extinction (*p*-value < 0.05) is 0.98 and 0.88 for case 10 and case 12 (see Tables 3 and 4), respectively. Case 10 is the only profile with more than 5 points for comparison and the conditions
observed during this case study are such that it is investigated more in Sect. 3.2. Case 12 does not serve as an ideal case as it has only has five points of comparison. Overall, the $N_a$ comparisons are considered to be closed relatively successfully when compared to results of Schlosser et al. (2022)'s evaluation of HSRL-2+RSP-derived $N_a$ using $N_a$ measured by the LAS.

## 3.2 Case Study

A case flight (Research Flight 131 on 3 March 2022) is chosen from Table 3 to examine how well ambient in-situ aerosol optical and microphysical measurements produced by ISARA compare with analogous RSP retrievals, which has not been shown in literature to date. On this day, a "unicorn" module was performed (Sorooshian et al., 2023), where the Falcon aircraft performed a spiral from an altitude of 238 m to an altitude of 4,499 m to fully vertically-sample a rich aerosol layer identified as having urban/pollution and dust species as determined by the HSRL-2 aerosol typing algorithm (Burton et al., 2012). The



Falcon spiral began at 14:56 UTC and ended at 15:22 UTC while the RSP sample time was at 15:19 UTC. The distance of the Falcon-in-situ spiral to the nearest valid RSP-King Air overpass was 1.8 km for this clear-sky aerosol scene over the ocean. No cloud contamination was identified in the HSRL-2, RSP, or camera images.

The three-day Hybrid Single Particle Lagrangian Integrated Trajectory (HYSPLIT) back trajectories are run at the altitudes of
500, 1500, and 3000 m above sea level, at 16:00 UTC on 3 March 2022, and at the location of the "unicorn" spiral (Fig. 11). These back trajectories use the meteorological data from the North American Mesoscale Forecast System (NAM) $12 \times 12\,\mathrm{km}^2$ HYSPLIT meteorological data set. From these back trajectories, it is evident that the air mass that was sampled during this case was outflow from North America. This information adds more evidence that the fine-mode particles being sampled were anthropogenic in origin and are likely organic and sulfate-dominated mixtures.

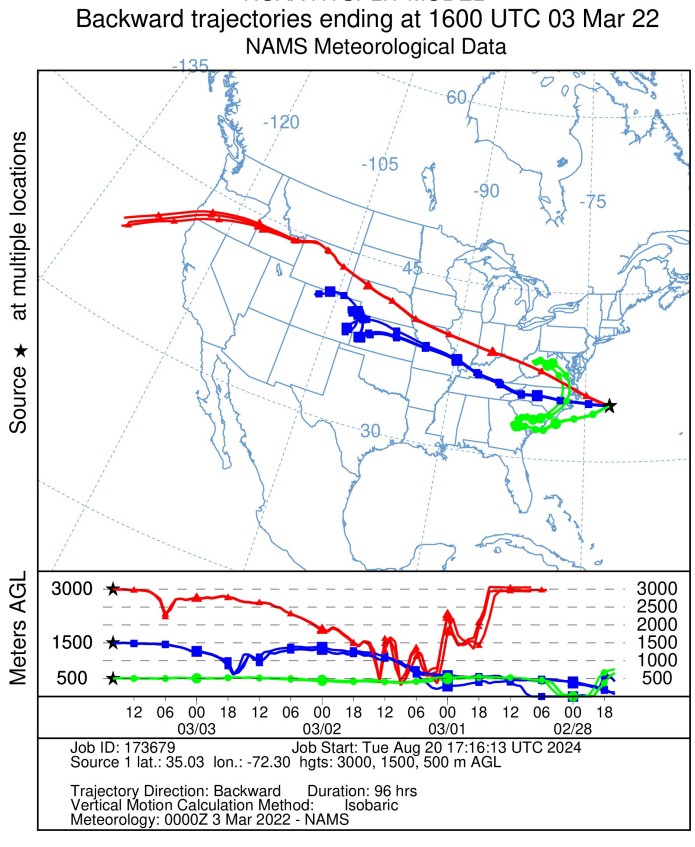

**Figure 11.** Three-day Hybrid Single Particle Lagrangian Integrated Trajectory (HYSPLIT) back trajectories end at the altitudes of 500, 1500, and 3000 m above sea level, at 16:00 UTC on 3 March 2022, and at the location of the "unicorn" spiral (Case 10) from Research Flight 131.

With these flight conditions in mind, ISARA is used to generate vertically-resolved ambient aerosol properties along the spiral (Fig. 12). Panel (a) shows the result from the vertically-resolved HSRL-2+RSP-derived $N_a$ product, while panel (b) shows




the ISARA-derived ambient aerosol extinction coefficient versus the extinction coefficient derived from the HSRL-2. The remaining panels, (c)–(h), show the vertical profiles in-situ-derived RH, $\kappa$, RRI, IRI, fine-mode SSA, total SSA, and fine-mode $r_{eff}$.

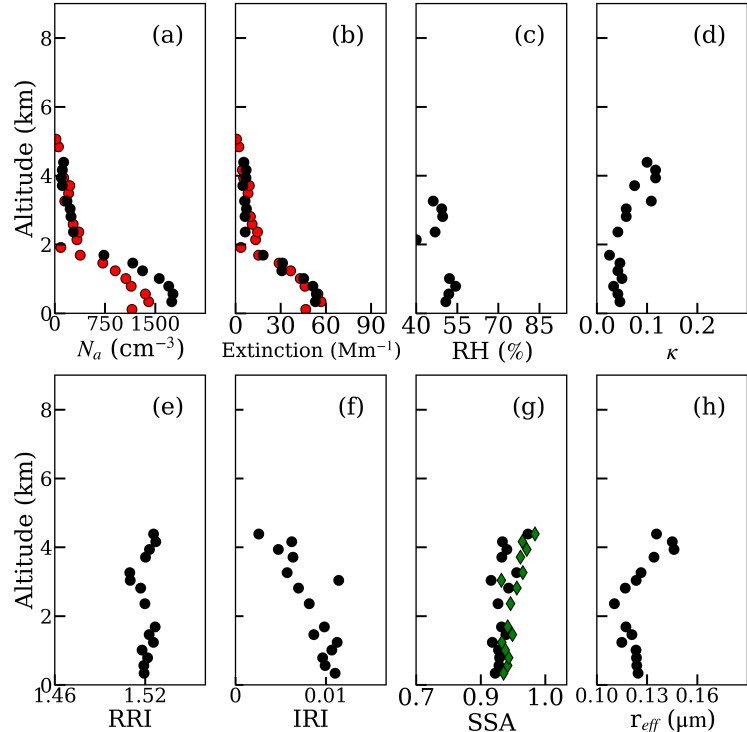

**Figure 12.** Vertical profiles of (a) RSP+HSRL-2-derived (red points) and ISARA-derived (black points) $N_a$, (b) HSRL-2-derived (black points) and ISARA-derived (red points) ambient extinction coefficient,(c) in-situ-derived ambient RH, along with ISARA-derived (d) $\kappa$, ambient (e) RRI, (f) IRI, (g) fine-mode (green diamonds) and total (black diamonds) SSA, and (h) fine-mode $r_{eff}$ from case study 10 that occurred during Research Flight 131 on 3 March 2022. The HSRL-2- and RSP-derived AOD are 0.08 and 0.12, respectively.

It appears there are moderately absorbing aerosol particles closer towards the surface, which is supported by ambient IRI values near 0.015 (Fig. 12(f)) and lower SSA values near 0.90 (Fig. 12(g)). There is also an increasing fine-mode $r_{eff}$ seen above 2 km (Fig. 12(h)). It is also seen that ISARA-derived RH range is 40–55% (Fig. 12(c)), which is not unexpected for a marine environment far off-shore in the winter (Sorooshian et al., 2019). A potential reason for detecting low-RH aerosol is the presence of smoke from fires (Fig. 13).



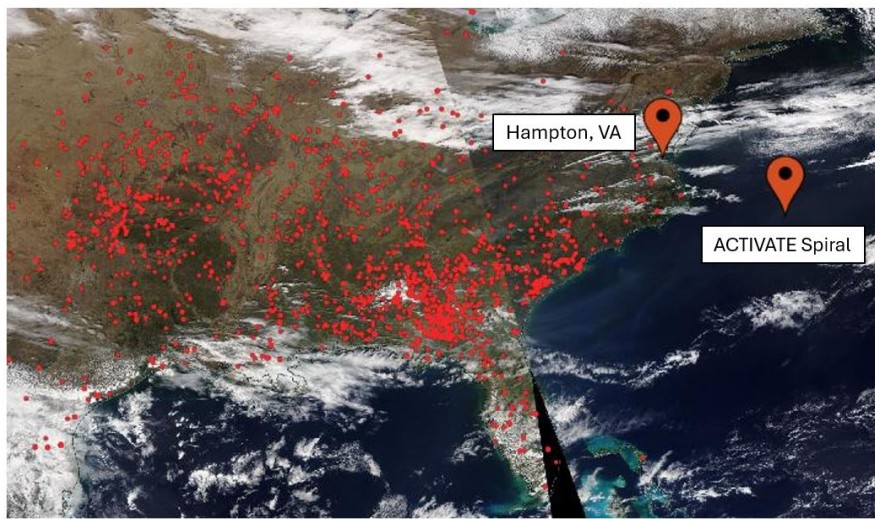

**Figure 13.** NASA Worldview satellite image of fire presence (red points) during Research Flight 131 (3 March 2022) case flight with locations of LaRC and the Falcon flight spiral labeled. This image is used to identify sources of smoke on this day and comes from the VIIRS Fire and Thermal Anomalies product available from the NOAA-20 satellite (Schroeder et al., 2014).

Although smoke aerosol seems present (Fig. 13), aerosol loading is low (AOD of 0.08 (HSRL-2) and 0.12 (RSP)) and the HSRL-2 is not detecting smoke counts in the vertical profiles shown (Table 3). It may be more likely that the aerosol species being measured are becoming drier due to colder temperatures in the winter season and being present at altitudes further above the surface. Future work will be done to investigate the impacts on the consistency analysis that would result from increasing the fidelity of the RH measurements.


With these conditions in mind, the vertical profile of ISARA results are then averaged using the extinction coefficient to provide a column-averaged result (Table 5). For comparison, the data from case 12 are also shown on Table 5. Similar to case 10, the ISARA- and RSP-derived total SSA is in poor agreement, the fine-mode $r_{eff}$ and SSA are in moderate agreement, and the remaining aerosol properties are in good agreement. Note that the vertical extent of the Falcon's profile is 1064 m. In addition
to the limited coarse-mode sampling, the limited vertical extent of the Falcon's profile might be impacting the consistency in this case.



**Table 5.** Ambient optical and microphysical aerosol particle properties for case studies 10 and 13 derived from the ISARA and the RSP. Aerosol particle properties compared include the column-averaged fine-mode SSA, total SSA, RRI, IRI, fine-mode $r_{eff}$, and $N_a$. The RSP retrieved fine- and coarse-mode AOD at 532 nm for each of these cases are shown on Table 3.

| Case | Fine-mode SSA | | Total SSA | | Fine-mode RRI | | Fine-mode IRI | | Fine-mode $r_{eff}$ (µm) | | $N_a$ (cm$^{-3}$) | |
| # | in-situ | RSP | in-situ | RSP | in-situ | RSP | in-situ | RSP | in-situ | RSP | in-situ | RSP |
| 10 | 0.91 | 0.91 | 0.89 | 0.95 | 1.52 | 1.49 | 0.015 | 0.017 | 0.12 | 0.15 | 1369 | 1250 |
| 12 | 0.95 | 0.99 | 0.92 | 1.00 | 1.49 | 1.48 | 0.007 | 0.001 | 0.12 | 0.15 | 1009 | 1158 |

Overall, in-situ and RSP derived products are in good agreement under ideal conditions (Table 5). The only parameter that deviates significantly from the RSP retrieval error (see Stamnes et al., 2018) is total SSA, which deviates by 0.06 from ISARA. This disagreement in total SSA is likely due to coarse-mode sea salt aerosol particles being under-sampled by the in-situ instruments due to the low counting efficiency of the CDP of particles in this size range. This could also explain why the aerosol extinction coefficient retrieved by in-situ methods is systematically low as compared to HSRL-2.

## 4 Conclusions

This study introduces the In Situ Aerosol Retrieval Algorithm (ISARA), a Python model that calculates ambient aerosol particle properties through the retrieval of complex refractive index (CRI) and hygroscopicity parameter ($\kappa$). What is beneficial about ISARA is that it can account for contribution from coarse-mode (ambient particle diameter > 1.0 µm) when making in-situ retrievals, enabling one-to-one comparison with ambient remote sensing retrievals and ultimately promoting rigorous verification (i.e., consistency) of field campaign aerosol data at large. To test the ability of ISARA to perform systematic consistency, the algorithm is used on data from the Aerosol Cloud meTeorology Interactions oVer the western ATlantic Experiment (ACTIVATE) mission, which is chosen due to the diversity of aerosol and meteorological conditions in the study region and the campaign's large volume of statistics. Systematic consistency in this study is a three-fold effort: 1) internal consistency, which compares ISARA-retrieved ambient in-situ measurements with corresponding data from ACTIVATE's in-situ instruments, 2) synthetic consistency, where ambient in-situ data calculated from theoretical size distribution and composition data are compared to ISARA-derived in-situ values, and 3) external consistency, where ISARA-derived in-situ measurements are used to evaluate corresponding remote sensing retrievals from ACTIVATE's Second Generation High Spectral Resolution Lidar (HSRL-2) and Research Scanning Polarimeter (RSP) instruments.

Overall, this study demonstrates the successful retrieval of ambient aerosol properties from in-situ data in all three consistency analyses. Internal consistency analysis show near-perfect correlations ($r \geq 0.96$), strong agreement (NRMSD = 1%), and generally low bias (absolute MRB < 10%) between ISARA-calculated and measured in-situ data for the dry scattering coefficient,



dry absorption coefficient, wet scattering coefficient, and *f*(RH). Synthetic consistency analysis shows that errors in the forward model itself do not have a substantial influence on retrieved ambient aerosol properties since retrieved values of CRI and $\kappa$ for spherical particles with expected measurement noise are found to have a forward modeling error (i.e., NRMSD) of 9% and 8%, respectively. However, it is important to note that the assumptions of perfectly spherical particles, of a single $\kappa$ for all

fine-mode particles, and of a spectrally flat CRI can still introduce limitations in the current version of ISARA's retrievals since aerosol particles from species such as sea salt, smoke, and dust can be non-spherical, can have inhomogeneous composition, and can have a wavelength dependant CRI.

External consistency between vertically-resolved HSRL-2-derived extinction and with column-averaged RSP-derived fine-

mode $r_{eff}$, CRI, and SSA under ideal conditions is also deemed successful. The RSP-derived ambient total $N_a$ and fine-mode IRI, RRI, SSA, $r_{eff}$, and $N_a$ were all shown to be within expected error as compared to collocated in-situ data. The HSRL-2 extinction coefficient ($C_{ext}$) is well correlated ($r = 0.79$) with the ISARA-derived extinction coefficient, but the in-situ-derived extinction coefficient appears to be low (MRB $\pm$ SRB = 37 $\pm$ 45%) compared to the corresponding HSRL-2 measurement. This bias is likely due to under-sampling of coarse-mode aerosol species within the in-situ measurements. The total SSA is

also underestimated by ISARA, as seen by ISARA-derived total SSA being low by 0.06 relative to the RSP-derived total SSA.

While there are limitations and implicit errors in the ISARA retrievals as discussed above, these products are shown to be useful for validating remote sensing data overall. The CRI and $\kappa$ products are also useful for the modeling community. In further iterations of ISARA, improvements such as extending the analysis to include non-spherical particles, improving the coarse-

mode representation, and adding non-soluble portions of the aerosol distribution can be implemented. Overall, it is hoped that ISARA can be used beyond the ACTIVATE field campaign to advance towards the complete closure of field campaign aerosol data sets in general.

*Code availability.* The ISARA codebase can be found at https://github.com/sdmitrovic/ISARA_code. A dedicated website for ISARA has been created, where instructions on how to download and use this code are found. The website is located at https://sdmitrovic.github.io/

ISARA_code/ and updates are continuously being made.

*Data availability.* ACTIVATE airborne data are available through https://asdc.larc.nasa.gov/project/ACTIVATE (ACTIVATE Science Team, 2020).

*Author contributions.* SD and JS prepared manuscript with all co-authors involved in review and editing. JS, SD, and SS performed all consistency analyses detailed in this study. All other authors provided input for the manuscript and/or participated in data collection and

processing.



*Competing interests.* The authors declare that they have no conflict of interest.

*Acknowledgements.* Funding for this research was provided by the NASA ACTIVATE mission, a NASA Earth Venture Suborbital-3 (EVS-3) investigation funded by NASA's Earth Science Division and managed through the Earth System Science Pathfinder Program Office. A.S. and S.D. were supported by ONR grant N00014-22-1-2733. J.S.S. was supported by the NASA Postdoctoral Program at NASA Langley Research Center, administered by Oak Ridge Associated Universities under contract with NASA. JSR was supported by the Office of Naval Research Code 322. The authors gratefully acknowledge the NOAA Air Resources Laboratory (ARL) for the provision of the HYSPLIT transport and dispersion model and READY website (https://www.ready.noaa.gov) used in this publication. We wish to thank the pilots and aircraft maintenance personnel of NASA Langley Research Services Directorate for their work in conducting the ACTIVATE flights.





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
