# Peer review of "Closing the Gap: An Algorithmic Approach to Reconciling In-Situ and Remotely Sensed Aerosol Properties"

_EGUsphere, 2024_

## Author Comment (AC1)

**Response to Reviewer 1**

We thank the reviewer for their feedback and constructive criticism, which have helped us significantly improve our manuscript. We have taken care to address each comment with a direct response. The text from your comments are shown in black and our responses are shown in blue. Responses include the manuscript text that was changed, removed, or added.

**Overall Notes** This paper introduces a python tool aimed at associating in situ aerosol measurements obtained during aircraft field campaigns with remote sensing aerosol retrievals, by addressing the need to compensate for the limited coarse-mode throughput of aircraft inlets and to hydrate in situ samples when comparing with ambient (remote sensing) observations.

As these calculations must be made if in-situ field data are to be used to validate remote sensing retrievals quantitatively, the algorithm presented here represents a useful tool for such applications. The observations used to test this approach were acquired during the ACTIVATE field campaign, and the scope of the present study is limited to fine-mode sulfate and organic aerosol, and a coarse mode taken as sea salt. They impose assumptions that limit considerably the applicability of the current implementation - constant refractive indices over the spectral range, spherical particles, parameter assumptions required to calculate the hydrated CRI, etc. However, these are stated clearly, which is as much as one can ask in an AMT paper. The remote sensing data were obtained from the HSRL-2 and RSP aircraft instruments, which avoids some of the sampling differences that arise when in situ measurements are compared with spacecraft measurements. As such, the approach seems most applicable for validating aircraft field measurements.

In summary, the paper develops a useful tool and presents a thorough analysis of its performance. For general application, there are significant limitations in the assumptions made, but given that the analysis is circumscribed to a narrow set of relatively favorable conditions, I think this is acceptable for publication in AMT, perhaps with minor modifications as suggested below. Thank you for the comment. We agree with this assessment and, while some of the text has changed, this description of our work is still true. We address each of the suggested changes below, but no specific changes are made as a result of this summary.

**A Few More Specific Notes** Line 244. It might be useful to mention how far the aircraft travels in 45 seconds, to provide a sense for the horizontal resolution of the SMPS and other, aggregated measurements. Thank you for the comment. The text "that can travel of 8 km across the ground in 45 seconds" has been added to the end of this sentence.

Line 270. Might be worth noting that remote sensing is more sensitive to volume than number concentration specifically for particles smaller than the observing wavelength. For particles larger than the observing wavelength, sensitivity is greater to particle cross-sectional area. Thank you for the comment. We agree this is relevant information and we have added it to the paragraph that now reads as follows:

> In general, remote sensors are not as sensitive to particle number concentration as they are to particle surface area and volume concentrations. For particles larger than the remote sensor's observing wavelength, the remote sensor is most sensitive to particle cross-sectional area. For particles smaller than the observing wavelength, the remote sensor is most sensitive to volume concentration. Given the sensitivity of remote sensors to surface area and volume concentration, this work also discusses the logarithmic size-resolved aerosol particle surface area concentration ($a^o$) and logarithmic size-resolved aerosol particle volume concentration ($v^o$).

Line 454. Comparing Fig. 1 with Fig. 5, and taking the y-axis scales into account, I would say "...overall, much less variance." Thank you for the comment. We agree this is a fair assessment and have changed the text to be "..., but the synthetic data see overall much less variance"

Line 469. As this is synthetic data, doesn't the statement here just mean that the numerical coding was done correctly? Not a bad thing to mention, but the statement here makes the observation sound more fundamental. Thank you for the comment. We agree this finding has been overstated and have changed the sentence to be "To demonstrate

the functionality of this analysis, the synthetic data generation and retrieval processes were repeated with zero measurement noise, which results in a rate of successful retrievals of 100%."

Line 494. A word seems to be missing from this sentence.
Thank you for the comment. We appreciate catching the error. We have changed the text to be "...which suggests that the ISARA-derived products are less reliable at relatively low scattering signal (i.e., signal $< 5\,\mathrm{Mm}^{-1}$)"

Lines 498-500. By way of explanation, wouldn't the 700 nm channel likely be the most sensitive to coarse-mode particles, for which many of the assumptions might be less applicable?
Thank you for the comment. We agree that this increase in MRB with increasing wavelength is notable and while it is only a few percent difference, we have added the following text to capture this: "Because this bias increases with increasing wavelength, it is possible that some of this discrepancy is due to larger particles that are more commonly comprised of dust, but this trend only accounts for a few percent difference in MRB".

Section 3.1.3. Just wondering how representative of the entire column the in situ data sampling might be. I realize the HSRL-2 data are height-resolved, which can help assess the vertical heterogeneity compared to the in situ sampling.
Thank you for the comment. We have analyzed this with some level of detail in Section 3.2 with regards to the RSP-derived and ISARA-derived products.

Figure 10. There appears to be a lot of scatter in the data, which the text does not seem to acknowledge. This is probably not surprising - in addition to the limitations discussed in the paragraph about this figure, given the likely horizontal and vertical variability in particle concentration combined with differences in sampling.
Thank you for the comment. To address this we have added error bars and standard deviations where appropriate throughout the text. We agree that there is a lot of scatter, however this appears to fall within the standard deviations of many of the points. The text has been changed as follows to address this: "Similar to $C_{\mathrm{ext}}$ and LR, the standard deviations in the HSRL-2+RSP-derived $N$ often encompass the 1-to-1 line.".

---

## Author Comment (AC2)

**Response to Reviewer 2**

We thank the reviewer for their feedback and constructive criticism, which have helped us significantly improve our manuscript. We have taken care to address each comment with a direct response. The text from your comments are shown in black and our responses are shown in blue. Responses include the manuscript text that was changed, removed, or added.

This manuscript presents a methodology to compare and reconcile aerosol observations from various platforms, so called ISARA algorithm, developed for the ACTIVATE mission data. It provides a useful and rather comprehensive example of attempting to achieve closure between diverse in-situ and remote sensing measurements. The study clearly addresses a complex problem and contributes relevant information for aerosol measurement harmonization. However, it would benefit from clarifying its broader motivation, better articulating its limitations, and improving the language and structure in some sections.

My main concern is that the manuscript abstract, and introduction do not seem to precisely describe the motivation of the study. The introduction frames the problem mainly as a data closure challenge, potentially giving the impression that a general solution (i.e., ISARA) will be offered. For example the sentence "Despite the important findings from these studies, systematic and streamlined closure of aerosol data sets has not been yet achieved." However, the ISARA algorithm appears heavily tuned to specific conditions, relying on a priori information about aerosol composition, size, and shape, additionally being very limited in the atmospheric and aerosol conditions where it can be applied. This is perfectly understandable, but a clearer explanation of how broadly applicable the algorithm is would strengthen the outcome. The introduction could therefore more broadly discuss the complexity of in-situ vs remote-sensing aerosol validations, compilations and simulations – as for setting the scheme. I hope that authors consider this, and could slightly streamline the introduction and structure of the paper to better reflect the content.

In summary, this manuscript addresses a complex and relevant topic in atmospheric aerosol measurement by providing a detailed example of integrating multiple observation platforms. The methodology is simple but thoroughly described, and the analysis is well presented, but the applicability and limitations of the ISARA algorithm require clearer discussion. The manuscript would benefit from a refined motivation and improved language in certain sections. I recommend minor revisions before acceptance at AMT.

We thank the reviewer for their thoughtful and constructive feedback. In response to the concerns raised, we have revised the abstract and introduction to better articulate the overall motivation of the study. Specifically, we now clarify that this work focuses on assessing consistency across platforms as a necessary step toward achieving rigorous external closure, rather than offering a universal solution to the closure problem. We also explicitly acknowledge the limitations of the ISARA framework, including its dependence on assumptions about particle shape, composition, and the applicability to specific aerosol regimes. These clarifications are intended to better align the manuscript's framing with its scope and contributions. We believe these revisions strengthen the manuscript and more accurately convey the goals and context of our analysis.

**Minor Rather Technical Comments: Abstract:**

– L6: Suggest removing "aircraft" as the methodology can apply to other in-situ data platforms as well. Or can it? Please consider this also when writing introduction. Thank you for the comment. We do agree that the framework offers a great deal of flexibility in the in-situ platform used. We have removed "aircraft" from this sentence and made subsequent changes within the introduction as well.

**Introduction:**

– L33: Remove the word "parameter". Thank you for the comment. We have addressed this error. We also removed the example to discuss specific parameterizations later in the text.

– L51: Consider adding 1–2 sentences summarizing key findings from past studies for context. Thank you for the comment. We have added the following sentences to the text to provide more information on the findings of a few of these

studies: "These studies consistently find that the extinction and backscatter coefficients derived from in-situ instruments are systematically low compared to those derived from HSRL-2." and "In particular, Pistone et al. (2019) found poor agreement between in-situ- and RSP- derived total SSA.".

– L52: Clarify "these instrument data". Which instruments are being referred to? Thank you for the comment. We have clarified the statement to be "... remote sensing data'

– L57: Suggest rephrasing to "...using a Nafion membrane dryer in the sampling line". Thank you for the comment. The text has been altered according to the comment and is now: "...using a Nafion dryer or heating in the sampling line".

– L86: Check for a possible extra "is". Thank you for the comment. The extra is has been removed from this sentence.

– L91: Sentence structure is unclear-both content and grammar could be improved. Also, please clarify if the methodology is expected to be broadly applicable or limited to specific platforms and conditions. Thank you for the comment. The methods described in this paper are open source and flexible due to using MOPSMAP to handle the optical properties. The sentence has been changed to be: "While the current study focuses on the consistency analysis between in-situ- and remote sensing-derived aerosol properties of the more common spherical aerosol particles, it is hoped that the framework described in this study serves as an open source foundation that can be easily expanded and used to fully understand the information train between all manner of measurements and therefore enable systematic closure of field campaign aerosol data."

**Measurements:**

– L198–202: Some grammar and sentence construction issues—please revise for clarity. Also, consider using micrometers (μm) instead of nanometers (nm) for >1 μm sizes. Thank you for the comment. We have updated the units to μm where there weren't ranges going into the nm scale or significant figures would require multiple digits after the decimal place. We have clarified the sentence in question, which now is "The effective upper size cut is $D = 5\,\mu m$ for all 2020 data."

– L202: Nephelometers and absorption measurements are introduced here without prior explanation. Reorganizing or cross-referencing earlier sections may improve flow. Thank you for the comment. We have added the Section 2.1 to provide a primer on particle properties and general background for these measurements.

– L213: Was the PSAP measurement also conducted at <40% RH? Please specify. Thank you for the comment. The PSAP sample stream is dried by heating the optical block to 35°C. The text has been updated to include this sentence.

– L218: Replace "variety of errors" with a more descriptive phrase identifying specific artifacts or correction needs. Thank you for the comment. We have replaced "variety of errors" with "transmittance and flow errors".

– L223: How much data was excluded by the $>1\,\mathrm{Mm}^{-1}$ cutoff for scattering and absorption? Justify the threshold. Thank you for the comment. This comment led to us evaluating the $1\,\mathrm{Mm}^{-1}$ and determining that 0 was a better value for absorption. This yielded a 10x increase in the data available for retrieval. This threshold was changed accordingly to 0 $\mathrm{Mm}^{-1}$. The results and conclusions have been updated accordingly.

90      – L245: Clarify what is meant by "most useful analysis" of profiles extending >1 km. Thank you for the comment. The profiles are most useful for comparison with the RSP-derived products. We have added the following text to clarify this information: "As such, the external consistency analysis is most useful from vertical profiles where the in-situ platform samples the column of air above an arbitrary ground point. Vertical profiles where the extent is more than 1 km are most useful for comparing with the column-averaged aerosol particle properties derived from the RSP-measurements.".

95

     – L257: Consider rephrasing "1-second data" as "native time resolution data" for clarity. Thank you for the comment. We tried to clarify this with the following text "The data that are in their native resolution are averaged to 45 seconds using the NASA standard merging tool."

100      – L282, L286, L334: "ACTIVATE region" should be defined more precisely—preferably with coordinates or a map. Also specify which areas or conditions were excluded and why. Thank you for the comment. We have clarified the region that ACTIVATE was bounded by within the text. The following sentence was added to Section 2.2: "The extent of the North Atlantic region that was sample during ACTIVATE was within bounds of 58–78°W and 28–42°N.". We have described the cloud filtering process and the focus on when the Falcon (i.e., the in-situ platform) was performing vertical profiles

105 in coordination with the King Air. It is outside the scope of this work to describe the methodologies that guided the ACTIVATE mission.

     – L289: Clarify "mid-point particle diameter." Why is geometric mean diameter not used? Mid-point particle diameter is synonymous with geometric mean diameter. The text has been clarified as follows: "Note that this process uses the

110 mid-point (i.e., the geometric mean) particle diameters from each SMPS and LAS channel.".

     – L285–295: The assumption of sulfate-only aerosol seems oversimplified. Given the limited RRI range and the potential presence of organics or other compounds, the assumptions behind the ISARA-derived optical properties need better justification. Also, specify how wavelength conversions were done and what values of AAE or SAE were used. Thank

115 you for the comment. We agree that the limited RRI does limit the scope of this study, however, looking at Li et al. (2023), it would appear that most mixtures of aerosol particles will have an apparent real refractive index (RRI) that falls in the 1.51–1.55 range. As such we have updated the text with the following: "Furthermore, external mixtures of many aerosol species have an apparent RRI that falls between 1.5 and 1.58 (Li et al., 2023). If an external mixture of aerosols is dominated by a RRI outside of this range, it is likely that the assumptions of a spectrally flat CRI and sphericity are not

120 longer valid.". Additionally, AAE and SAE were not used. We calculate the coefficients at their native measurements. For example, dry scattering coefficients were calculated at 450, 550, and 700 nm wavelengths and dry absorption coefficients were calculated at 470, 532, and 660 nm wavelengths. The final ambient properties were again recalculated using MOPSMAP at each of the HSRL-2 wavelengths.

125      – L326: Missing preposition—please review. Thank you for the comment. This sentence has been changed to the following: "After calculating the set of forward-modeled $C_{\text{scat,wet}}$, we use $\gamma$ to derive the "measured" $C_{\text{scat,wet}}$ (i.e., $C_{\text{scat,RH}=80}$).".

     – L334: Again, clarify the threshold inconsistency between L223 and here. How much absorption data were actually usable above $1\,\text{Mm}^{-1}$? Thank you for the comment. As discussed in the response above, we have re-evaluated the use of

130 $1\,\text{Mm}^{-1}$ minimum threshold. Additionally, this sentence was not directly connected to the figure so it has been corrected as follows: "It is observed that aerosol particles in the ACTIVATE region generally have low absorption (IRI $\leq 0.01$) and low hygroscopicity ($\kappa \leq 0.1$).".

- L615: The manuscript states that the region was chosen for "diversity of aerosol and meteorological conditions," yet the assumptions made to enable closure seem to contradict this. Consider rephrasing or qualifying this claim. Thank you for the comment. We agree that as written this is somewhat contradictory. We have attempted to resolve this contradiction by focusing on the large volume of data available. The text has been changed as follows: "...large volume of statistically-rich aerosol measurements collected over three years of operations. Although the ACTIVATE region does feature a variety of aerosol and meteorological conditions, many of the ACTIVATE missions were carried out in cloud-free conditions without detectable influence from dust or smoke, making the data set well-suited for the consistency analysis performed in this study.".

---

## Author Comment (AC3)

**Response to Reviewer 3**

We thank the reviewer for their feedback and constructive criticism, which have helped us significantly improve our manuscript. We have taken care to respond directly to each comment. The text of your comments is shown in black, and our responses are shown in blue. The responses include the manuscript text that was changed, removed, or added.

The paper by Dmitrovic et al. describes a Python toolkit (ISARA) aimed at achieving closure between lidar and polarimeter observations of ambient aerosol properties on one hand, and in-situ measurements of the PSD, scattering and absorption on the other hand, which are limited by drying and by inlet effects. The toolkit is validated through in-situ observations (internal consistency), simulations (synthetic consistency) and remote sensing observations (external consistency). It is my feeling that the stated goals of the paper are of high importance, but that they are not sufficiently developed beyond the technical stage to be proposed for a publication. I would invite the authors to consider a more scientific and less technical approach in writing this paper, and to allow the time and the effort that this can represent at this stage I suggest to REJECT the manuscript. I will be willing to help with reviewing a revised manuscript in case the the below MAJOR points are addressed in full.

Thank you for the comment. We agree that the goals of the paper are of high importance. We also agree that the science focus of the paper needed some expansion. We therefore added a detailed analysis of more aerosol properties that was not described in the previous draft, including comparisons against RSP-derived coarse-mode properties that have never been evaluated in previous studies. Since our results show good agreement to within measurement uncertainties compared against advanced remote sensors (lidar and polarimeter observations) we do not agree that the paper is not developed past the technical stage. We further feel that the technical approach is an important development on its own, since it is a foundation that can be used to directly replicate our findings using the code and data provided. Although our results already show significant scientific promise, they can also be readily improved and expanded upon in the future. We show consistency between remote and in situ aerosol products for several ambient aerosol properties that have not been evaluated by previous studies, and we indicate which properties are not consistent. We present a comprehensive analysis using spherical particle models. It is outside the scope of this paper to perform additional analyses of more complex particle types such as dust and smoke since they require us to use modeling of non-spherical and heterogeneous particle shapes and to revisit many of the assumptions made in the remote sensing and in-situ retrievals themselves. Future work will involve demonstrating how we can leverage the additional capability of libraries like MOPSMAP to evaluate these more complex particle types. We have made extensive revisions of the manuscript to address the scientific shortcomings and to address each of your specific comments below. As such, we feel that the paper should not be rejected and we welcome the reviewer to reconsider our revisions.

**MAJOR POINTS:**

1. The paper takes a technical stance to the task, e.g. insisting on unnecessary details such as the use of the python programming language and the MOPSMAP scattering code. I would invite to present an algorithm, and to discuss its assumptions, limitations and uncertainties, rather than a piece of software. Concerning the scattering calculations, given that the authors only assume spherical particles, MOPSMAP is only a library allowing Mie scattering calculations and any other Mie scattering code would achieve the same results, therefore the scientific approach is to indicate that the framework of Mie theory is being used, and not to indicate which scattering library is used. The usage of the MOPSMAP code could most probably be of interest in the case of non-spherical scattering, but non-spherical scattering is not envisaged here.

Thank you for the comment. As discussed above, we agree that there needed to be more description of the science behind aerosol properties. The structure of the paper has been changed accordingly. We have added Section 2.1 to add more scientific background to our technical description. In the revised draft, we now clearly explain the assumptions, limitations, and uncertainties associated with this method. Furthermore, the analysis performed in this study can be easily replicated by accessing the software and data products associated with this study, which we believe is important in its own right, since we intend ISARA to be used as an open source toolkit for field campaign scientists to use.

2. There has been previous literature attempting the same task, and this needs to be accounted for and mentioned. I bring here for example the IRRA approach of Tsekeri et al (AMT, 2017). I am sure that by searching the literature the authors can also find other references on the same topic. The authors should explain how their paper fits within the existing

literature, and how similar or different it is from other approaches, indicating pros and cons of each. Note that "previous works" are also cited at line 547, but no references are given. Thank you for the comment. We have added more detail in the introduction and in the methods section that indicate how this work is a successor to previous works. While Tsekeri et al. (2017) did do a retrieval algorithm using in-situ dry measurements and probes, their algorithm requires the lidar data as part of this algorithm. This is distinctly different from the approach taken in this work that does the retrieval independently of the lidar measurements and then compares the aerosol properties afterward. We have made note in several places through the text that the coarse-mode aerosol properties are prescribed and the size-distribution of the coarse-mode is taken from wing mounted probe measurements and cited studies that have done analysis of coarse-mode aerosols with wing mounted probes.

3. The stated goal is a "rigorous external closure" (see line 4). I find this to be a real contradiction with the state-of-the-art achieved by the paper itself, given that it is a "preliminary consistency" and a "preliminary effort" (see lines 19 and 72). This may question whether more work is needed before a paper is submitted to AMT. The sentence at lines 546-548 seems to confirm this impression. Thank you for the comment. We have removed the word "preliminary" from this line since we have added several novel comparisons in the revised draft. These results agree within the uncertainties of the measurements. This sentence is now written as "While this study focuses on spherical, sulfate-dominated aerosol mixtures, its overall success demonstrates that ISARA has the potential to support systematic and streamlined closure of aerosol datasets across diverse field campaigns and aerosol regimes.". The focus of this work is to prove the functionality of ISARA under conditions that are generally realistic but relatively ideal in terms of ignoring the possible presence of non-spherical aerosols within the marine environment. Furthermore, we state that rigorous external closure over a wide range of all possible marine and terrestrial scenarios will require future studies and changes to some of the assumptions made during this study, but will not require a fundamental change to the underlying software (i.e., ISARA).

4. The computations are done for a specific atmospheric scenario only, with a mixture of sulfate and organics in the fine mode and sea salt in the coarse mode (lines 104-109 with a clear mention that "this will limit the scope of this study"). Whereas this is fine in itself, it removes generality away from the ISARA approach and from its stated goal of a generally applicable approach. It is important to state the limited scope from the onset of the paper, rather than add the limitation further down the line. Moreover, given that ACTIVATE operated also in Bermuda (line 136) why is the influence of dust aerosols not also investigated? Thank you for the comment. We have added the following text to the abstract to list the assumptions used in this study more clearly from the start: "While this study focuses on spherical, sulfate-dominated aerosol mixtures, its overall success demonstrates that ISARA has the potential to support systematic and streamlined closure of aerosol datasets across diverse field campaigns and aerosol regimes.". Furthermore, while dust may have been observed during ACTIVATE, it is outside the scope of this paper to analyze these more complex aerosol types until we are confident in the comparisons using more typical aerosol types. We have reiterated this point throughout the text.

5. After clarifying the issues affecting in-situ observations, the authors claim (lines 89-90) that "the ISARA attempts to overcome these limitations by estimating the contribution of coarse-mode particles". It is unclear how this can be achieved, and I would state that this is not possible without a dedicated measurement for the coarse particles (e.g. using an open-path OPC). A CDP is later mentioned at lines 226-234, therefore the problem may perhaps simply be of being explicit and honest from the onset about the fact of using a dedicated measurement for the coarse mode. Thank you for the comment. We agree that this text was misleading and have changed the text to clarify that we are using dedicated probe measurements. In this case, we preferentially used the CAS, but relied on the CDP or FCDP where the CAS was not available.

6. Figure 9: ISARA shows underestimation. Figure 10: low level of closure. This does not look good for the ISARA method Thank you for the comment. We do not feel that this underestimation is a critical weakness because the comparison shown yields results within a reasonable uncertainty. We have made this point evident throughout the text by adding

the corresponding retrieval uncertainties and standard deviations where possible. This algorithm represents a solid beginning, and future improvements are likely possible.

7. The sentence "a potential reason for detecting low-RH aerosol is the presence of smoke from fires" (lines 588-589) is in contradiction with the assumed marine environment, but is also presented in a way that makes it appear unjustified. Please note that water vapour is a combustion product, hence fires should contribute to increasing absolute humidity and not to decrease it. The whole narrative at lines 590+ appears incoherent: note that figure 13 shows fires and not smoke aerosols as stated here. It is suprising also to read that aerosols "are becoming drier due to colder temperatures" given that lowering the temperature (with constant WVMR) has the effect of increasing the RH. I agree with the sentence on line 593 that "further work will be done" and I feel that it must be done before re-submitting the paper. We agree that this text was erroneous and we have removed it. We now focus attention to the presence of winter conditions, which is far more important than any amount of water vapor generated from combustion. This is especially true because no smoke aerosols were detected in the profile and because the profile was located far from any source of smoke. If we expanded the scope of this study to include more atypical particles, the paper would be incredibly long and unfocused. Instead, we aim to demonstrate that this method works well for spherical aerosols found in marine environments, and that the overall approach is also useful for future research.

8. Overall good agreement (line 602) feels an unjustified statement here. The same applies to the "successful retrieval" (line 622). The evidence is not so clear-cut. Thank you for the comment. After performing extensive revisions throughout the paper, we feel that overall these statements are justified but have made it a point to indicate which properties were shown to be consistent and which were not.

**MINOR COMMENTS:**

5. "In-situ instruments cannot efficiently sample coarse-mode particles due to limitations in the inlet cutoff diameter" (line 65). This statement is incorrect given that there are several open-path airborne instruments such as the CDP, the CAPS, etc. that overcome these limitations (initially developed for cloud particles and later extended to use in aerosol layers). It is well-known that the FAAM research aircraft has successfully sampled coarse and giant dust particles up to 300 um diameter with such probes (see e.g. Ryder et al, ACP, 2015; Marenco et al, ACP, 2018). Thank you for the comment. We have changed the text throughout the paper to clarify that we are using the coarse-mode measurements in the same way as the sources you have cited.

6. The definition of the fine-mode (0.09-1 um dry diameter) and coarse-mode (> 1 um ambient diameter) regimes (lines 83-84) is weird given that there could be an overlapping zone between fine and coarse (particles with dry diameter < 1 um and ambient diameter > 1 um would be in both regimes). I would suggest to consistently refer to either the dry or the ambient diameter for discriminating the two modes. Thank you for the comment. This was an error on our part. We have changed the word dry diameter to ambient diameter.

7. Line 6: the symbol $\kappa$ is undefined. At the end of page 14, the relationship between $\kappa$ and f(RH) must be explained. Thank you for the comment. We have decided to remove the explicit parameterization of $\kappa$ from the abstract and moved it to the introduction, which is expanded further in Section 2.4. In this section, we explain that  is related to scattering and absorption coefficients through the growth factor equation abd relative humidity.

8. Joint flights (line 135): it is unclear which two airplanes performed joint flights. Thank you for the comment. Because we address this directly in the following paragraph, we have removed the word "joint" from this sentence.

9. "The novel vertically-resolved aerosol particle number concentration" (line 184): I suggest to add the word "estimate" to this statements, given that number concentration from lidar is estimated based on assumptions. Thank you for the comment. We have added the "estimate" to this sentence.

10. Inlet cutoff at 5,000 nm (line 199): indicate how the cutoff diameter was determined. Thank you for the comment. This information was determined by using ground station measurements collocated to the aircraft measurements. Although it is outside the scope of this paper to describe this process, we have cited McNaughton et al. (2007) that evaluated this cutoff diameter.

11. For random variables such as the scale factor (line 365), the $\kappa$ (line 367) and IRI (line 370), the random distribution used must be given. For the scale factor, you can observe the random distribution in Fig. 5. For the random distributions of and IRI you can observe the random distribution in Fig. 5. We feel that showing the random distributions of these variables would require an additional separate figure that is unnecessary given the two figures already present with this information embedded.

12. Nephelometers (line 202) are being mentioned before being introduced (line 213 and following). I suggest that the full instrument set must be introduced before discussing their details such as installing a cyclone. Thank you for the comment. We have added Section 2.1 to address this concern.

13. Equation 1 using RH of 20 and 80% is inconsistent with having observations at 40 and 85% (line 215). Please clarify. We have changed the text and redefined $f$(RH) more clearly in Eq. 15. The measurement of humidified scattering was done at 85% RH, however, we use $\gamma$ to calculate $f$(RH) for the 80/40% RH

14. Line 248: "the methods standardized by the merging tool". It is unclear what the authors refer to. Thank you for the comment. We agree that this text was ambiguous. We have changed the sentence to be: "Data in their native resolution are averaged to 45 seconds using the NASA merging tool.". A description of how the merging tool works is found in https://www-air.larc.nasa.gov/missions/etc/onlinemergedoc.pdf, which is cited in the previous sentence of the text.

15. Symbols: there is an unclear use of symbols N_a and n_0 (seeming to refer to the same variable). Also, in lines 293 and 294 C_calc refers to different variables, hence I suggest using different symbols. Thank you for the comment. We have corrected our terminology throughout the paper to make the distinction between N and $n^o$ clear. We have also corrected the $C_{\text{calc}}$ term to have the "abs" or "scat" subscript where appropriate.

16. Equations 3 and 7: the bounds of the integral should be log(Dmin) and log(D_max), and not dlog(Dmin) and dlog(Dmax). Thank you for the comment. We have corrected this error.

17. Line 270: remote sensing sensors are normally considered more sensitive to the surface area and not the volume. Please correct. Thank you for the comment. As another reviewer pointed out, this statement was not fully correct. The sensitivity of the remote sensor to surface area or volume depends on the observational wavelength. We have added more detail with the following text: " In general, remote sensors are not as sensitive to particle number concentration as they are to particle surface area and volume concentrations. For particles larger than the remote sensor's observing wavelength, the remote sensor is most sensitive to particle cross-sectional area (i.e., surface area concentration). For particles smaller than the observing wavelength, the remote sensor is most sensitive to volume concentration.".

18. Stitched data (line 279): this must be documented. Thank you for the comment. We have explicitly documented this step through the following text: "After this step, the SMPS and LAS size distributions are stitched at a diameter of 94 nm, which is the upper bound of the size-range by the SMPS and the lower bound of the LAS size-range.".

19. Full range of particle sizes (line 279): this is incompatible with the inlet cutoff. Please clarify. Thank you for the comment. We felt that this sentence was redundant and unclear, so we have removed it. This information is discussed directly above in the previous paragraph.

20. Figure 1 caption: I suggest to use percentiles instead of minimum and maximum to reduce the influence of outliers. Thank you for the comment. We have changed Figure 1 to reflect the 10th and 90th percentiles and updated the corresponding caption.

21. CRI averaging (line 291): it is unclear over which dataset the averaging is being done. Thank you for the comment. In the following sentence, we now state: "For a CRI to be valid for averaging, all three of the computed scattering coefficients must be within 20% of the corresponding measured dry scattering coefficient $\left( \frac{|C_{\text{scat,calc}} - C_{\text{scat,RH=40}}|}{C_{\text{scat,RH=40}}} < 0.2 \right)$ and all three of the calculated absorption coefficients must be within $1\,\text{Mm}^{-1}$ of the measured absorption coefficients $\left( |C_{\text{abs,calc}} - C_{\text{abs,RH=40}}| < 1\,\text{Mm}^{-1} \right)$". This now unambiguously defines what conditions are required for a CRI to be considered valid for averaging.

22. Co-location (line 401). As observations are done on-board the same platform, they are certainly co-located, therefore this is unclear. Thank you for the comment. For clarity, we have removed the word "collocation" from this sentence.

23. Line 404: I suppose that data are removed when the cost function is larger than a threshold, not lower. Please correct. Thank you for the comment. We have corrected this error and replaced the "<" with ">" in addition to removing the extra "to be" in the text.

24. "coarse mode AOD is limited to < 0.1": why? As stated in the text directly above this quote we state "To limit the presence of coarse-mode aerosol particles in this analysis". We have added the following text to the preceding paragraph to better motivate this decision: "Because a significant presence of coarse-mode particles would be atypical within the altitudes that are sampled by the Falcon, we attempt to limit the amount of coarse particles in the columns of compared data.".

25. LDR (line 407): clarify if you refer to VLDR or PLDR, because they bare not the same. The threshold at 0.13 (line 410) is not so small, at least for VLDR, and could indicate dust presence. Thank you for the comment. We have clarified the mathematical definition of LDR. Because the HSRL-2 description already stated that this is aerosol-specific data, we feel that an additional distinction between PLDR and VLDR is not needed.

26. Successful retrieval rate (line 482): please clarify if the retrieval rate means that a solution is found, or that the solution found is close to the observations). Thank you for the comment. This is simply the number of successful retrievals over the number of attempts, where the success is based on the maximum thresholds in the deviation of scattering and absorption. The text has been clarified by adding "$\left( \frac{\text{number of successful retrievals}}{\text{number of attempts}} \right)$" to this sentence.

27. Lines 483-486. The authors raise an important limitation and they should perform a more in-depth analysis to disentangle the causes. Note that for measurement noise (mentioned by the authors) this should be possible to address with the current data. Concerning the issue of non-sphericity, this can be run using MOPSMAP. Thank you for the comment. We agree that more in-depth analysis could be done for future studies, but we feel it would distract from the focus of the paper on more typical marine cases. We have also added a discussion to further support the low signal issues that were observed.

28. Acronyms (NRMSD, MRB, NMAD) must be explained when first used. Thank you for the comment. We have removed the discussion of MAD from the text as it is redundant to NRMSD. Additionally, we have ensured that all consistency statistics are defined when first used.

29. Line 565: specify that you refer to case #10 of table 5, because there are two flights on this date. Thank you for the comment. As mentioned above, we have removed the extra case from this table and updated all the text accordingly to be consistent.

30. "are likely organic and sulfate-dominated mixtures": this sentence is unexplained and unsupported by evidence Thank you for the comment. We have added the following text to support this statement: "This information indicates that the fine-mode particles being sampled were anthropogenic in origin and are likely sulfate-dominated mixtures with organic aerosol species. This is what would also be expected for this marine environment based on climatological evidence (Braun et al., 2021).".

31. "marine environment" (line 588): This statement seems inconsistent with the Hysplit trajectories showing continental influence. Thank you for the comment. We do not feel that these are contradictory statements. A notable feature of the marine environment of the Northwest Atlantic is the persistent influence of anthropogenic outflow. We have clarified this in the previous response.

32. "the data from case 12 are also shown in Table 5" (line 597): why? Thank you for the comment. We agree that showing these data here was unnecessary and we have removed this case from the included tables.

33. "1064 m": it is actually 238-4499 m Thank you for the comment and for catching this typo. We have corrected the value to be 4261 m.

34. Table 5 caption should read "case studies 10 and 12" Thank you for the comment. We have changed this table to include only data from case study 10, which is now case study 7 in the revised draft.

---

## Referee Report (RR2)

**Closing the Gap: An Algorithmic Approach to Reconciling In-Situ and Remotely Sensed Aerosol Particle Properties**

The authors reported the ISARA retrieval framework in detail and attempted to validate it with ACTIVATE mission data. I appreciate the plethora of information re: the approach and the comparison datasets. However, I found the logic to be confusing at times despite the high level of details (see my specific comments and questions below). In addition, it would be helpful if the authors validated the retrieved aerosol properties (kappa) with any available aerosol chemical composition from the same mission. Perhaps it is because the manuscript is rather long, there are quite a few typos and odd sentence structures. This manuscript would benefit from a major revision that I hope will address my questions and comments below.

**General comment:**

For particle diameters > 1  $\mu$ m, the authors should consider reporting them in  $\mu$ m instead of nm (e.g., 3488 nm on L281).

The authors should consider updating the symbol of surface area concentration from dA/dlogD to the conventionally used name of dS/dlogD.

In a few places (e.g., L427), the authors used the term "aerosol particles" but all aerosols are particles.

There are a lot of repeated definitions of acronyms (e.g, HSRL-2 on L233, Table 2 title and content) that should be cleaned up. There are also instances of using acronyms first then defining them later (e.g., CRI appeared on L156 then defined on L177).

Was there relevant aerosol chemical composition available from the ACTIVATE mission (e.g., AMS) that the authors could have used to compare with derived hygroscopicity parameter kappa and real refractive index?

**Specific comments/questions:**

L120-122: the authors already defined the diameter range for fine mode and coarse mode particles on L95-97, so remove them here.

L166: define what phase function represents and why it is introduced

L205-211: is it necessary to list out the dates of the ACTIVATE deployments instead of referencing the campaign overview by Sorooshian et al. (2023)? This information does not seem to be used later.

L239-247: a lot of repeated information on HSRL-2 products. Please rewrite this part of the paragraph to be more concise.

L 274: extra words "the 30"

L275: what is "ram heating" in the context of SMPS data?

Table 1: in HSRL-2 section, missing a comma between "532" and "1064 mn". Also, repetitive definitions of HSRL-2 and RSP in both the table title and the table content. Consider adding acronym of extinction coefficient, backscatter coefficient, etc. in parameter description for clarity.

L291: "success rate" of...what exactly? Please be specific. Is it in retrieving coarse-mode particles?

L319: extra word "are"

L339: water vapor "density" or mixing ratio?

L346: travel "up to" 8 km?

L369-370: repeated definitions of  $n^{\circ}$ ,  $a^{\circ}$ ,  $v^{\circ}$  (previously defined on L190-197). This repeat of definitions also happened to other acronym and symbols (e.g., Table 2 title). Please re-check the other sections of the manuscript as well.

L383: why did the authors restrict calculated absorption coefficients to within 1 Mm-1 of the measured values? Restricting the calculated values within a constant 1 Mm-1 may bias the filtering process to higher absorption values, where 1 Mm-1 difference is a smaller fraction of the total value (e.g., 10% of 10 Mm-1) vs. smaller absorption values (e.g., 50% of 2 Mm-1). Should this restriction be a percentage difference like with scattering?

L427: the authors stated that the derived hygroscopicity for aerosol in the ACTIVATE region is low ( $\kappa \le 0.1$ ) but also concluded that the aerosol population is sulfate-dominant. This value of  $\kappa$  seems really low. Organic is often assumed to have kappa value of 0.1. Sulfate (inorganic) are usually with kappa > 0.4. An aerosol population dominated by sulfate and organic as stated by the authors should have kappa >> 0.1.

L432: misspelling with extra "i" in scattering

Figure 3: this does not seem like a correct representation of the calculation process. On L415-420, you mentioned that C\_RH,meas was actually calculated from C\_dry,meas and estimated gamma from Eq. 7. Really, the authors were using a single measured C\_RH,meas(85%) and C\_dry,meas to estimate gamma. Then, they used gamma to derive C\_RH,meas at 80% - which now is an estimated value and not a measurement. Why didn't the authors use C\_RH,meas(85%) from the nephelometer instead?

L452: extra word "aerosol" or "particle"

L461-462: typo with "," between number and μm

L458-464: the use of CAS size distribution to derive coarse-mode particle properties contradict with L320-325 in section 2.3.2, where the authors said to use CDP mainly and CAS only if the other 2 datasets (CDP and FCDP) are not available. Please review and correct the appropriate section. Also, if the author chose CDP over CAS, what is the reasoning when CAS provides a wider range of available aerosol diameters?

L506: the reported % does not need to be accurate to 2 decimal points (e.g., 26.49%), especially when the authors reported the biases as full % point (L587).

L515-517: why did the authors remove HSRL-2 AOD < 0.08 where HSRL-2 AOD uncertainty is 0.02 (stated in Table 1)?

L521-524: I did not follow the logic. What is this 50% AOD difference criteria [between HSRL-2 and RSP derived AODs] based on? Similarly, how did the authors arrive at the filtering criteria of coarse-mode AOD being <0.1?

L555: extra "respectively"

Figure 5: the authors could add lines that represent the range of measured data from Figure 1 on Figure 4 to show the range in synthetic data is smaller than the actual measurement (L578-579). Also, there are currently 2 panels (b) in this figure.

L587: did the authors mean "respectively" instead of "respectfully"?

Table 3: update the table title since it also includes information for Table 4. Also, the authors repeated the same information in the first paragraph immediately following Table 3.

L699: remote sensors or remote sensing retrievals?

Figure 11: why don't we have LR > 2 km for HSRL-2 data? Both C\_ext and C\_bsc are available at these altitudes for HSRL-2 (panel b, e). Also, panel (i) seems to be missing from Figure 11.

L745: compared to case 7, case 9 with high smoke counts and high HSRL AOD would be interesting to investigate further compared to case 7.

Figure 12: the authors should add uncertainty to the calculated kappa, RRI, IRI that were derived with ISARA.

Conclusion: I am not sure why the authors defined ISARA, CRI, etc. again when they had been mentioned all throughout the manuscript. Please remove this redundant information.

L792: Similar to the work...

---

## Referee Report (RR3)

**Closing the Gap: An Algorithmic Approach to Reconciling In-Situ and Remotely Sensed Aerosol Particle Properties**

I would like to thank the authors for addressing some of my previous comments. For the new manuscript, there are enough typos that I would advise the authors to proofread more carefully next time (e.g., L980). In terms of content, despite the authors providing more explanation, I still find the manuscript unclear at times. Below are my questions and comments that suggests this manuscript needs at least MINOR revisions (I think it really sits right between MINOR and MAJOR revisions), but I think the authors can address the following relatively quickly.

**General comment:**

I am not clear in the way the authors set up the synthetic consistency analysis. Specifically, in response to Reviewer 3's comment on expanding the variety of synthetic size distributions, the authors gave Figure 5 as an example of a bimodal size distribution that appears unimodal to me. Are there other examples of size distributions that the authors could offer for different layers of the sampled atmosphere?

Figure 15 shows a clear high bias in remote-sensing retrieved values (e.g., C\_ext) compared to the ISARA values that the authors did not explain well. Besides the reported statistics, the authors seem convinced that the bias is from under-sampling of coarse-mode particles. Given the high uncertainty in retrieved kappa and refractive index values, I would be more convinced in the bias coming from one of these retrieved values.

I am still unclear on the uncertainty associated with the gamma parameterization. Is there a reported uncertainty somewhere on the manuscript that I may have missed?

**Specific comments/questions:**

L204: The ACTIVATE "mission" or "field campaign". The authors' response of to a similar comment made by Reviewer 3 was not satisfactory. On L110 in the Introduction section, the authors already used the term "ACTIVATE field campaign" as well as "ACTIVATE mission" on L138, so there should be no issue to use something similar on L204. "The ACTIVATE featured..." sentence reads awkwardly, otherwise.

L212: Did you mean S I (theta) is the angular light intensity and not S 1 (theta)?

L369: "... 1 um aerodynamic cutoff for scattering coefficient measurements only." Then delete the next sentence. As the sentence is currently, it reads a little odd.

L432: change "an relative accuracy" to "a relative accuracy"

L446-447: The following sentence seems vague to me: "As such, the external consistency analysis is most useful from vertical profiles where the in-situ platform samples the column of air above an arbitrary ground location." Instead of "above an arbitrary ground location," it would sound better to say "to an approximate altitude of > 230 m" (based on reported spiral altitudes in Table 3).

L456-457: The first sentence ("As discussed above, ... for ACTIVATE 2021-2022.") summarized the same info from L365-380, so it is redundant and should be removed.

L490: Replace "measurement" with "value". You did not technically measure gamma but calculate it from the measured scattering coefficients.

Figure 4 caption: for (a) IRI...

L525: Specify that "...properties are presented in Sect. A **at the end of the main manuscript**." Otherwise, it is hard to tell whether this Sect. A is in the SI document or somewhere else. I also think the authors can just remove the list of statistical metrics here since you already put it in Sect. A. You just need to write out the terms before using the acronym in the text.

L553: Remove "if" from "..., if the CAS data are used." Or it would be best if the authors remove the sentences on L553-554. The authors are repeating what coarse mode aerosol data to use as stated previously on L428-429.

L585: Remove "With the ground truth size distributions generated,"

Figure 5: The example size distribution is unimodal, so wouldn't it better to show an example of bimodal size distribution?

L605: "is" adjusted

Table 4: Is HSRL-2 AOD from the reported HSRL-2 data that accounts for the entire sampled column from ~9 km downward looking to the ground? The reported in-situ AODs are from Falcon's vertical profiles that mostly got up to 1-2 km based on Table 3. This is not an apple-to-apple comparison which Table 4 seems to imply. This goes back to the previous comment #6 from Reviewer 3 for the R2 version of this manuscript. I would suggest the authors to add a footnote or be explicitly clear in the text for Table 4 that there is a difference between the 2 AOD columns. Otherwise, the authors could add a "HSRL-2 equivalent AOD with in-situ profile" (or something to that effect) that only accounts for the sampled altitude performed by the Falcon (e.g., up to 1.47 km for case #9).

L684: Similar to Sect. A, specify that "...B6 and B7 in Sect. B at the end of the main manuscript."

L710: Extra space between "non-zero" and "measurements"

L731: ...there "are" still some cases

L780-793: Mention Fig. 10 somewhere in these 2 paragraphs when you describe Case 7

L792: change "scene" to "scenario"

Figure 10: Explain what the red and blue lines represent either in the caption or in the legend

Figure 13: Panels (i, k, l) need units for the x-axis

L816-817: "There also appears to be two noticeable nonspherical coarse aerosol layers within the column as evident from the spikes in LDR between 3 and 5 km" – does this refer to the vertical profile of LDR in Fig. 13c? I do not see any spike in LDR values in specified altitude range.

L819-820: "The lower layer of coarse aerosol have LR  $\approx$  35 sr. The upper layer of coarse aerosol has LR  $\approx$  45 sr, is less depolarizing, and appears to be more absorbing than the lower layer."

- + Change "have" to "has"
- + Fig. 12 panels (g) and (h) both show LR but at 2 different wavelengths, so please clarify which wavelength these sentences refer to.

L867-899: "The latter of the two findings is evident from both the profile's limited vertical extent of 1.02 km and in Table 4, which shows that the ISARA-derived AOD is only 20% of the RSP- and HSRL-2-derived AOD for this profile." – since the Falcon completed a limited spiral up to 1.02 km only, why not cut off HSRL-2 AOD value from altitude above maximum-sampled altitude by the Falcon? This way the only difference between HSRL-2 AOD and in-situ AOD would be the missing sampled altitude from the ground level to the bottom of the spiral.

L957-958: "The consistency statistics between HSRL-2- and ISARA-derived ambient LR and LDR are generally worse, relative to C\_ext and C\_ext,..." – did you mean C\_ext and C\_bsc?

L981: "These ranges are the highest if any of the comparisons within this data set" – this sentence does not make sense.

L 986: "Overall, the N comparisons are considered to be closed relatively successfully when..." – do the authors mean "...comparisons are considered to be relatively successful when..."?

Figure 15: Since the presented statistics in Table 7 and the discussion of plotted data in Figure 15 focus on the blue points (data from vertical profiles with 3+ points), I highly suggest the authors to plot the red points on Fig. 15 as greyed out points (gray points with alpha value < 0.5 if plotted with Python). Otherwise, the existing Fig. 15 panels are too busy on the eyes. The authors could also show a version of Fig. 15 with ALL red data points highlighted and the blue points grayed out in the SI document for clarity and to pair with Table S3.

+ For the blue data points, do the outliers (e.g., elevated HSRL-2 532-nm C\_bsc values in panel (e) correspond to low/high altitude bins? I.e., is there any clear bias in sampling/retrieval when you look at ALL the blue points vs. just Case 7/9 earlier? Expanding on your explanation for the relative difference between HSRL-2 and ISARA values would add value to the manuscript. As it stands, the generalized study section reads more like a list of statistical values.

L1003: Remove "the" before "ISARA"

L1049: extra word "the"

---

## Author Response (AR2)

**Response to Editor**

We thank the editor for their feedback. We have taken care to respond directly to each comment. The text from your comments are shown in black, and our responses are shown in blue. The responses include the manuscript text that was changed, removed, or added.

Thank you for your changes to the first draft of the manuscript. Two of the reviewers request major revisions, and I agree with their assessment. There is a lot of good information here, but some of the discussion is unclear and there appear to have been some decisions made that could be changed to improve the analysis. In addition to the comments of the reviewers, I have the following comments/questions:

- 1. Why do you apply the gamma parameterization (eq. 15) to extrapolate extinction to RH80 and ambient RH? The gamma parameterization is an arbitrary functional form (see, e.g., https://doi.org/10.5194/acp-16-4987-2016). Since you already have a kappa value, why not just apply the Koehler equation to calculate the growth to these other RH values? It seems like an unnecessary step to go to the gamma parameterization, and it adds uncertainty, especially at high RH values.
  - Thank you for the comment. Both  $\kappa$  and  $\gamma$  are somewhat arbitrary forms. At this point in the retrieval step, we do not have a CRI or  $\kappa$ . The reason that we do not use  $C_{\rm RH,meas(85\%)}$  directly is that the RH associated with that measurement is not reported. To reduce the uncertainty with what exact RH to prescribe, we opted to used the  $\gamma$  parameterization to fix the RH to 80% for the retrievals. We feel no changes are needed.
- 20 2. I want to confirm that you converted the values in the ICARTT-format files for size distributions and scattering and absorption values from the reported STP conditions to ambient T and P when calculating extinction for comparison with the lidar profiles. I strongly suspect you did (given the good agreement on the "unicorn" profile), but I'd like it clearly stated in the text somewhere.
- Thank you for the comment. As you suggested, we performed the adjustments to compensate for temperature and pressure. To clarify this we added text to Sect. 3.2.2 and 3.3. We added the following text to Sect. 3.2.2 "These data are provided with a standard temperature and pressure conversion factor that translates the standard number concentrations from standard temperature and pressure to ambient temperature and pressure conditions (See Sect. 3.3)". We also added the following text to Sect. 3.3: "The first step of calculating the ambient aerosol properties is to convert the number concentrations from standard temperature and pressure to ambient temperature and pressure using. After temperature and pressure adjustment, the hygroscopic adjustments are performed."
  - 3. You state that you use an extinction-weighted average across HSRL-2 altitude bins when comparing the ISARA data products with the HSRL-2 profiles. I understand why you do this for the comparison with RSP, whose signal will be weighted by extinction, but are you also doing this to compare with HSRL-2? It seems a simple unweighted mean would be appropriate for independent samples being averaged into HSRL-2 altitude bins for comparison with extinction within that bin as measured by the lidar. Please explain.
    - Thank you for the comment. We are only using extinction weighted averaging for the column-averaged data. We are using simple unweighted means for the vertically resolved data. We clarified the text as follows:

Once the desired data from the vertical profiles are identified, we select the remote sensing profile that is the nearest (horizontally) to a Falcon profile and is within 6 min of the start or stop time of the Falcon's profile. Once a viable profile collocation is identified, ISARA products are averaged to the 225 m HSRL-2 bins for the HSRL-2 comparisons or weighted by extinction and averaged through the entire profile for RSP comparisons. For the case of the HSRL-2-ISARA comparisons, this averaging results in altitude-resolved properties for a given column of air. For the RSP-ISARA comparisons, this results in a column-average property. Note that for AOD, the ISARA-derived ambient extinction is vertically integrated with sample altitude and does not account for the altitude bins above or below the sample altitudes. This allows us to estimate the amount of the total aerosol column that was sampled by the Falcon.

- 4. In several of your figures you redefine parameters (e.g., in Fig. 5 you restate the equations for number, surface, and volume size distributions). This is not necessary.
- Thank you for the comment. The redundant definitions from Fig. 5 have been removed.

5. I agree with the reviewer that additional synthetic size distributions should be evaluated. I suggest picking one that is characteristic of the marine boundary layer, one for the clean background free troposphere, one for smoke cases, and then the one you have for the mission-typical size distribution. It's helpful to choose a single size distribution observation for each of these cases, because averaging many size distributions tends to significantly broaden the standard deviation. It would also be useful to fit lognormal curves to each of these distributions and then use the lognormal fits to do the analysis—this aids in repeatability of your calculations by others if needed.

Thank you for the comment. We updated our synthetic analysis to include variability in the shape of the size distribution. We accomplished this by fitting the size-distributions that are observed in the ACTIVATE data and selecting the best fits from these to serve as the source of possible size-distributions that are available for the synthetic data generation. We updated all the sections related to the methods and results of the synthetic consistency analysis accordingly.

- 6. In your internal consistency checks you use a range of 1.51-1.55 for the RRI. The RSP retrieval gives a lower retrieved RRI. In many cases organic particles or sulfate particles might have a lower RRI. You may want to expand the range of allowed RRI to lower values.
  - Thank you for the comment. While it would be nice to investigate wider RRI ranges, it is not feasible due to the assumptions made with the particle sizing. We have used a wider range to account for some sizing uncertainties and for uncertainties in the scattering coefficient measurements. Finally, we did do a preliminary analysis to see if our measurements were sensitive to a wider range, but found that making the range wider does not improve our retrieval success rate. We feel there are no changes needed.
- 7. In Figure 7 you show excellent agreement between the ISARA-calculated and measured scattering coefficient values. It looks great! However, this is showing only the data that achieve a "match" and eliminates the  $\sim$ 40% of data for which ISARA did not provide a good match. Please plot all of the data (probably including the poor matches as grey points (you'd have to change the background color)) to show the performance of the ISARA approach when it fails as well as when it succeeds. Same comment for Fig. 8.
  - Thank you for the comment. We are not sure how to assign values as failed retrievals. The successful retrievals are an average of values within a given threshold. To do this we would need to rerun the ISARA several times with various thresholds on the successful retrieval. This method was proven successful in Sawamura et al. (2017), for example. This type of internal consistency is rarely published with any significant detail, and we feel it is outside of the scope of this work to analyze the impacts of various thresholds and optimization methods might be best. Additionally, RC4 has already indicated that this manuscript is lengthy and we would agree. Finally, the updated revision has many improvements that have increased the science value presented here. In our opinion, this optimization analysis deserves its own study. We feel there are no changes needed.
- 8. In Sect. 3.1.3 you don't really explain how the HSRL-2 and in situ data were compared. Are the profiles sloping or spiral? How many lidar shots are combined to produce the lidar profile for comparison? How are the times aligned (start to stop of the profile vs. start to stop of the lidar). This is a four-dimensional matching problem and there needs to be more detail here about how these comparisons were put together. That would help the reader understand some of the poorer agreement cases evident in Fig. 9, as opposed to the "unicorn" case.
  - Thank you for the comment. As we discuss in this section, Schlosser et al. (2024) was written to completely detail the collocation process. The following detail was added to explain the Falcon profiles better:

For this work, we limit the collocation periods to those where the Falcon performed a controlled ascent or decent. These vertical profiles are identified using the leg identifier available as part of the ACTIVATE data set (Sorooshian et al., 2023). The points of interest are classified as part of a Falcon spiral, ascent, or decent vertical profile. During the ascent and decent profiles, the Falcon would limit its airspeed to 180 m s-1 while performing a vertical profile while continuing to move in one direction. During the spiral profiles, the Falcon would spiral around a predetermined location while ascending or descending in addition to limiting its airspeed to 180 m s-1.

9. In Fig. 9, please use more of the plotting space in panels c and i. Also, the highest altitude in the profile was <4.5 km (line 723), yet the color scale goes up to 5.5 so not all of the dynamic range is used.

Thank you for the comment. We increased the used space in panels c and i. Additionally, we have removed the altitude color bar and instead binned the points based on whether or not the corresponding in-situ profile had more than 3 points.

The blue points are points where the profile had 3 or more points, and the red points correspond to the entire data set.

- 10. In the AOD comparison (Table 4), is the AOD for HSRL-2 calculated over the altitude range of the Falcon, or did you just use the archived AOD from the HSRL file? You need to describe how this comparison was calculated. What did you do with any missing in situ data from the profile (e.g., in Fig. 11 not every HSRL-2 altitude bin has a corresponding ISARA value)?
- Thank you for the comment. We removed the altitude color mapping and instead categorize the points based on whether a profile had 3 or more coincident points and color them accordingly. We updated the figure caption and associated text to explain the details. Additionally we discussed this briefly in another comment, but we do not account for missing points in the column when integrating the ISARA-derived properties. This is because we do not expect to be consistent with something like AOD and we can estimate the amount of aerosol sampled by the Falcon by comparing the resulting AOD. We have added the following text to explain this "Note that for AOD, the ISARA-derived ambient extinction is vertically integrated with sample altitude and does not account for the altitude bins above or below the sample altitudes. This allows us to estimate the amount of the total aerosol column that was sampled by the Falcon."
- 11. There are several typos floating around. a) In Table 11 Altitude is in m, not km. b) Lines 269 and 276 the size distributions are n0, not N. c) Table 1 HSRL LDR is at only 355 and 532 wavelengths (1064 is extrapolated). e) Line 300, "size-dependent". f) Line 760, "predominant".

  Thank you for the comment. Points a-c have been corrected. Point e is no longer relevant as this text was changed. Point f does not seem to be in error. Was there something about the use of predominant that is incorrect? We do not see any point d.
  - 12. Line 291 "success rate" appears here before you've explained the ISARA algorithm, so it was puzzling. Thank you for this comment. We have moved this discussion of "success rate" in to Sect 3.3.
- 13. "Stitching" is not explained (combining the LAS and SMPS size distributions).
   Thank you for pointing this out. The sentence now reads "These calibrations resulted in good stitching between the SMPS and LAS distributions, where stitching refers to the process of combining aerosol size distributions from different instruments into a continuous data set. They also led to consistent integrated number concentrations measured when compared to ancillary CPC measurements (see Figure 7 of Sorooshian et al., 2023)."
- 14. Table 2. Systematic uncertainties for size distributions are given as 10%. In diameter? In number concentration? Across all size ranges? Similar questions for the CAS and FCDP and CDP probes. I would think that counting statistics would be a source of random uncertainty in these instruments as well.

Thank you for the comment. These uncertainties are given in terms of number concentration. A random uncertainty is not provided with the data.

15. Line 372. Coarse mode particle may not contribute to concentrations, but can be absolutely important to surface area and volume.

Thank you for the comment. We agree that surface area and volume concentrations are important. We removed this sentence as it is not really accurate anyway given we are using the CAS for the measurements of coarse aerosol.

16. Fig. 1. Could you please use standard decades for x-axis labels? I don't know how to interpret the existing labels. Thank you for the comment. We changed our x-axis labels to standard decades here and in the other figures showing size-distribution.

17. Line 421. Why do you use the smallest k values that produce scattering that matches to within one percent? For RRI and CRI you use the mean of all values that match. Why the difference here? Thank you for the comment. This change would be a deviation from how this type of retrieval was done in Sawamura et al. (2017). We have already made several improvements over the previous and we would like to limit the number of changes made.

18. Line 427, Could you compare your kappa values with those derived from the AMS that I believe was on the Falcon for ACTIVATE? This is quite a low kappa for an aged, remote aerosol (e.g., https://acp.copernicus.org/articles/21/15023/ 2021/acp-21-15023-2021.pdf).

Thank you for the comment. RC4 had a very similar comment. Reiterating what was conveyed to them, there is AMS data available as part of the ACTIVATE dataset; however, that data would add even more length that they have stated is 160 already a long paper. Additionally, there is currently a separate paper analyzing the differences between the different hygroscopicity parameterizations. The convenience of the f(RH) derivation is that the retrieved hygroscopicity is internally consistent with the optical coefficient measurements.

19. when discussing biases (e.g., Fig. 6 and elsewhere), please clearly state how the bias is calculated. Is it simulated minus measured? In Fig. 6 the axis labels are unclear. Is "calculated" the synthetic data, or is that "measured"? Thank you for the comment. We have clarified the x- and y-axis labels. For clarity, the bias is always calculated as y - x. We made statements in each of the consistency analysis sections where we explain which of the properties are used as the reference (y) and which are used for comparison (x).

20. In section 3.1.2, did you consider uncertainties in the gamma value for the extinction parameterization? Thank you for the comment. We generate our dry and wet synthetic properties and add noise separately to each synthetic property. It is outside of the scope of this work to analyze the uncertainties in the  $\gamma$  and  $\kappa$  parameterizations as compared to real particle growth.

21. Table 3. Explain what "smoke counts" are. It appears elsewhere later, but is unexplained when it first appears. Thank you for the comment. This is referring to the aerosol typing product derived from the HSRL-2. To clarify this, we added "This study also uses the advanced aerosol typing product provided by the HSRL-2 team (Burton et al., 2012, 2013). Specifically, we count the number of bins flagged as smoke in the HSRL-2 aerosol typing product within each  $225 \text{ m} \times 60 \text{ s}$  bin. This HSRL-2-derived smoke count is used to analyze for the presence of elevated smoke layers." to Section 3.2. We also added "typing product" after HSRL-2 in this sentence.

- 22. Table 3. What is "platform separation"? Is this the average horizontal offset during the period of the profile? What is the range of values of separation (12 m is phenomenally close–shorter than fuselage length). How these profiles are compared needs some discussion, as noted above.
  - Thank you for the comment. We define platform separation as the horizontal separation between the center point of the Falcon's profile and King Air. We added the following clarifying text to the caption of Table 3: "the horizontal separation between the center point of the Falcon's profile and King Air (i.e., platform separation)".
- 23. Table 5. "Optical N" is a pretty arbitrary comparison since the size distribution slopes so dramatically around these sizes. If you defined optical N as > 80 or as >100 you would probably get a very different value. I'm not sure of the value of this product (I guess it goes into the calculation of sigmaext for comparison with RSP, but still seems arbitrary). Is 90 chosen because it is the LAS lower size limit?
- Thank you for the comment. Because the particle size-distributions are in terms of ambient diameters, this is not necessarily the lowest LAS bin. It is possible, it is one of the SMPS bins. That being said, the primary justification is to match with the optically active particle size-distribution that is theoretically observed in the HSRL-2 RSP measurements. This decision is explained in more detail in Schlosser et al. (2022), but that work was limited to assuming the aerosol diameters are dry.
- 24. Table 6. I really don't understand why, if Cext and Cbsc are very well correlated with the calculated values, the ratio of these is not at all correlated with the calculated values. Would you please look into this?
   Thank you for the comment. Note the comparisons in the revised manuscript are much improved, relative to the previous version. These improvements are mostly due to the consideration of a non-spherical coarse aerosol. That being said, LR consistency is still generally worse as compared to extinction and backscatter coefficients. One reason the LR is not as well correlated as the extinction and backscatter coefficients is because LR has fewer points of comparison. Additionally, there is more noise in the HSRL-2-derived LR.
- 25. Line 826, you state that under-sampling of coarse-mode particles is likely to blame for in situ/remote disagreements but never really explore this. Your size distributions go up to several microns behind the inlet, and the cloud probes are quite capable of measuring from a couple of μm and larger. Do you have evidence that the cloud probes are undersampling? What is the basis for this statement? It's pretty important if our in situ techniques are not good enough to evaluate remote sensing methods. Why is the unicorn case so good if we can't measure coarse particles well? Might it be that the other profiles are so short that it's hard to match things up in less than ideal circumstances? This would argue that closure studies need to be performed extremely carefully only in ideal circumstances.
- Thank you for the comment. We do agree that achieving spatiotemporal collocation is challenging. In our revisions, we were also able to improve our disagreement between the ISARA- and the remote sensing-derived properties. Still, we think that undersampling of the number concentration of aerosol with diameter greater than 5 μm (N>5 μm) under background conditions (<1 cm-3) is still a limitation of the in-situ instruments. For clarity, we added the following to the introduction: "The measurements of number concentration are less accurate for background concentrations (N < 1 cm-3), which is common for aerosol with diameters > 5 μm (Baumgardner et al., 2001).".
  - 26. I agree that the case study "unicorn case" should be presented first, to show how the comparisons are done in an ideal situation. Then you can show the results for all the profiles together.

    Thank you for the comment. We changed the order in which these sections are presented. We added a second case study
- 225 that is presented in the case study section (Sect. 4.3)

**Response to Reviewer 2**

We thank the reviewer for their feedback. We have taken care to respond directly to each comment. The text from your comments are shown in black, and our responses are shown in blue. The responses include the manuscript text that was changed, removed, or added.

I consider the revised version of the manuscript ready for publication, pending a final check for any remaining typographical errors. I would also encourage the authors to reconsider the subheadings in the Methods section.

Thank you for the comment. We appreciate your effort in reviewing our revised manuscript. We made further improvements in this revision that were guided by the comments from the other reviewers and the editor. We feel the new version has even more scientific value.

**Response to Reviewer 3**

- We thank the reviewer for their feedback. We taken care to respond directly to each comment. The text from your comments are shown in black, and our responses are shown in blue. The responses include the manuscript text that was changed, removed, or added.
- I feel that the authors have been through a large effort towards focusing more on the science of their paper than the previous version. They have taken an open mind and they have taken most comments into account. Congratulations because I can see your big efforts!

The starting point was however quite far from acceptable (if you remember my previous review). The authors have covered a great deal of the distance needed, but I feel that there are still some MAJOR points that need addressing, hence at this stage I recommend a MAJOR REVISION.

Thank you for your continued efforts in reviewing this manuscript. Your comments have been integral to the marked improvements in the consistency results and further increased the scientific value of this work. In its newly revised version, we feel we have fully addressed each of your major and minor points.

**The MAJOR POINTS of concern are highlighted below:**

- 1. The synthetic consistency analysis is a bit limited because the authors have chosen to keep a fixed particle size-distribution, varying only the total number (see lines 476-477 and 578-580). This prevents a true synthetic consistency check, which should include exploring different PSDs, different mixes between larger and smaller particles, etc. I feel that this could easily be addressed and that there are no justifiable reasons for this self-limitation. I therefore invite the authors into exploring different dominant particle sizes using modelled monomodal and bimodal PSDs.

  Thank you for the comment. We updated our synthetic analysis to include variability in the shape of the size distribution. We accomplished this by fitting the size-distributions that are observed in the ACTIVATE data and selecting the best fits from these to serve as the source of possible size-distributions that are available for the synthetic data generation. We updated all the sections related to the methods and results of the synthetic consistency analysis accordingly.
- 2. The external consistency presented in section 3.1.3 is not fully convincing. In particular, figure 9 shows a low bias and I feel that the authors are too simplistically trying to dismiss this bias (lines 676-677; 699-701) as not being something to discuss more in-depth. The fact is that the data in Figures 9a,b,d,e show that the data are not near the 1:1 line (as they should be) and they follow a different dependency. This point had been raised already in the forst review (points 6 and 8).
  - Thank you for the comment. In the revised version of the manuscript, we show marked improvements to the results. The majority of improvements are related to our treatment of nonspherical coarse aerosol. We updated all of the methods, results, and conclusions to reflect these improvements and adjustments to our methods.
- 3. Similarly for the LR dependency, which seems not to be responding to the microphysical parameters (line 682) but this does not seem to be evaluated critically.

  Thank you for the comment. In our revised manuscript we made significant improvements in the LR consistency. These
  - Thank you for the comment. In our revised manuscript we made significant improvements in the LR consistency. These improvements stem from adding using two distinct spheroid approximations for the coarse aerosol properties. These improvements are reflected throughout the methods, results and conclusions. We still agree that LR consistency has more room for improvements even over our best efforts. There is a functional limitation in the available shape we can optically represent. Additionally, improved measurements of coarse aerosol composition and optical properties could help.
- 4. The case study (figures 11 and 12) shows a lot of promise, and it is much more convincing than the external consistency analysis in section 3.1.3. I suggest to present the case study before the full dataset so as to show what is working and convincing before showing something a little more problematic.

Thank you for the comment. We reorganized the sections to accommodate this comment.

- 5. The authors have tried to avoid my question about VLDR and PLDR (first review, point 25, and line 524 of the revised manuscript). However it is quite an important point because the chosen threshold of 13% need to be understood by the reader and the reviewer. Until this is clarified, it is not possible for me to understand this choice fully. I am under the impression that the authors maybe do not know the distinction between VLDR and PLDR, and have therefore not fully understood the question: to advance on this point, I invite them to discuss it with their lidar experts.
- Thank you for the comment. We reviewed our analysis presented in the previous work and found the we were indeed using the volume LDR instead of the aerosol LDR. This realization significantly changed our view on the HSRL-2 data as it made us realize that even our unicorn case had measurable presence of nonspherical aerosol as evident from the HSRL-2-derived 532 nm LDR which was as high as 0.18. This provided us with a unique opportunity to explore nonspherical aerosol in our analysis. While our fine aerosol are still constrained to spherical approximations, we use the HSRL-2-derived LDR and LR to inform our assumptions on the coarse aerosol. We have updated the entirety of the text to explain the added nonspherical information. This has led to a marked improvement in the  $C_{\rm ext}$ ,  $C_{\rm bsc}$ , and LR consistency. Additionally it has opened the discussion to LDR, which has been added throughout the text. Additionally we now specify that the LDR in question is the aerosol LDR in Table 1.
- 6. The text at lines 752-753 raises an important question: how doe the authors account for layers that are not sampled in-situ? Perhaps, using the HSRL-2 data a rigorous way to account this is possible. In any case, this needs discussion. As the goal of this work is to establish consistency rather than complete closure, we do not expect to retrieve AOD. We added the following text for clarity "Note that for AOD, the ISARA-derived ambient extinction is vertically integrated with sample altitude and does not account for the altitude bins above or below the sample altitudes. This allows us to estimate the amount of the total aerosol column that was sampled by the Falcon."
  - 7. There are several minor points of revision that I highlight in the attached annotated manuscript. Although they are minor, their number is large, hence they overall constitute a major point of concern.

    Thank you for the comment. Your minor points have been restated below along with our response to each.
- 315 Line 1: <del>particle</del>

Thank you for the comment. An aerosol is a suspension of solid particles or liquid droplets in air or another gas. While it is common to use aerosol to refer to just the particles, we feel this needs to be made clear within the context of our study. We initially refer to the particles as "aerosol particles" to provide context to our study. However, we removed "particles" in the subsequent text.

- Line 2: impacts effects
   Thank you for the comment. We changed "impacts" to "effects".
- Line 5: typically
   Thank you for the comment. We removed the word "typically".
- Line 7: software framework methodological framework
   Thank you for the comment. We changed "software" to "methodological".

- Line 11: measurement in-situ and remote sensing Thank you for the comment. We replaced "measurement" with "in-situ and remote sensing". - Line 11: "...ISARA-calculated aerosol data..." I would suggest to be explicit and indicate the list of properties that ISARA calculates. Thank you for the comment. We feel this level of detail would be difficult to capture succinctly in the abstract and 335 would be distracting from the findings presented. We added the text "intensive and extensive" to add more generic - Line 12: data properties 340 Thank you for the comment. We replaced "data" with "properties". - Line 18: Move aerosol to be written before imaginary refractive index. Thank you for the comment. We made this change. - Line 18: "...probable under-sampling..." explain why. 345 Thank you for the comment. We attribute this to low aerosol concentrations ( $< 1 \text{ cm}^{-3}$ ) for aerosol larger than 5 µm that are measured throughout the mission. We added the following explanation to the abstract: "low background concentrations ( $N < 1 \text{ cm}^{-3}$ ) of coarse aerosol sizes greater than 5 µm.". We also added the number concentration of aerosol with diameter greater than  $5 \mu m (N_{>5 \mu m})$  information for each of the cases to Table 4. 350 - Line 20: overall Thank you for the comment. We removed "overall" from the text. Line 20: has could have Thank you for the comment. We replaced "has" with "could have". 355 - Line 23: "central", Line 25: "central", Line 27: "key", Line 27: "critical", Line 35: "critical", there is an inflation of emphasizing words (central, key, critical, etc.). I think that modesty dictates that you should have one emphasizing word at most (or even none?). Usually understatement raises interest in your work, whereas emphasis very seldom 360 persuades readers. Thank you for the comment. We removed the emphasizing words on lines 25, 27, and 35, but have kept "central" on line 23. - Line 44: "Tsekeri et al., 2017". In the response to my review (point 2 of response to reviewer 3) you have given an explanation of the differences between the work by Tsekeri and yours. I feel that the paper would benefit from 365 having these comments in its introduction instead, as they put the reader into the wider context of different possible approaches and their pros and cons. Thank you for the comment. We moved the following text from the conclusion to the introduction:

wing probe measurements.

tioning these high errors.

This work extends the methods established by Sawamura et al. (2017) by attempting to account for coarse aerosol in the bulk aerosol properties after retrieving the fine CRI and  $\kappa$ . Similar to the work of Tsekeri et al. (2017), we account for the contribution of coarse (ambient aerosol diameter > 1.0  $\mu$ m) particles using

- Line 66: "these measurements". I suppose you want to say "the humidification corrections" or "the humidification curves"? in any case "these measurements" is unclear and does not indicate to what you want to refer when men-

Thank you for the comment. The text "these measurements" is referring to the ambient RH measurements. The text is now "these RH measurements" for clarity.

- Line 66: "15%". I would not regard 15% as a very large error for the humidity-corrected properties. Are you sure 380 that it is not much bigger?

> Thank you for the comment. While it may seem trivial, a 15% error in ambient RH means that a measured RH of 80% can be as low as 68% or as high as 92%. We feel no changes are needed given this can be a significant source of error that propagates through to calculating the properties of humidified aerosol.

- Line 77: placed behind inlets Thank you for the comment. This text has been added.

- Line: 78: "typical". I would not regard 5 um as "typical" unless (like here) there is an isokinetic inlet. Regular inlets could have a much smaller cutoff diameter.

Thank you for the comment. We removed "typical" from this sentence.

- Line 79: open-path. Thank you for the comment. We added "open-path" to the text.

- Lines 79-80: "sizing can be highly uncertain". please explain better why you think that wing-mounted probes are more uncertain than probes behind an inlet?

Thank you for the comment. We agree that they are at least similar to the errors derived from the measurements behind the inlet. We reworded this text as follows:

Wing-mounted open-path probes are commonly used to estimate coarse aerosol properties, but are designed in such a way that the sizing can be uncertain (e.g., Reid et al., 2003, 2006). The measurements of number concentration are less accurate for background concentrations ( $N < 1 \text{ cm}^{-3}$ ) of aerosols with D > 5 µm (Baumgardner et al., 2001). In addition to the coarse sampling limitations, particles are lost between the external inlet of the aircraft and the inlets of the instruments (Baron and Willeke, 2001; Kulkarni et al., 2011).

- Lines 80-81: "In addition to the coarse sampling limitations, particles are lost between the external inlet of the aircraft and the inlets of the instruments (Baron and Willeke, 2001; Kulkarni et al., 2011).". Move this sentence to line 75.

Thank you for the comment. We moved this line to the start of the paragraph. We also reworded it it it as follows: "In addition to the measurement limitations in measuring relative humidity and hygroscopicity, particles are lost between the external inlet of the aircraft and the inlets of the instruments (Baron and Willeke, 2001; Kulkarni et al., 2011).".

- Line 86: Python based. This is scientifically irrelevant. You may want to add the technical details (programming language, Mie scattering library used) at the end, otherwise they risk to shadow your scientific work and make it appear like a programming effort.

Thank you for the comment. We removed this "Python Based" and moved this information to the Code Availability section.

- Lines 88-89: the Fortran-based Modeled Optical Properties of Ensembles of Aerosol Particles package (MOPSMAP; Gasteiger and Wiegner, 2018) a Mie scattering library. Thank you for the comment. We reworded the text to be as follows "Specifically, the algorithm uses a Mie scattering

library (e.g., Mishchenko et al., 2002; Bohren and Huffman, 2008) in conjunction with measured size distributions and optical coefficients to retrieve refractive indices and hygroscopicity from a "common" suite of in-situ instruments.". We move the information about MOPSMAP to the Code Availability Section.

Line 97: "dry". The same applies to wet sea salt.
 Thank you for the comment. We removed "dry" from this sentence.

Line 97: "are difficult". yes they have a larger size, but this does not make these particles "difficult to consider"!
 Thank you for the comment. We agree this was poorly constructed. We reworded these sentences for clarity:

Common coarse species that can have diameters >  $1.0\,\mu m$  are sea salt and dust (Hussein et al., 2005). As discussed above, the larger (D >  $5\,\mu m$ ) sizes of these coarse species are difficult to measure in background concentrations, however each species also poses a unique challenge from an optical perspective. Dry sea salt is non-spherical, non-absorbing, and very hygroscopic, which translates to larger values of  $\kappa$  (Sorribas et al., 2015; Moosmüller and Sorensen, 2018; Ferrare et al., 2023). Similar to sea salt, dust can also be non-spherical. In contrast to sea salt, dust can be moderately absorbing and has a complex refractive index (CRI) that is dependent on wavelength (Voshchinnikov and Farafonov, 1993; Veselovskii et al., 2010; Wagner et al., 2012; Sorribas et al., 2015).

Line 101: overcome these limitations by using use
 Thank you for the comment. We replaced "overcome these limitations by using" with "uses" as "use" would not have been grammatically correct.

Line 102: open-path
 Thank you for the comment. We added the text "open-path".

- Line 102: sample coarse-mode permit Thank you for the comment. We removed "sample coarse-mode" and reworded the sentence as follows: "This work uses wing-mounted open-path probes to estimate the contribution of the coarse aerosol on the calculated ambient optical and microphysical data (Ryder et al., 2015; Tsekeri et al., 2017; Ryder et al., 2018)".

Line 103: , which is common for studies looking to account for coarse-mode
 Thank you for the comment. We removed the text ", which is common for studies looking to account for coarse-mode".

Line 124: "the coarse is assumed to be sea salt". specify: dry or wet sea salt?
 Thank you for the comment. We added the specification of "humidified sea salt".

Lines 131-132: "limited ability of in-situ instruments to sample coarse particles". I do not agree with the stated "limited ability". Open-path instruments such as the ones that you use permit to sample the coarse mode.
 Thank you for the comment. Please see our response to your comment regarding Line 18. We feel we have addressed this concern with the added information regarding low concentration of N>5 turn.

- Line 152: (liquid)

Thank you for the comment. We feel this is an incorrect relation. Sulfate particles are well approximated as spheres in both solid and liquid states, for example. Also, dust is not always approximated as a sphere even when humidified. No changes are made as a result.

- Line 152: (solid)
   Thank you for the comment. We addressed this in the previous comment. No changes are made as a result.
- Line 160: composition complex refractive index
   Thank you for the comment. We replaced "composition" with "CRI".
- Lines 169, 170, 173 and 187: extrinsic extensive
   Thank you for the comment. We replaced the word "extrinsic" with "extensive" here and throughout the remaining text.
  - Lines 170, 172, 173, and 188: intrinsic intensive
     Thank you for the comment. We replaced the word "intrinsic" with "intensive" here and throughout the remaining text.
  - Line 174: therefore using any Mie scattering library would produce equivalents results
     Thank you for the comment. We added the text "; therefore, using any Mie scattering library would produce equivalent results.".
  - Line 186: where dN is the number of particles counted in a bin
     Thank you for the comment. We added the text. "where dN is the number of particles counted in a bin.".
- Lines 193-194: "For particles larger than the remote sensor's observing wavelength, the remote sensor is most sensitive to particle cross-sectional area (i.e., surface area concentration). For particles smaller than the observing wavelength, the remote sensor is most sensitive to volume concentration.". This statement is not correct. The cross-section is Q \* A where Q = extinction efficiency and A is the area. Your statement is equivalent to saying that for small particles Q ~ R, but in fact typical Q curves are not like this. You can check it with your Mie code, or you can google it. For example: https://www.researchgate.net/profile/Carynelisa-Haspel/publication/26454094/figure/fig3/AS:669075336359945@1536531399888/Theextinction-efficiency-Q-ext-as-a-function-of-size-parameter-x-of-polystyrene. png This is the same as comment number 17 of my first review, not satisfacorily addressed.
   We rewrote this sentence for clarity and relevance to the paper as follows: "For remote sensing of fine- and coarse aerosol particles, polarimeters and high-spectral-resolution lidar that operate in the visible to near-infrared part of the spectrum are sensitive to the aerosol effective radius, effective variance, real refractive index, and single-scattering albedo (Hansen and Travis, 1974; Burton et al., 2016; Stamnes et al., 2018)."
  - Line 204: campaign

- If we were to write the acronym out this sentence would read "The Aerosol Cloud meTeorology Interactions oVer the western ATlantic Experiment campaign...". The extra word "campaign" is redundant with the word experiment in the ACTIVATE acronym. We followed the examples of previous studies published related to ACTIVATE (e.g., Sorooshian et al., 2023, 2019).
- Line 212: "During the first five and a half ACTIVATE deployments". move the description of the two aircraft used to before this text, and indicate their respective role (in-situ / remote sensing)
   Thank you for the comment. We moved the next paragraph to be before this paragraph. This paragraph does state explicitly what roles each aircraft is serving. Specifically, the text states "..lower-flying HU-25 Falcon aircraft collected in-situ data..." and "...higher-flying King Air at approximately 9 km would conduct remote sensing and launch dropsondes...".

- Line 269: "N". this should be dN and not N, given that N is the total number of particles regardless of their size bin (not size-resolved)

Thank you for the comment. We addressed this by replacing N with  $n^{\circ}$ . The text is now as follows: "In-situ measurements of dry logarithmic size-resolved aerosol number concentration ( $n^{\circ}$ )...".

- Line 277: "N". dN Thank you for the comment. We changed N to be  $n^{\circ}$ .

- Lines 284-285: "The impact on the absorption coefficient from particles above 1 µm is assumed to be negligible in the calculation of extinction coefficients." You have the tools to easily calculate it (the Mie scattering library), therefore why this assumption?

We make this assumption to avoid having to apply two different size distributions during the ISARA CRI retrieval. We would need to use the entire distribution up to  $3.49\,\mu m$  one for the absorption coefficient, which is not behind the cyclone, and the truncated size distribution for the scattering coefficient, which is behind the cyclone. In reality these are only particles in the 2.24 to  $3.49\,\mu m$  range which have size parameters that fall in the maximally scattering range observed in Moosmüller and Sorensen (2018). We updated this sentence to read as follows: "The impact on the absorption coefficient from particles between 2.25 and  $3.49\,\mu m$  is negligible in the calculation of absorption coefficients (Moosmüller et al., 2009; Moosmüller and Sorensen, 2018). This assumption may result in an overestimation in the IRI in some cases, which is important to consider."

Lines 286-292:

It is important to note that the optical particle size can be greater than the aerodynamic size by a factor of 1.2 to 1.8 depending on the particle density and shape. Additionally, the cyclone has a 50% efficiency at 1  $\mu$ m. Due to the difference between the aerodynamic and optical particle sizes and the imperfect nature of the cyclone, we use a maximum cutoff diameter of 2  $\mu$ m for the upper bound of the LAS size distribution. To further motivate this decision, using lower thresholds of 1.5 and 1.8  $\mu$ m resulted in a success rate of 33% and 10%, respectively. There is also a decrease in the internal consistency as all measures of bias increase when a cutoff of less than 2  $\mu$ m is used.

These six lines are completely unclear to me.

We updated this to read as follows:

For the cyclone, it is important to note that the optical size can be greater than the aerodynamic size by a factor of 1.2 to 1.8 depending on the density and shape. Additionally, the cyclone has a 50% efficiency at 1  $\mu$ m. Due to the difference between the aerodynamic and optical sizes and the imperfect nature of the cyclone, we select the LAS bin that has a  $D_{\rm gm}$  of 2  $\mu$ m as the last bin of size distribution. This bin samples aerosol with diameters ranging from 1.79 to 2.25  $\mu$ m. While this bin extends to aerosol sizes as high as 2.25  $\mu$ m this bin was chosen to limit the erroneous over-truncation of the size distribution. This decision is discussed in more detail in Sect. 3.3. The impact on the absorption coefficient from aerosol between 2.25 and 3.49  $\mu$ m is negligible in the calculation of absorption coefficients (Moosmüller et al., 2009; Moosmüller and Sorensen, 2018).

- Line 317: and  $C_{scat}$ , measured Thank you for the comment. We added "and  $C_{scat,measured}$  are".
- Line 320: "0 Mm-1". can the signal be less than zero? please explain how this is possible Thank you for the comment. The measurement uncertainty can lead to the in-situ measurements having signals less than zero, which happens more at lower signals. We added the following text for clarification: "Note that the measurement uncertainty can lead to the in-situ measurements having signals less than zero. This is most important

**- Lines 321-331:**

Measurements of ambient liquid water content (LWC) and cloud drop number concentration ( $N_{\rm d}$ ) are used to classify in-situ data as cloud-free, ambiguous, or cloud. This classification becomes important because ISARA retrievals are performed for cloud-free cases. Ambient LWC and  $N_{\rm d}$  are both derived from ambient particle size distribution measured by a Cloud and Aerosol Spectrometer (Droplet Measurement Technologies CAS; Baumgardner et al., 2001; Lance, 2012), a Fast Cloud Droplet Probe (SPEC FCDP; Kirschler et al., 2022), and a Cloud Droplet Probe (Droplet Measurement Technologies CDP; Sinclair et al., 2019). The CAS, CDP, and FCDP measure particles in the ambient D size ranges of 0.5–50  $\mu$ m, 2–50  $\mu$ m, and 3–50  $\mu$ m, respectively. Measurements are considered cloud-free where LWC is less than 0.001 g m-3, respectively (Schlosser et al., 2022). Because the CAS, CDP, and FCDP provide redundant measurements of LWC, this work relies on the CDP primarily and only uses FCDP for flights where the CDP was not being used. The CAS is only used as a backup for LWC in the case the CDP and the FCDP are unavailable.

What about clouds that have an IWC instead of LWC, and which have particles larger than 50 um (as is usual for ice clouds)? how are you excluding those? perhaps you limit the measurements to T > 0? Please state in the paper how you avoid ice clouds.

Thank you for the comment. Ice is not a concern for two reasons. First, we limit the maximum aerosol diameter of the size distribution to  $20\,\mu m$  so these data will not influence our coarse aerosol measurement. Second, the ice particles would not survive isokinetic sampling. The following text has been added accordingly: "The sampling inlet flag also filters out ice clouds. Filtering out stray ice aerosol is not required as they will not survive the isokinetic sampling process."

- Line 345: "shortcoming". why be so negative? you could say that it impacts the horizontal resolution without expressing a judgement
  - Thank you for the comment. This has been rephrased and the text is now as follows: "results in a spatial resolution of 8 km, which assumes a ground speed of  $180 \text{ m s}^{-1}$ ".
- Line 347: "vertical profiles where the in-situ platform samples the column of air above an arbitrary ground point".
   I doubt that the aircraft does "vertical profiles above an arbitrary ground point". Typically they will be slant profiles or spirals. Please clarify

Thank you for the comment. While we feel this information would be distracting here, We added the following text to Sect. 3.5:

For these comparisons, the ISARA-derived products are acquired where the Falcon performed a controlled ascent or decent are identified using the leg identifier available as part of the ACTIVATE data set (Sorooshian et al., 2023). The points of interest are classified as part of a Falcon spiral, ascent, or decent. During the ascent and decent profiles, the Falcon would limit its airspeed to  $180\,\mathrm{m\,s^{-1}}$  while performing a vertical profile while continuing to move in one direction. During the spiral profiles, the Falcon would spiral around a predetermined location while ascending or descending in addition to limiting its airspeed to  $180\,\mathrm{m\,s^{-1}}$ .

Additionally We added "Further details on the Falcon's vertical profile are discussed in Sect. 3.5" after the text associated with this comment.

- Line 368: "(8)". this equation is not needed (it is the same as equation 3) Thank you for the comment. We removed the redundant equations.

Line 379: ehannel bin
 Thank you for the comment. We replaced "channel" with "bin".

- Line 380: "the average of all valid CRI values". why not a weighted average, with the weight inversely proportional to the difference between C\_calc and C\_RH=40?

Thank you for the comment. Using a weighted average in this way assumes that the measured value has no measurement error. Additionally, we are limiting the number of changes made from Sawamura et al. (2017), which is the basis of this work. We added the following explanation to the text: ", which was shown to be effective for previous studies that this work expands upon (e.g., Sawamura et al., 2017)".

Line 390: "(9)". equation not needed (like for equation 8)
 Thank you for the comment. See the above response to the comment on line 368. No actions have been taken.

Line 407: RRI CRI
 Thank you for the comment. We replaced "RRI" with "CRI".

- Line 421: "the smallest  $\kappa$  values are taken". why the smallest? we need (1) to know the reason for taking the smallest, and (2) to be wary of the fact that the minimum and the maximum can be prone to outliers

Thank you for the comment. As discussed in the comment regarding CRI averaging, we are trying to limit the number of changes made. Currently we are focusing on the consideration of coarse-aerosol and providing the foundation for future work. In response to your second point, there are several levels of averaging due to the retrieval step needing to be performed at the SMPS resolution. As such, we feel the impacts of outliers have already been limited. No actions have been taken.

Line 425: fine-mode
 Thank you for the comment. We added the text "fine-mode".

Line 427: "low hygroscopicity". how does this low hygroscopy match the assumption that they are sulphates?
 Thank you for the comment. We agree this is not very representative of pure sulfate aerosol species but common for mixtures of organic aerosol species with sulfate aerosol species. We changed the text as follows to capture this information:

The commonly observed low absorption is expected given the frequency of sulfate and secondary organic aerosol Nakayama et al. (2015). That being said, the observed IRI is between 0.01 and 0.08 in 32% of the data, which indicates the presence of moderately absorbing aerosol species such as aged smoke and dust. The observed low hygroscopicity in many of the retrievals also indicate organic aerosol species are present (Petters and Kreidenweis, 2007). Only 20% of the data had  $\kappa$  >0.2, which is the upper limit of the range of  $\kappa$  for organic aerosol species (Massoli et al., 2010).

- Lines 429-432:

The ISARA-derived IRI and  $\kappa$  are combined with the measured ambient RH and dry size distribution data and are used to calculate ambient scattering and absorption coefficients ( $C_{\rm scat,amb}$  and  $C_{\rm abs,amb}$ , respectively) for the total (e.g., bulk) particle size-distribution ( $0.003 \leq D \leq 20\,\mu m$ ), the fine particle size-range ( $0.1 \leq D \leq 1\,\mu m$ ), the coarse particle size-range ( $1 \leq D \leq 20\,\mu m$ ), and the optically active particle size-range ( $0.1 \leq D \leq 20\,\mu m$ ).

Move the text at lines 455-465 to here so that we can understand how you account for the coarse mode Thank you for the comment. We left this paragraph where it is, but it has shifted up to just after lines 429-432 as a result of moving the ambient aerosol equations to the appendix.

- Line 434: "(16)". equation not needed like for eq. 8
   Thank you for the comment. See the above response to the comment on line 368. No actions have been taken.
- Line 437: "the following equations". put the equations on this page in a table
   Thank you for the comment. We feel that a table is not ideal for displaying equations, but we can understand how this series of equations is distracting here. We added these to the appendix (Section A).
- Lines 476-477: "the shape of the size distribution is fixed and only the total number concentration is allowed to vary.". this is very limited. I suggest to take a different approach, using e.g. a lognormal where you can vary the number of particles and their geometric mean radius, or even bimodals resulting from the sum of two lognormals. I feel that the approach taken here is too simple and therefore not fully persuading.
   Thank you for the comment. We updated our synthetic analysis to include variability in the shape of the size distribution. We accomplished this by fitting the size-distributions that are observed in the ACTIVATE data and selecting.

bution. We accomplished this by fitting the size-distributions that are observed in the ACTIVATE data and selecting the best fits from these to serve as the source of possible size-distributions that are available for the synthetic data generation. We updated all the sections related to the methods and results of the synthetic consistency analysis accordingly. We have extended the synthetic data analysis to include variability in the aerosol size distribution. We have extensively updated Sections 3.4 and 4.1 to reflect these improvements.

- Line 506: "note that 26.49% synthetically-generated measurements did not fall within appropriate delta thresholds". explain why this happens
Thank you for the comment. This is directly as a result of measurement noise. We updated the subsequent sentence as follows for clarity: "To demonstrate the functionality of this analysis, the synthetic data generation and retrieval processes were repeated with zero measurement noise, which results in a rate of successful retrievals of 100%. Synthetic consistency analysis can be extended further to include nonspherical aerosols, aerosols, aerosols, aerosols, aerosols, aerosols.

retrieval processes were repeated with zero measurement noise, which results in a rate of successful retrievals of 100%. Synthetic consistency analysis can be extended further to include nonspherical aerosols, aerosols without a constrained RRI, and increasing the number of successful retrievals under higher noise and lower signal conditions (e.g., lower aerosol concentrations, weakly scattering or weakly absorbing aerosol).".

- Line 524: "LDR". Specify if this is volume depolarisation ratio (a property of the atmospheric volume comprising molecules and particles) or particle depolarisation ratio (a property of the particles only). This is quite important to understand your quantitative threshold at 13%, because these quantities can be quite different. If the lidar data are simply labelled "LDR" then please contact the scientists that work with that instrument to answer the question, because we need to know to which of the two depolarisation ratios you refer. This is the same as comment number 25 of my first review, not correctly addressed.

Thank you for the comment. We addressed this topic in response to your major point 5 above. Most important to note again here is that we have clarified that we are using the aerosol LDR.

Line 532: "the ISARA products data are weighted by extinction". Please clarify if this is a vertical integration giving columnar quantities. In the present formulation, this is unclear.

Thank you for the comment. Your understanding is correct. We clarified this in the text as follows:

For the case of the HSRL-2-ISARA comparisons, this averaging results in altitude resolved properties through for a given column of air. For the RSP-ISARA comparisons, this results in a column-average property. Note that for AOD, the ISARA-derived ambient extinction is vertically integrated with sample altitude and does not account for the altitude bins above or below the sample altitudes.

- Line 545: "(30)". Usually the standard deviation has (n\_p 1) at the denominator, to denote the fact that the mean is not indipendent from the data. It is my impression that this should be the correct approach here too. Please see: https://en.wikipedia.org/wiki/Bessel%27s\_correction
  - Thank you for the comment. We added the missing "- 1" from the standard deviation equations to use Bessel's correction. There have been some updates to the standard deviations presented, but no changes were significant enough to change the overall results.

- Line 562: "range-normalized root-mean square deviation". Again you are using in this formula the extremes of a variables (max and min) which are prone to be influenced by outliers. I encourage you to explore a metric that is more robust, e.g. using percentiles. On using the word "range" please note that in remote sensing this indicates the distance from the instrument (e.g. the variable R in the lidar equation). On the other hand in statistics this indicates the difference between max and min. So in this case, to avoid ambiguity, perhaps you could omir the word "range" here.

We updated this text to "normalized root-mean square deviation".

- Lines 579-580: "Future work could explore the impacts of adjusting the synthetic size distribution creation process to analyze the impact of low total concentration conditions.". As already highlighted, I encourage the authors into doing a proper study of the synthetic consistency, making use of the full variability of inputs and using lognormals. Thank you for the comment. As discussed in two previous responses, we have extended the synthetic data analysis to include variability in the aerosol size distribution. We have extensively updated Sections 3.4 and 4.1 to reflect these improvements.
- Line 587: "-0.9". -0.9 is a very large bias on the IRI which is considered in this paper < 0.08. But if this is a relative bias then I'd like to see a % symbol immediately after the "0.9"</li>
   Thank you for the comment. To clarify, this was written in terms of percent. To ensure this is clear in the text, we reworked this notation throughout the text. We no longer use the ± notation except when discussing the standard error of the mean in the statistical analysis.
- Figure 6: "ISARA-retrieved versus synthetic". use these terms for the graph axes instead of measured and calculated, which are less clear
   Thank you for the comment. We replaced "calculated" with "ISARA-derived" and "measured" with "synthetic" in this figure.
- Line 609: points that had the successful retrieval of CRI and 12319. given that there is the same number of points, 12319, there is no need for a repetition
   Thank you for the comment. This was an error on our part. The values here have changed slightly as we adjusted how we adjust the dry scattering measurements. We updated the numbers in the text..
- Line 610 "successful retrieval". It is unclear what is a successful retrieval. A retrieval can be successful in terms of identifying a solution (the algorithm converges) or because the solution is close enough to the observations (which is more stringent). Please clarify. This is the same as comment 26 of my first review, not satisfactorily addressed. Thank you for the comment. In this work we define a successful retrieval as one that is close enough to the in-situ observations. We have gone through the text and clarified what a successful retrieval is defined as. Text has been added in the retrieval section and associated with this comment at line 610. The following text has been updated accordingly:

A final check is performed to establish a successful CRI retrieval. Here we ensure  $\overline{CRI}$  results in scattering and absorption coefficients that meet the same thresholds of 20% and 1 Mm-1, respectively.

"In this step, a successful  $\kappa$  retrieval is where a  $\kappa$  was found with a corresponding  $\Delta C_{\rm scat, wet} < 1\%$ ." and "i.e., a solution was produced within the required retrieval thresholds".

- Line 613: also
   Thank you for the comment. We added "also".
- Line 617: "of NRMSD". add the reference to figure 7 here
   Thank you for the comment. We added " (see Figure 8)".

 Line 621: "this bias increases with increasing wavelength,". from the figure it looks like it increases with DE-CREASING wavelength

Thank you for the comment. We agree this statement is incorrect and have removed it.

- Figure 7: "Figure". use "ISARA-calculated" as x-axis label
  Thank you for the comment. We replaced "calculated" with "ISARA-derived" in the figure x-axis label.
- Figure 8: "Figure". 1) use "ISARA-calculated" as x-axis label and (2) pay attention to the location of the figure titles. At the moment, "f(RH)" looks like the x-axis label for the top plot, whereas it is the title of the bottom plot. Thank you for the comment. We replaced "calculated" with "ISARA-derived" in the figure x-axis label.
- Table 3: "Table". add the location of each profile in the table. Could be a description such as "ocean, 1000 km East of North Carolina"
   Thank you for the comment. We have added the midpoint each of the Falcon's profiles to this table. With the addition of Fig. 10, this should provide the information requested without adding more information to an already large table.
- Table 3: "number of smoke counts". explain what this is
   Thank you for the comment. This is referring to the aerosol typing product derived from the HSRL-2. To clarify this, we added "This study also uses the advanced aerosol typing product provided by the HSRL-2 team (Burton et al., 2012, 2013). Specifically, we count the number of bins flagged as smoke in the HSRL-2 aerosol typing product within each 225 m × 60 s bin. This HSRL-2-derived smoke count is used to analyze for the presence of elevated smoke layers." to Section 3.2. We also added "typing product" after HSRL-2 in this sentence.
  - Table 3: "the RSP-derived total, fine-mode, and coarse AOD, the HSRL-2-derived AOD, in-situ-derived AOD," this description of Table 3 is incorrect
     Thank you for catching this error. We replaced "the RSP-derived total, fine-mode, and coarse AOD, the HSRL-2-derived AOD, in-situ-derived AOD," with "HSRL-2-derived LDR".
  - Line 661: "Data from the 49 profiles". I am getting confused now. It is unclear if you use 49 or 10 profiles. Please make this clear in the text.

    Thank you for the comment. The method here has changed from the previous revision. Our generalized study, which has changed in title from the statistical analysis, focuses on the consistency within the subset of 10 profiles. We also compare those consistency statistics to that observed in the entire set of 49 profiles. The text here is moved to later in the text and changed to: "In this section we focus our analysis on the data from the 10 vertical profiles of altitude-resolved ambient  $C_{\rm ext}$ ,  $C_{\rm bsc}$ , LDR, LR and N that had more than 3 points of comparison. For this section we have applied the *a priori* assumption to the coarse aerosol as described in Sections 3.5 and 4.3. Additionally, we compare the consistency results from this subset of 10 profile to the entire set of 49 profiles.".
  - Line 670: "These MRB indicate that the in-situ data data is biased low". This low bias is very clearly seen in figures 9a,b,d,e and I don't feel that simply using aggregate parameters such as MRB and r is sufficient to discuss this evident low bias. I don't think that the authors can conclude that "the MB are fairly small" (line 676). Moreover, I'd expect the authors to play with the parameters that they can change (e.g. refractive index) to try and address this bias and correct the method. At this stage, from this plot I cannot agree that closure can be claimed between the remote sensing and the in-situ This had been highlighted in my previous review (MAJOR points 6 and 8, not successfully addressed).

Thank you for the comment. We feel that the quantitative consistency statistics used in this study are useful for this type of analysis. While some trends seem obvious from scatterplots, the quantitative comparisons can often be surprising. That being said, we have rephrased our language throughout the manuscript to be more clear what these metrics are actually indicating. As was discussed in previous comments, our current revisions shows marked improvement in all consistency statics used, which is largely due to adding in spheroidal approximation for the coarse aerosol shapes. We feel the improved correlations and reductions in biases do support the findings as they are stated in the current revision.

- Lines 676-677: "it is observed that the MB  $\pm$  SB are fairly small and range from  $1 \pm 11 \, \text{Mm}^{-1}$  to  $31 \pm 31 \, \text{Mm}^{-1}$ , which suggests that the MRB are partially inflated by low signal or noisy conditions". ??? Thank you for the comment. We removed this text.
- Lines 682-683: "low correlation does not necessarily imply a lack of agreement.". I am puzzled by this statement because figure 9h clearly shows that the LR calculation is not responding to the microphysical parameters (it is limited between 25 and 50). Perhaps the parameters used are not sufficient to model the LR. I would expect the authors to comment more critically on this point.

Thank you for the comment. We agree that the previous version of the manuscript showed very poor LR consistency. As discussed in several previous comments, our current analysis show much better agreement in LR. We still offer a critical view on these comparisons and think there is room for improvement in LR and LDR; however, there are several limitations that include 1) the limited number of particle shapes available in our optical library, 2) the in-situ instrument suite does not sample coarse optical properties, and 3) sampling coincidence between can be difficult to get perfect and can always introduce uncertainty.

- Lines 699-701: "sensors is likely due to the loss of particles from the diameter cutoff of the inlet and through the in-situ sampling pathways as discussed in the Introduction and undersampling of the coarse aerosol particles by the CAS, CDP, and FCDP. Although in-situ values are lower than the HSRL-2 ones, reasonable agreement is evident by the MB ± SB ranges." as already discussed, I thinbk that these statements are not supported by the scatter plots in figure 9

Thank you for the comment. We agree with this statement. In addition to improving these comparisons, we changed the text as follows to offer more explanations: "This discrepancy with the remote sensing retrievals is possibly due to difficulties in data coincidence, due to loss of aerosols from the diameter cutoff of the inlet and through the in-situ sampling pathways as discussed in the Introduction and undersampling of the coarse aerosol, or due morphologic and composition complexities.".

- Line 722: over the Atlantic, XXX km East of the coast of North Carolina
   Thank you for the comment. We added "On this day, a "unicorn" module was performed over the Atlantic that was
   ∼450 km east of the coast of North Carolina.".
- Line 733: likely to be
   Thank you for the comment. We added "likely to be".
- Figure 11: "Figure". the case study figures 11 and 12 show a lot of promise and are more convincing than the overall external consistency statistics of section 3.1.3. I suggest to show the case study before section 3.1.3, so as

to persuade the reader about the method, and to be more critical in section 3.1.3 highlighting why the method is not universally as good as for the case study

Thank you for the comment. This change has been made. The sections have been reorganized to facilitate this change.

- Line 751: "the near surface aerosol are distinctly different". we have no way to say that the aerosol type is different; only that the Cext/Cbsc is larger.
  - Thank you for the comment. We do agree that the LR is larger. We changed the sentence as follows for clarity: "From the  $1064 \,\mathrm{nm} \, C_{\mathrm{ext}}$  and the 355, 532, and  $1064 \,\mathrm{nm} \, C_{\mathrm{bsc}}$ , it is possible that the near surface aerosol are likely different in size from the next highest data point  $(300 \,\mathrm{m})$ ."
- Lines 751-752: "is no available comparison with the ISARA-derived properties as the Falcon did not sample that low in the atmosphere." This raises an important question: how do you account for layers not sampled < 238 m and > 4499 m, when doing the comparison with remote sensing columnar properties (e.g. AOD and variables in Table 7)? could you attempt using the lidar profile to quantify the correction for those layers that are not sampled in situ? Thank you for the comment. As the goal of this work is to establish consistency rather than complete closure, we do not expect to retrieve AOD. We added the following text for clarity: "Note that for AOD, the ISARA-derived ambient extinction is vertically integrated with sample altitude and does not account for the altitude bins above or below the sample altitudes. This allows us to estimate the amount of the total aerosol column that was sampled by the Falcon."

**Response to Reviewer 4**

We thank the reviewer for their feedback. We have taken care to respond directly to each comment. The text from your com-865 ments are shown in black, and our responses are shown in blue. The responses include the manuscript text that was changed, removed, or added.

The authors reported the ISARA retrieval framework in detail and attempted to validate it with ACTIVATE mission data. I appreciate the plethora of information re: the approach and the comparison datasets. However, I found the logic to be confusing 870 at times despite the high level of details (see my specific comments and questions below). In addition, it would be helpful if the authors validated the retrieved aerosol properties (kappa) with any available aerosol chemical composition from the same mission. Perhaps it is because the manuscript is rather long, there are quite a few typos and odd sentence structures. This manuscript would benefit from a major revision that I hope will address my questions and comments below.

Thank you for the comment. We have attempted to clarify and improve our logic throughout the text. We reorganized the paper 875 to include a significant primer on aerosol properties. Additionally, we have extended our synthetic data analysis to account for variations in the size distribution. We have also reorganized the results to present the synthetic closure before the internal closure and the case studies to be before the generalized external closure. We also improved our consistency metrics throughout, which is primarily as a result of the efforts made to account for nonspherical coarse aerosol. In our extensive revision, we feel we have successfully addressed the majority of your comments. We would like to directly address why we are not analyzing data from the ACTIVATE AMS. That data would add even more length that you have stated is already a long paper. Additionally, there is currently a separate paper that is under review analyzing the differences between the different hygroscopicity parameterizations. The convenience of the f(RH) derivation is that the retrieved hygroscopicity is internally consistent with the optical coefficient measurements.

**General comment:**

- For particle diameters > 1 μm, the authors should consider reporting them in μm instead of nm (e.g., 3488 nm on L281). Thank you for the comment. We followed your suggestion and reported um instead of nm for diameters above 1 um throughout the text.
- 890 - The authors should consider updating the symbol of surface area concentration from dA/dlogD to the conventionally used name of dS/dlogD.
  - Thank you for the comment. We replaced "dA/dlogD" with "dS/dlogD" throughout the text.
- In a few places (e.g., L427), the authors used the term "aerosol particles" but all aerosols are particles. Thank you for the comment. An aerosol is defined as a mixture of a solid or liquid particle in a gas (see https: 895 //glossary.ametsoc.org/wiki/Aerosol). While it is common to use aerosol to refer to just the particles, this needs to be made clear within the context of our study. We initially refer to the particles as "aerosol particles" to provide context to our study, however refer to them as just aerosol in sections where particle is an ancillary word.
- 900 - There are a lot of repeated definitions of acronyms (e.g., HSRL-2 on L233, Table 2 title and content) that should be cleaned up. There are also instances of using acronyms first then defining them later (e.g., CRI appeared on L156 then defined on L177).
  - Thank you for the comment. In the case of CRI, it was defined in full on Line 100. We have removed the redundant definition on Line 177.
  - Was there relevant aerosol chemical composition available from the ACTIVATE mission (e.g., AMS) that the authors could have used to compare with derived hygroscopicity parameter kappa and real refractive index? Thank you for the comment. There is AMS data available as part of the ACTIVATE dataset; however, that data would

- add even more length to what you have stated is already a long paper. Additionally, there is currently a separate paper analyzing the differences between the different hygroscopicity parameterizations. The convenience of the f(RH) derivation is that the retrieved hygroscopicity is internally consistent with the optical coefficient measurements.
  - Specific comments/questions:

 L120-122: the authors already defined the diameter range for fine mode and coarse mode particles on L95-97, so remove them here.

Thank you for the comment. These repeated definitions have been removed.

- L166: define what phase function represents and why it is introduced
   Thank you for the comment. We added the following: "Within the angular scattering matrix, the first element is the phase function (a1), which describes the angular distribution of scattered light intensity by a particle. It is normalized over all directions so that it represents the probability of scattering into a given direction. In the context of this study, it defines the angular scattering characteristics relevant to the viewing geometries of the lidar, polarimeter, and in-situ instruments. Note that only a1 is considered here, as polarization effects are beyond the scope of this work."
  - L205-211: is it necessary to list out the dates of the ACTIVATE deployments instead of referencing the campaign overview by Sorooshian et al. (2023)? This information does not seem to be used later.
     Thank you for the comment. We agree, this information is extraneous. We have removed this information.
- L239-247: a lot of repeated information on HSRL-2 products. Please rewrite this part of the paragraph to be more concise.
   Thank you for the comment. This repeated information has been removed and the paragraphs have been combined.
- L274: extra words "the 30"
   Thank you for the comment. The sentence now reads: "The LAS sampled aerosols that were actively dried with a 6" Monotube dryer (Perma-Pure, Model 700) for all flights except those conducted between 14 May and 30 June 2021, which were only dried passively."
- L275: what is "ram heating" in the context of SMPS data?
   Thank you for the comment. This passive drying occurs as a result of the isokinetic inlet and the air increasing in speed as it is sampled through that inlet. To clarify this information, we updated the text as follows: "All SMPS data rely exclusively on passive drying from ram heating (i.e., air speeding up as it enters the aircraft inlet) and a generally warmer cabin temperature than ambient air. In addition to passive drying, the aerosols sampled by the LAS are actively dried with a 6" Monotube dryer (Perma-Pure, Model 700) for all flights except those conducted between 14 May and 30 June 2021, which were only dried passively."
  - Table 1: in HSRL-2 section, missing a comma between "532" and "1064 mn". Also, repetitive definitions of HSRL-2 and RSP in both the table title and the table content. Consider adding acronym of extinction coefficient, backscatter coefficient, etc. in parameter description for clarity.
- Thank you for the comment. We removed "1064" as 1064 nm LDR is not an HSRL-2 product. We removed the repeated HSRL-2 and RSP definitions from the title but did not remove them from the table as otherwise there would be excess white space and we feel it is not detracting from the table's information or purpose.

- L291: "success rate" of...what exactly? Please be specific. Is it in retrieving coarse-mode particles?
   Thank you for the comment. We are referring to the success rate of ISARA for a given set of data. We have clarified this as follows "The successful retrieval rate (i.e., the success rate) of ISARA is defined as the number of points where the required fitting thresholds are met divided by the number of data points with measurements required to perform the retrieval (successrate = number of successful retrievals number of attempts)."
- L319: extra word "are"
   Thank you, we have removed "are" from this sentence.

- L339: water vapor "density" or mixing ratio?
   Thank you for the comment. You are correct, this should be mixing ratio. We replaced "density" with "mixing ratio".
- L346: travel "up to" 8 km?
   Thank you for the comment. We clarified the statement as follows: "The 45-second resolution results in a spatial resolution of 8 km, which assumes a ground speed of 180 m s-1."
- L369-370: repeated definitions of no, ao, vo (previously defined on L190-197). This repeat of definitions also happened to other acronym and symbols (e.g., Table 2 title). Please re-check the other sections of the manuscript as well.
   Thank you for the comment. We removed redundant definitions throughout the manuscript.
- L383: why did the authors restrict calculated absorption coefficients to within 1 Mm-1 of the measured values? Restricting the calculated values within a constant 1 Mm-1 may bias the filtering process to higher absorption values, where 1 Mm-1 difference is a smaller fraction of the total value (e.g., 10% of 10 Mm-1) vs. smaller absorption values (e.g., 50% of 2 Mm-1). Should this restriction be a percentage difference like with scattering?

  Thank you for the comment. This process was originally developed by Sawamura et al. (2017), which is where these thresholds ware first used. In the context of the current work we have already made several improvements over the methods developed by Sawamura et al. (2017), and want to limit the changes made in this iteration. As such, we feel it is appropriate to continue using their proven thresholds rather than investigate using our own. In future iterations we may be able to use optimal estimation, but it is outside of the scope of this study to investigate how that would change the results of the algorithm. Additionally, this change would only impact the internal consistency and would not have impact on the external consistency. We have made no changes based on this justification.
  - L427: the authors stated that the derived hygroscopicity for aerosol in the ACTIVATE region is low ( $\leq 0.1$ ) but also concluded that the aerosol population is sulfate-dominant. This value of seems really low. Organic is often assumed to have kappa value of 0.1. Sulfate (inorganic) are usually with kappa > 0.4. An aerosol population dominated by sulfate and organic as stated by the authors should have kappa > 0.1.
    - Thank you for the comment. Echoing our comment response to RC3 regarding this topic, we agree this is not very representative of pure sulfate aerosol species but common for mixtures of organic aerosol species with sulfate aerosol species. We changed the text as follows to capture this information:

The commonly observed low absorption is expected given the frequency of sulfate and secondary organic aerosol Nakayama et al. (2015). That being said, the observed IRI is between 0.01 and 0.08 in 32% of the data, which indicates the presence of moderately absorbing aerosol species such as aged smoke and dust. The observed low hygroscopicity in many of the retrievals also indicates organic aerosol species are present (Petters and Kreidenweis, 2007). Only 20% of the data had >0.2, which is the upper limit of the range of for organic aerosol species (Massoli et al., 2010).

L432: misspelling with extra "i" in scattering
 Thank you, we have fixed the spelling accordingly.

- Figure 3: this does not seem like a correct representation of the calculation process. On L415-420, you mentioned that  $C_{\rm RH,meas}$  was actually calculated from  $C_{\rm dry,meas}$  and estimated gamma from Eq. 7. Really, the authors were using a single measured  $C_{\rm RH,meas}(85\%)$  and  $C_{\rm dry,meas}$  to estimate gamma. Then, they used gamma to derive  $C_{\rm RH,meas}$  at 80% - which now is an estimated value and not a measurement. Why didn't the authors use  $C_{\rm RH,meas}(85\%)$  from the nephelometer instead?

Thank you for the comment. We do not use  $C_{\rm RH,meas(85\%)}$  directly as the RH associated with that measurement is not reported. To reduce the uncertainty with what exact RH to prescribe, we opted to used the parameterization to fix the RH to 80% for the retrievals. We feel no changes are needed.

- L452: extra word "aerosol" or "particle"
   Thank you for the comment. We removed "particle".
- L461-462: typo with "," between number and μm
   Thank you for catching this typo. It has been corrected.

- L458-464: the use of CAS size distribution to derive coarse-mode particle properties contradict with L320-325 in section 2.3.2, where the authors said to use CDP mainly and CAS only if the other 2 datasets (CDP and FCDP) are not available. Please review and correct the appropriate section. Also, if the author chose CDP over CAS, what is the reasoning when CAS provides a wider range of available aerosol diameters?

Thank you for the comment. To clarify, we do favor the CAS for coarse-mode aerosol properties. For cloud filtering, we favor the CDP.

- L506: the reported % does not need to be accurate to 2 decimal points (e.g., 26.49%), especially when the authors reported the biases as full % point (L587).

Thank you for the comment. While we do feel it is appropriate to truncate the bias estimate, we feel that significant figures are important when considering the length of the data set. In the updated version of this manuscript the number is now 9.26% this translates to the actual number of failures which is 919. If we truncate 9.26% to 9.3%, we get a 923

that is not the true number. We feel no changes are needed.

- L515-517: why did the authors remove HSRL-2 AOD < 0.08 where HSRL-2 AOD uncertainty is 0.02 (stated in Table 1)?
  - Thank you for the comment. To clarify, The HSRL-2 lowest limit of detection is 0.08. The following text has been added: ", which is the detection limit for the HSRL-2".

L521-524: I did not follow the logic. What is this 50% AOD difference criteria [between HSRL-2 and RSP derived AODs] based on? Similarly, how did the authors arrive at the filtering criteria of coarse-mode AOD being <0.1?</li>
 Thank you for the comment. This 50% AOD difference criteria criteria has been used in previous works (see Schlosser

- et al., 2024), and limits the possibility that the HSRL-2 profile is not comparable to the closest RSP data point. In this case the data point is thrown out, which removes it from the data set. In response to the second part of your comment, in the most recent draft we removed the coarse-mode AOD threshold as it did not seem needed for this study which can benefit from having more coars-mode aerosol comparisons. As such, the external consistency process has been updated.
- L555: extra "respectively"
   Thank you for catching this error. We removed this extra instance of "respectively".
  - Figure 5: the authors could add lines that represent the range of measured data from Figure 1 on Figure 4 to show the range in synthetic data is smaller than the actual measurement (L578-579). Also, there are currently 2 panels (b) in this figure.
- Thank you for the comment. We have added dashed magenta lines for the 10th and 90th percentiles from the ACTIVATE data set. We also corrected the panel labels.
  - L587: did the authors mean "respectively" instead of "respectfully"?
     Thank you for this comment. We replaced "respectfully" with "respectively".

- Table 3: update the table title since it also includes information for Table 4. Also, the authors repeated the same information in the first paragraph immediately following Table 3.
   Thank you for catching these errors. We updated the table's title as follows:
- Ancillary data for each of the 10 case studies. Ancillary information includes the case number, the profile start and stop times, the associated RSP sample time, the minimum and maximum altitudes sampled by the Falcon (i.e., in-situ) aircraft, the number of smoke counts above 2.5 km identified by the HSRL-2 typing product, the HSRL-2-derived LDR, the horizontal separation between the Falcon and King Air, and the latitude and longitude of the center point of the Falcon's vertical profile. All dates and times are provided in coordinated universal time (UTC) and in the format "year-month-day" and "hour:minute:second", respectively.
  - L699: remote sensors or remote sensing retrievals?
     Thank you for this comment. We changed "remote sensors" to "remote sensing retrievals" for clarity.
- Figure 11: why don't we have LR > 2 km for HSRL-2 data? Both  $C_{\rm ext}$  and  $C_{\rm bsc}$  are available at these altitudes for HSRL-2 (panel b, e). Also, panel (i) seems to be missing from Figure 11.

  Thank you for the comment. This is because LR is not reported at those low  $C_{\rm ext}$  and  $C_{\rm bsc}$ .
- L745: compared to case 7, case 9 with high smoke counts and high HSRL AOD would be interesting to investigate further compared to case 7.
   Thank you for the comment. We have added a set of figures and tables for case 9 to contrast with case 7. We have also added corresponding results.
- Figure 12: the authors should add uncertainty to the calculated kappa, RRI, IRI that were derived with ISARA.
   Thank you for the comment. We added error bars for the uncertainties determined in the synthetic consistency analysis.

The error bars associated with RRI are too small to be noticeable.

Conclusion: I am not sure why the authors defined ISARA, CRI, etc. again when they had been mentioned all throughout the manuscript. Please remove this redundant information.
 Thank you for the comment. We removed the redundant information.

L792: Similar to the work...
 Thank you for catching this error. We corrected this.

---

## Author Response (AR3)

**Response to Editor**

We thank the editor for their feedback. We have taken care to respond directly to each comment. The text from your comments are shown in black, and our responses are shown in blue. The responses include the manuscript text that was changed, removed, or added.

Thank you for your revisions to the manuscript. The reviewer recommends minor revisions (edging toward major) and points out some issues of clarity that need to be resolved. One primary one is some confusion about AOD calculated from the HSRL in the cases of partial profiles where the in situ aircraft did not reach higher altitudes. Please be very clear in the manuscript how you are integrating the HSRL profiles and ensure that they are consistently calculated in values that are quoted in the text and tables. I would also like for you to remove the section on the Stokes parameters, which reads as a bit of tutorial and is not really discussed again in the manuscript (some of the terms do appear in later equations). You can place the Stokes discussion in the Appendix.

We addressed each of the points in this comment in the below "Detailed Comments" section. We reaffirmed that the ISARA-AOD is calculated with only the available data and is not directly comparable to the total column AOD derived from either the HSRL-2 or the RSP. Additionally, we moved the Stoke's vector discussion to Appendix A.

There were numerous typographical errors which I detail below. Each of them individually is small but they add up to a significant concern in the readability of the manuscript, especially in the case of missing words in sentences. Please address these carefully, proofread thoroughly, and run a spelling and grammar check prior to submission of a revised manuscript.

After addressing each of the comments, we made sure to carefully proofread this draft. We apologize for the frequency of typos in the previous version.

I find this work to be scientifically very useful and interesting. As I know from my own work, consistency analysis using multiple techniques and platforms is challenging. While the publication process has been long and trying, I think the end result will be a very solid contribution to the field. Thank you for your efforts.

Thank you for all of your effort in this peer revision of this manuscript.

**Detailed comments:**

1. When effective radius and effective variance are first used in the text, please refer to the appropriate equations in Appendix A.

We added references to Equations B4 and B5 where these terms are first discussed in the text.

2. Line 239. Incomplete sentence.

We completed this sentence as follows: ", and  $n^{\circ}$  is the logarithmic size-resolved aerosol number concentration".

- 3. Line 240. "The term (blank) is used per convention to represent...." We added the missing term " $n^{\circ}$ ".
- 4. There is no need to fully describe the Stokes parameters (lines 195-217). This information could be placed in Appendix A or in the Supplemental Material.

As suggested, we moved this information to Appendix A.

5. Line 320 Define lidar ratio and refer to the appropriate equation in the Appendix A. We updated the text to include both the written out definition of lidar ratio and the reference to Eq. B9. The text has been updated as follows:

Unlike standard elastic backscatter lidars such as Cloud-Aerosol Lidar with Orthogonal Polarization (CALIOP), the HSRL-2 has the ability to measure total and molecular backscatter separately from which aerosol backscatter and extinction can be derived (Hair et al., 2008). The HSRL-2 measurements can also be used to derive the ratio of aerosol extinction to backscatter coefficients, i.e., the lidar ratio (LR; see Eq. B9), as well as the linear depolarization ratio (LDR; see Eq. B10) (Burton et al., 2016, 2018).

6. Line 335 RRI already defined.

We remove this redundant definition.

7. Lines 338 to 341. I believe the "native" reported resolution of the HSRL is 1s, which is then averaged to 60s. I cannot access the data files to confirm.

The HSRL-2 team has confirmed that the publicly available data are subsampled from the native resolution. No changes needed.

- 8. Line 348. Do you not mean 225 x 15 m as the size of each bin for which you determine smoke counts? This is written correctly as is. 60 s is the native horizontal resolution and 225 m is the native vertical resolution. For clarification, we added "(vertical resolution × horizontal resolution)" where this convention is first used.
- 9. Line 371. I believe the optical diameter (approximately the geometric diameter) is the aerodynamic diameter \*divided\* by the square root of (the relative density times the inverse of the shape factor). Thus it is less, not greater, than the aerodynamic diameter. This might affect your choice of maximum size bin when you integrate size distributions to compare with the aerodynamically filtered optical measurements.

Thank you for the comment and for the correction. We have removed this text. This cutoff choice was governed significantly based on the success rate of ISARA observed in the internal consistency results.

- 70 10. Line 387. Change "dependant" to "dependent". We changed "dependant" to "dependent".
  - 11. Line 402 "AS" should be "As". We changed "AS" to "As".

12. Line 485 change "an" to "and". We changed "an" to "and".

- Line 505 change to "reported by Nakayama et al."
   We added "as reported by".
- 14. Fig. 4. Specify that f(RH) is at 80% RH in the caption or legend. (Figure 9 also.) We added "550 nm f(RH) at RH = 80%" to Figs. S9 and S14.
- 85 15. Line 522. Sentence ends with "using".

  We changed the text to "...using the standard temperature conversion factor provided with the data.".

16. Line 524. Change to "properties were examined". We changed the text to "properties are examined". 17. Lines 527-539. You may want to present these as a commented, in-text list rather than as a numbered list to help save We changed the list format to be in-text. 18. Line 539. Define NRMSD (refer to equation in appendix). We defined NRMSD and added the reference to Eq. C8. Where applicable, we also added references to associated equations for rest of the terms in this list. 19. Fig. 5 caption. Change "meats" to "meets". We changed "meats" to "meets". 20. Line 595. Change "synthetic in situ measurements of dry" to "synthetic dry in situ measurements". We changed "synthetic in situ measurements of dry" to "synthetic dry in situ measurements". 21. The sentence spanning lines 596 to 597 needs a verb. To address this, we changed "Measurement noise bounded..." to "Measurement noise is bounded...". 22. Line 617. Change "idyllic" to "ideal". We changed "idyllic" to "ideal". 23. Line 621. By different "transport paths of aerosol" do you mean spatial and temporal heterogeneity? Or something about plumes? This is a little confusing. We were referring to the spatial and temporal heterogeneity. To address this, we changed "transport paths" to "spatial and temporal heterogeneity". 24. Line 629. Mis-spelling of "closest". We corrected the spelling of "closest". 25. Line 641. "Controlled" ascent or descent? I hope they were controlled! We removed "controlled" from the text. 26. Line 643. Change "decent" to "descent" twice. We changed "decent" to "descent" in three places.

27. Line 645. It would be helpful to mention the typical ascent or descent rate of the Falcon, and the typical vertical distance

We replaced "airspeed to  $180 \,\mathrm{m\,s^{-1}}$ " with "rate of altitude change to  $\sim 150 \,\mathrm{m\,min^{-1}}$ " in both places of the text and added covered in a 45-s SMPS scan, to get a sense of the vertical resolution of the in situ measurements.

"The SMPS scans typically cover 112–113 m in a 45 second scan." to the end of this paragraph.

- 28. Line 649. I'm not sure how you average the ISARA values to the 225 m HSRL bins. Assuming an ascent or descent rate of 300 meters per minute (1000 feet per minute), each 45-s SMPS-based ISARA bin should be about 225 m, so you won't be averaging those to match the HSRL bins. Or do you mean that you average the HSRL data to match the ISARA bins? This might be backwards in the text.
- The Falcon measurements are tied directly to time but only indirectly to position through the combination of airspeed and ascent/descent rates. From a time perspective, the 60 second resolution of the HSRL-2 is larger than the SMPS scan of 45 seconds. Additionally, the Falcon's descent rate was closer to 150 m min-1, which further justifies the descision to average to the HSRL-2 profiles.
  - 29. Line 661. This sentence needs a verb and initial capital letter. We corrected this sentence as follows: "Additionally, we categorize points where LDR > 0.08 and LR > 35 as a distribution of spheroids, CRI = 1.52 + 0.0043i, and  $\kappa = 0.1$ ."
  - 30. Line 684. Please change "Section B" to "Appendix B". We changed "Section" to "Appendix".

- 31. Lines 685-686. I don't know what "lost" values mean, or how the 97.4% success rate was calculated. Please clarify both. We did define the success rate metric with Eq. 13. By "lost" we were referring to failed retrievals. We clarified the text as follows: "Out of all possible retrievals, 2.60% failed the  $\kappa$  but had a successful IRI retrieval. This retrieval success rate (see Eq. 13) is 97.40%, which is close to the 95.84% success rate observed in the ACTIVATE retrieval of  $\kappa$ ."
- 32. Line 722. Capitalize "size". This sentence is nonsensical. We removed this sentence.
- 33. Line 765. When comparing the HSRL-2-integrated AOD with the ISARA derived values, are you integrating the HSRL 2 over the altitude range of the profile shown in Table 3? This is not stated anywhere in the text where I can find it, and would logically go in this paragraph.
  - We would like to note that we do not aim to represent the ISARA-derived AOD as a total column AOD. We made effort to clarify this in Sect. 4.5, which has text that states "Note that for ISARA-derived AOD, the ambient extinction is vertically integrated with sample altitude and does not account for the altitude bins above or below the sample altitudes. This allows us to estimate the amount of the total aerosol column that was sampled by the Falcon.". For further clarity we changed the second sentence to "This allows us to estimate the amount of the total aerosol column that was sampled by the Falcon and compare that to the standard column AOD products derived from the HSRL-2 and the RSP" after "...Falcon".
- 34. Line 784. You may want to mention that this was the deepest profile conducted by the Falcon with viable data for comparison with the HSRL-2.
  - To capture this information, we removed the last sentence and put in another point for this list. The text added is "this profile extends more than 4 km, which is the largest extent of any profile by 2 km over the next largest profile".
- 170 35. Line 790. Remove the hyphen between "vertically-sample". We removed the hyphen.

36. Fig. 13, panel I, x-axis needs units.

We added units to the x-axis of panel (i) on Figs. S18 and S5.

37. Line 863. "are considered to be in good to moderate agreement this profile, which depends on the property" needs to be more understandable.

We changed this sentence to be "The ISARA- and RSP-derived fine properties are within  $2 s_{\text{weight}}$  of their respective uncertainty estimations for this profile. Fine  $r_{\text{eff}}$ , IRI, and spectral SSA are all within  $1 s_{\text{weight}}$  of their respective uncertainty estimations." We also reorganized this paragraph and the subsequent paragraph for clarity.

38. Line 870. Change to "One possibility is that there are some..."

We changed "One possibility there are some..." to "One possibility is that there are some...".

39. Line 874. Sentence is incomplete.

We added the term " $\kappa$ " to complete this sentence.

40. Line 910. Change "art all" to "at all". We changed "art all" to "at all".

- 41. Line 942. "Additionally The range is 15—19%, with the lowest observed in the 532 nm wavelength." Please remove the unneeded capitalization of "The". Also, it is not clear to which parameter the range of 15-19% refers. We replaced "The" with "the". The missing term "NRMSD" was also added.
- 42. Line 944. Change "This low bias result is also..." to "Similar low bias results are also...". We changed "This low bias result is also..." to "Similar low bias results are also...".
  - 43. Fig. 15. The figure caption does not describe panels j, k, or l. We changed the caption as follows:

Scatterplots of the following altitude-resolved aerosol properties: (a)  $355 \,\mathrm{nm} \, C_{\mathrm{ext}}$ , (b)  $532 \,\mathrm{nm} \, C_{\mathrm{ext}}$ , (c)  $1064 \,\mathrm{nm} \, C_{\mathrm{ext}}$ , (d)  $355 \,\mathrm{nm} \, C_{\mathrm{bsc}}$ , (e)  $532 \,\mathrm{nm} \, C_{\mathrm{bsc}}$ , (f)  $1064 \,\mathrm{nm} \, C_{\mathrm{bsc}}$ , (g)  $355 \,\mathrm{nm} \, \mathrm{LDR}$ , (h)  $532 \,\mathrm{nm} \, \mathrm{LDR}$ , (i)  $1064 \,\mathrm{nm} \, \mathrm{LDR}$ , (j)  $355 \,\mathrm{nm} \, \mathrm{LR}$ , (k)  $532 \,\mathrm{nm} \, \mathrm{LR}$ , (l) N, using  $2020-2022 \,\mathrm{ACTIVATE}$  data. The blue points are data from vertical profiles that contained 3 or more points, the red points are data from all vertical profiles, and the dashed line represents the one-to-one line. The error bars shown indicate the standard deviation of a given aerosol property. The consistency statistics for these data are shown in Table S10 for profiles with 3 or more data points and in Table S3 for data from all profiles.

44. Line 975. Change "is more" to "are more...". We changed "is more..." to "are more...".

- 45. Line 980. Please fix "range is 0.39–0.74 that decreasing with increasing wavelength". We replaced "decreasing" with "decreases".
  - 46. Line 986. Change to, "These results show relative biases that are..." We replaced "These results have a relative bias that are..." with "These results show relative biases that are...".

- 47. Line 1012. Please change "Section B" to "Appendix B" We replaced "Section" with "Appendix".
- 48. Line 1030. "(i.e., )".

  We changed "(i.e., )" to "(i.e., SRB ≈ 20%)" and put it at the end of the sentence.
  - 49. Lines 1049. "Consistency" We changed "consistent" to "consistency".
- 50. Line 1050 -1050. Please fix "...optical N Schlosser".

  Thank you for the commment. We changed "...optical N Schlosser et al. (2022)." to "...optical N (Schlosser et al., 2022).".
- 51. Supplemental Materials: Please state that Table S3 is for Case 9.
   Table S3 is for all data without filtering and not Case 9. We have clarified the caption as follows: "Consistency statistics that result from the comparisons of altitude-resolved spectral Cext, Cbsc, and LR, as well as N. The consistency statistics shown correspond to the unfiltered data set. The unfiltered data set are shown as gray circles in the scatterplots shown on Fig. S20.".
- 52. References: Please ensure that all references comply with the Copernicus guidelines. For example, many of the references use capitalization for the article titles, which is not consistent with the Copernicus standards. This saves the time for our copy-editing staff and helps speed the production of the manuscript.

  We have reviewed the references and corrected the ones that were not consistent with the Copernicus standards. The references appear to be correct now.

**240 Response to Reviewer 4**

We thank the reviewer for their feedback. We have taken care to respond directly to each comment. The text from your comments are shown in black, and our responses are shown in blue. The responses include the manuscript text that was changed, removed, or added.

- I would like to thank the authors for addressing some of my previous comments. For the new manuscript, there are enough typos that I would advise the authors to proofread more carefully next time (e.g., L980). In terms of content, despite the authors providing more explanation, I still find the manuscript unclear at times. Below are my questions and comments that suggests this manuscript needs at least MINOR revisions (I think it really sits right between MINOR and MAJOR revisions), but I think the authors can address the following relatively quickly.
- 250 We have carefully addressed each of your general and specific comments to add clarity where it was lacking. We also made sure to carefully proofread this revision. We apologize for the frequency of typos in the previous version.

**General comment:** I am not clear in the way the authors set up the synthetic consistency analysis. Specifically, in response to Reviewer 3's comment on expanding the variety of synthetic size distributions, the authors gave Figure 5 as an example of a bimodal size distribution that appears unimodal to me. Are there other examples of size distributions that the authors could offer for different layers of the sampled atmosphere?

We have replaced the previous example with one that has a more distinct bimodal size distribution. We also added the sentence "There are 473 good fits identified that comprise the final set of fitted size distributions." to Section 4.4. While some of these size distributions of this set do not appear as biomodal, the fitting constraints are still met. We believe that this provides an adequate variety of "real world" size distributions without needing extraneous information such as aerosol species or sample altitude.

Figure 15 shows a clear high bias in remote-sensing retrieved values (e.g., C\_ext) compared to the ISARA values that the authors did not explain well. Besides the reported statistics, the authors seem convinced that the bias is from under-sampling of coarse-mode particles. Given the high uncertainty in retrieved kappa and refractive index values, I would be more convinced in the bias coming from one of these retrieved values.

We would like to point out that we do mention this bias exists in two places in this section:

These MRB indicate that the in-situ data is biased low from the HSRL-2, showing that the in-situ instruments retrieve lower values of  $C_{\rm ext}$  than the HSRL-2 throughout the ACTIVATE campaign. Similar low bias results are also seen in Sawamura et al. (2017), which are MRB = 31% and 53% for California and Texas, respectively. The SRB for those cases are 5% and 11%, respectively. Compared to that work, we demonstrate marked improvement in our observed  $C_{\rm ext}$  consistencies.

and

It is observed that there is a systematic underestimation between ISARA- and HSRL-2-derived 355 and 532 nm  $C_{\rm ext}$  and  $C_{\rm bsc}$ ; however, this is more important at lower signals ( $C_{\rm ext} < 50\,{\rm Mm^{-1}}$  and  $C_{\rm bsc} < 1\,{\rm Mm^{-1}sr^{-1}}$ ). This discrepancy with the remote sensing retrievals is possibly due to difficulties in data coincidence, due to loss of aerosols from the diameter cutoff of the inlet and through the in-situ sampling pathways as discussed in the Introduction and undersampling of the coarse aerosol, due morphologic and composition complexities, or due to limitations of the hygroscopicity parameterizations. Although in-situ values are lower than the HSRL-2 ones, reasonable agreement is evident by the MB ranges.

We have also added the hygroscopicity parameterization as a source of error: "This discrepancy with the remote sensing retrievals is possibly due to 1) difficulties in data coincidence, 2) due to loss of aerosols from the diameter cutoff of the inlet and through the in-situ sampling pathways as discussed in the Introduction and undersampling of the coarse aerosol, 3) due morphologic and composition complexities, or 4) due to limitations of the hygroscopicity parameterizations.". Finally, we also mention this information among the list of limitations in the conclusion as well.

I am still unclear on the uncertainty associated with the gamma parameterization. Is there a reported uncertainty somewhere on the manuscript that I may have missed?

Because calculating  $\gamma$  is not a straightforward process, there is no generally acceptable way to quantify this error. This error is assumed to be much smaller than the 10% and 2 Mm-1 scattering coefficient error and was not relevant in previous similar works that relied on this parameter (e.g., Ziemba et al., 2013; Sawamura et al., 2017).

**Specific comments/questions:**

- 1. L204: The ACTIVATE "mission" or "field campaign". The authors' response of to a similar comment made by Reviewer 3 was not satisfactory. On L110 in the Introduction section, the authors already used the term "ACTIVATE field campaign" as well as "ACTIVATE mission" on L138, so there should be no issue to use something similar on L204. "The ACTIVATE featured..." sentence reads awkwardly, otherwise.

  We added "mission".
- 2. L212: Did you mean S\_I (theta) is the angular light intensity and not S\_1 (theta)? Note that this part of the text has moved to Appendix A. We corrected the notation here.  $S_I$  is the Stoke's vector. We changed this sentence to: "where  $S_1(\theta)$  is the angular light intensity and  $S_2(\theta)$ ,  $S_3(\theta)$ , and  $S_4(\theta)$  correspond to the proportion of light in various polarization states."
- 30. L369: "... 1 um aerodynamic cutoff for scattering coefficient measurements only." Then delete the next sentence. As the sentence is currently, it reads a little odd.

  We added "...for scattering coefficient measurements only." and removed the subsequent sentence.
  - 4. L432: change "an relative accuracy" to "a relative accuracy". We changed "an relative accuracy" to "a relative accuracy".
    - 5. L446-447: The following sentence seems vague to me: "As such, the external consistency analysis is most useful from vertical profiles where the in-situ platform samples the column of air above an arbitrary ground location." Instead of "above an arbitrary ground location," it would sound better to say "to an approximate altitude of > 230 m" (based on reported spiral altitudes in Table 3).

We changed "above an arbitrary ground location" to "to an altitude of 150-250 m".

- 6. L456-457: The first sentence ("As discussed above, ... for ACTIVATE 2021-2022.") summarized the same info from L365-380, so it is redundant and should be removed.
- We removed this sentence.

7. L490: Replace "measurement" with "value". You did not technically measure gamma but calculate it from the measured scattering coefficients.

We replaced "measurement" with "value".

8. Figure 4 caption: for (a) IRI...
We changed "of retrieved" to "for".

- 9. L525: Specify that "...properties are presented in Sect. A at the end of the main manuscript." Otherwise, it is hard to tell whether this Sect. A is in the SI document or somewhere else. I also think the authors can just remove the list of statistical metrics here since you already put it in Sect. A. You just need to write out the terms before using the acronym in the text.
  - We changed "Section" to "Appendix" for clarity. Additionally, we truncated the page space this list takes up by changing the format to in-text. We do feel this list saves effort in relating the acronyms to their definitions and equations without taking up more space further on in the text and has value.
  - 10. L553: Remove "if" from "..., if the CAS data are used." Or it would be best if the authors remove the sentences on L553-554. The authors are repeating what coarse mode aerosol data to use as stated previously on L428-429. We removed the sentences on L553-554 and added the word "primary" to the proceeding sentence.
  - 11. L585: Remove "With the ground truth size distributions generated,"

    We replaced "With the ground truth size distributions generated, we then..." with "Next, we...".
- 12. Figure 5: The example size distribution is unimodal, so wouldn't it better to show an example of bimodal size distribution?

We have replaced the previous example with one that has a more distinct bimodal size distribution.

13. L605: "is" adjusted
We added "is" before adjusted.

14. Table 4: Is HSRL-2 AOD from the reported HSRL-2 data that accounts for the entire sampled column from 9 km downward looking to the ground? The reported in-situ AODs are from Falcon's vertical profiles that mostly got up to 1-2 km based on Table 3. This is not an apple-to-apple comparison which Table 4 seems to imply. This goes back to the previous comment #6 from Reviewer 3 for the R2 version of this manuscript. I would suggest the authors to add a footnote or be explicitly clear in the text for Table 4 that there is a difference between the 2 AOD columns. Otherwise, the authors could add a "HSRL-2 equivalent AOD with in-situ profile" (or something to that effect) that only accounts for the sampled altitude performed by the Falcon (e.g., up to 1.47 km for case #9).

We did address this in the text where we stated "Note that for ISARA-AOD, the ISARA-derived ambient extinction is vertically integrated with sample altitude and does not account for the altitude bins above or below the sample altitudes.". For added clarity, we have repeated this statement to the caption of Table S7.

- 15. L684: Similar to Sect. A, specify that "...B6 and B7 in Sect. B at the end of the main manuscript." For clarity, we replaced "Section" with "Appendix" throughout the text.
- 365 16. L710: Extra space between "non-zero" and "measurements". We added the missing term " $n^{\circ}$ " between "non-zero" and "measurements".
  - 17. L731: ...there "are" still some cases We replaced "is" with "are".

- 18. L780-793: Mention Fig. 10 somewhere in these 2 paragraphs when you describe Case 7. We changed "the Falcon spiral began at 14:56 UTC and ended at 15:22 UTC, while the RSP sample time was at 15:19 UTC" to "As illustrated by Fig. S15, the Falcon spiral began at 14:56 UTC and ended at 15:22 UTC, while the RSP sample time was at 15:19 UTC".
- 19. L792: change "scene" to "scenario". We changed "scene" to "scenario".

- 20. Figure 10: Explain what the red and blue lines represent either in the caption or in the legend.

  We added "The blue and red lines represent the Falcon and King Air flight paths, respectively." to Figs. S15 and S1.
- 21. Figure 13: Panels (i, k, l) need units for the x-axis. We changed the caption as follows:

Scatterplots of the following altitude-resolved aerosol properties: (a)  $355 \,\mathrm{nm} \, C_{\mathrm{ext}}$ , (b)  $532 \,\mathrm{nm} \, C_{\mathrm{ext}}$ , (c)  $1064 \,\mathrm{nm} \, C_{\mathrm{ext}}$ , (d)  $355 \,\mathrm{nm} \, C_{\mathrm{bsc}}$ , (e)  $532 \,\mathrm{nm} \, C_{\mathrm{bsc}}$ , (f)  $1064 \,\mathrm{nm} \, C_{\mathrm{bsc}}$ , (g)  $355 \,\mathrm{nm} \, \mathrm{LDR}$ , (h)  $532 \,\mathrm{nm} \, \mathrm{LDR}$ , (i)  $1064 \,\mathrm{nm} \, \mathrm{LDR}$ , (j)  $355 \,\mathrm{nm} \, \mathrm{LR}$ , (k)  $532 \,\mathrm{nm} \, \mathrm{LR}$ , (l) N, using  $2020-2022 \,\mathrm{ACTIVATE}$  data. The blue points are data from vertical profiles that contained 3 or more points, the red points are data from all vertical profiles, and the dashed line represents the one-to-one line. The error bars shown indicate the standard deviation of a given aerosol property. The consistency statistics for these data are shown in Table S10 for profiles with 3 or more data points and in Table S3 for data from all profiles.

22. L816-817: "There also appears to be two noticeable nonspherical coarse aerosol layers within the column as evident from the spikes in LDR between 3 and 5 km" – does this refer to the vertical profile of LDR in Fig. 13c? I do not see any spike in LDR values in specified altitude range.

This paragraph was intended for Case 9 and has been moved to the appropriate spot in the text.

- 23. L819-820: "The lower layer of coarse aerosol have LR  $\approx$  35 sr. The upper layer of coarse aerosol has LR  $\approx$  45 sr, is less depolarizing, and appears to be more absorbing than the lower layer."
  - + Change "have" to "has".

We changed "has" to "have" in three places where we're referring to aerosol in this paragraph.

- + Fig. 12 panels (g) and (h) both show LR but at 2 different wavelengths, so please clarify which wavelength these sentences refer to.
- For clarity, this text is referring to Fig. S4.
  - 24. L867-899: "The latter of the two findings is evident from both the profile's limited vertical extent of 1.02 km and in Table 4, which shows that the ISARA-derived AOD is only 20% of the RSP- and HSRL-2-derived AOD for this profile." since the Falcon completed a limited spiral up to 1.02 km only, why not cut off HSRL-2 AOD value from altitude above maximum-sampled altitude by the Falcon? This way the only difference between HSRL-2 AOD and in-situ AOD would be the missing sampled altitude from the ground level to the bottom of the spiral.

As was mentioned in the previous comment, this text is referring to Case 9 and was moved to the appropriated place in the text. Additionally, we are not attempting to show that the ISARA-derived AOD can be made equivalent. It is

- useful to know that in Case 7, where the Falcon sampled a large portion of the column, we see good agreement between the AOD derived from the HSRL-2 and ISARA. We have also addressed this in the previous comment related to Table S7.
  - 25. L957-958: "The consistency statistics between HSRL-2- and ISARA-derived ambient LR and LDR are generally worse, relative to C\_ext and C\_ext,..." did you mean C\_ext and C\_bsc? We did intend to put  $C_{\rm bsc}$ . We changed the repeated " $C_{\rm ext}$ " to " $C_{\rm bsc}$ ".
  - 26. L981: "These ranges are the highest if any of the comparisons within this data set" this sentence does not make sense. We replaced "...respectively. These ranges are the highest if any of the comparisons within this data set" with "...respectively, and are larger than the values observed when comparing  $C_{\text{ext}}$ ,  $C_{\text{bsc}}$ , LR, and N.".
- 27. L 986: "Overall, the N comparisons are considered to be closed relatively successfully when..." do the authors mean "...comparisons are considered to be relatively successful when..."?

  We clarified the text as follows: "Overall, the N comparisons are considered to be consistent. These values are comparable to results of Schlosser et al. (2022)'s evaluation of HSRL-2+RSP-derived N using N derived from ISARA.".
- 28. Figure 15: Since the presented statistics in Table 7 and the discussion of plotted data in Figure 15 focus on the blue points (data from vertical profiles with 3+ points), I highly suggest the authors to plot the red points on Fig. 15 as greyed out points (gray points with alpha value < 0.5 if plotted with Python). Otherwise, the existing Fig. 15 panels are too busy on the eyes. The authors could also show a version of Fig. 15 with ALL red data points highlighted and the blue points grayed out in the SI document for clarity and to pair with Table S3.
- We have changed the color of the red points to gray and gave them a 50% transparency. The text referring to these points has been updated to reflect this change in appearance.
- + For the blue data points, do the outliers (e.g., elevated HSRL-2 532-nm C\_bsc values in panel (e) correspond to low/high altitude bins? I.e., is there any clear bias in sampling/retrieval when you look at ALL the blue points vs. just Case 7/9 earlier? Expanding on your explanation for the relative difference between HSRL-2 and ISARA values would add value to the manuscript. As it stands, the generalized study section reads more like a list of statistical values. We had originally colored the points displayed in Fig. S20 by altitude but did not observe any clear altitude dependence. We added the following text to capture this finding: "The biases in  $C_{\rm ext}$  and  $C_{\rm bsc}$  have no clear altitude dependence.". Additionally, we have already shown the distinct difference between Case 7 and Case 9 in the previous section. This information would seem redundant here. We discuss with some depth about the variety of limitation that our study has, and we cannot make any more specific statements regarding the reasons these relative differences occur.
  - 29. L1003: Remove "the" before "ISARA" We removed "the" from before "ISARA".

30. L1049: extra word "the" We removed the repeated word "the".

---

## Author Response (AR4)

**Response to Editor**

5

We thank the editor for their continued effort and feedback. We have taken care to respond directly to each comment. The text from your comments are shown in black, and our responses are shown in blue. The responses include the manuscript text that was changed, removed, or added.

Thank you for the changes to the manuscript that are responsive to the reviewer's concerns. I think this will be fine for publication following a couple of minor grammatical corrections:

- Lines 859-863: The word "layer" is the subject of these sentences, so please correct the subject-verb agreement (there "appear to be" layers; the lower layer "has a 532 nm LR of  $\sim$ 35 sr", the upper layer "has...", etc.
- We changed "appears" to "appear". We changed "have" to "has" for both of these instances and the two instances in Lines 865-867. Additionally, we reviewed the rest of the manuscript for similar subject-verb disagreement. We were unable to find any other issues.
- Line 865-867: The noun "aerosol" is singular. It is a system (singular) composed of a suspension of solid or liquid particles in a gas (a sea-spray aerosol, a biomass burning aerosol). So "the aerosol has". Alternatively you could say "the particles have" if you're referring to the particulate fraction of the system. You may wish to search the manuscript for other such circumstances. We changed "have" to "has" for both of these instances.
- I haven't checked thoroughly, but after moving sections around and changing equation numbering it is a great idea to get a fresh set of eyes on the manuscript to make sure that all of the equations are correctly referenced. There have been several modifications to this manuscript and it's easy for this kind of error to creep in.
  - We reviewed the equation references and did not find any mislabeling/numbering. No actions needed.
  - Thanks for correcting the reference formatting.
- 25 Thanks again for an interesting and informative manuscript.
  - You are welcome. Thanks for catching those formatting errors and for your continued effort in facilitating this peer revision process.